# Cryo-EM reveals how Hsp90 and FKBP immunophilins co-regulate the glucocorticoid receptor

Chari M. Noddings [1], Jill L. Johnson [2] & David A. Agard [1]✉

Hsp90 is an essential molecular chaperone responsible for the folding and activation of hundreds of 'client' proteins, including the glucocorticoid receptor (GR). Previously, we revealed that Hsp70 and Hsp90 remodel the conformation of GR to regulate ligand binding, aided by co-chaperones. In vivo, the co-chaperones FKBP51 and FKBP52 antagonistically regulate GR activity, but a molecular understanding is lacking. Here we present a 3.01 Å cryogenic electron microscopy structure of the human GR:Hsp90:FKBP52 complex, revealing how FKBP52 integrates into the GR chaperone cycle and directly binds to the active client, potentiating GR activity in vitro and in vivo. We also present a 3.23 Å cryogenic electron microscopy structure of the human GR:Hsp90:FKBP51 complex, revealing how FKBP51 competes with FKBP52 for GR:Hsp90 binding and demonstrating how FKBP51 can act as a potent antagonist to FKBP52. Altogether, we demonstrate how FKBP51 and FKBP52 integrate into the GR chaperone cycle to advance GR to the next stage of maturation.

Hsp90 is required for the functional maturation of 10% of the eukaryotic proteome[1]. Hsp90 'clients' are enriched in signaling proteins, such as steroid hormone receptors (SHRs), making Hsp90 an important clinical target[2–4]. SHRs, including the glucocorticoid receptor (GR), are hormone-regulated transcription factors that depend on Hsp90 for function throughout their lifetimes[5–10]. We previously established in vitro reconstitution of the 'GR chaperone cycle', revealing that GR ligand binding is inactivated by Hsp70 and reactivated by Hsp90 (ref. 11). In the GR chaperone cycle, understood in atomic detail through cryogenic electron microscopy (cryo-EM), GR ligand binding is regulated by a cycle of three distinct chaperone complexes[12,13]. In this chaperone cycle, GR is first inhibited by Hsp70 and Hsp40, then loaded onto Hsp90:Hop (Hsp70/Hsp90 organizing protein co-chaperone) forming an inactive 'GR–loading complex' (GR:Hsp70:Hsp90:Hop)[12]. Upon ATP hydrolysis by Hsp90, Hsp70 and Hop release, and p23 binds to form an active 'GR–maturation complex' (GR:Hsp90:p23), restoring GR ligand binding with enhanced affinity[13]. Cryo-EM structures of the GR–loading complex and GR–maturation complex revealed that Hsp70 and Hsp90 locally unfold and refold the GR ligand binding domain (GR$_{LBD}$) to directly regulate ligand binding.

In vivo, additional Hsp90 co-chaperones are found associated with the GR chaperone cycle[6,14], including the large immunophilins, FKBP51 and FKBP52 (ref. 6). FKBP51 and FKBP52 are peptidyl proline isomerases (PPIases) that contain an N-terminal FK1 domain with PPIase activity, an enzymatically dead FK2 domain, and a C-terminal tetratricopeptide repeat (TPR) domain, which binds the EEVD motifs at the C-termini of Hsp90 and Hsp70 (refs. 15–18). Additionally, the TPR domain contains a helical extension at the C-terminus (H7e), which binds the C-terminal domain (CTD) closed dimer interface of Hsp90 (refs. 19,20). Although FKBP51 and FKBP52 are 70% similar in sequence, these co-chaperones have antagonistic functional effects on GR in vivo[21]. FKBP51 inhibits GR ligand binding, nuclear translocation and transcriptional activity, while FKBP52 potentiates each of these fundamental GR activities[22–32]. FKBP51 and FKBP52 are also known to regulate the other SHRs[21,33]. Due to the critical importance of steroid hormone signaling in the cell, altered expression of FKBP51 or FKBP52 is associated with

[1]Department of Biochemistry and Biophysics, University of California, San Francisco, San Francisco, CA, USA. [2]Department of Biological Sciences, University of Idaho, Moscow, ID, USA. ✉e-mail: agard@msg.ucsf.edu

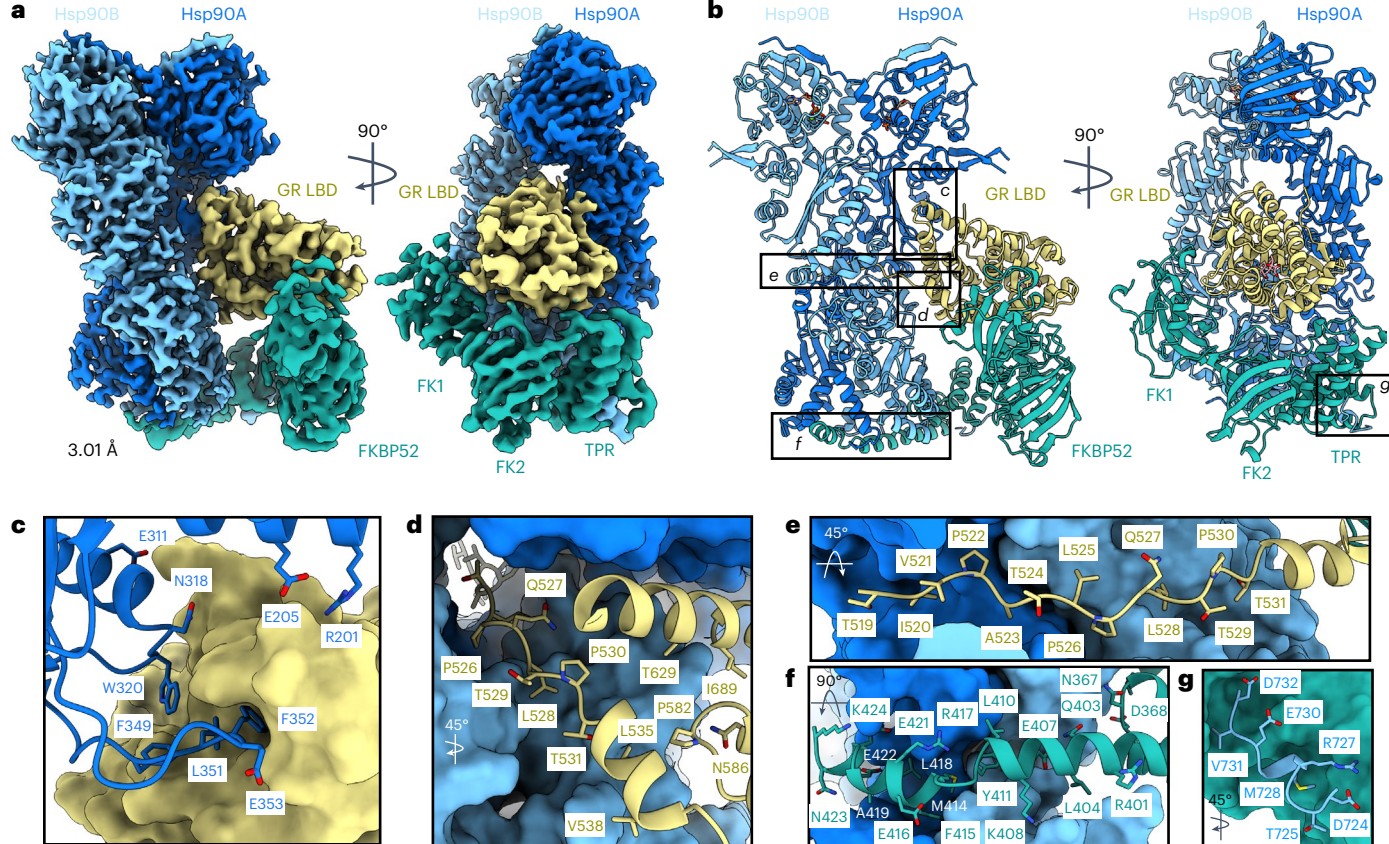

**Fig. 1 | Architecture of the GR:Hsp90:FKBP52 complex. a**, Composite cryo-EM map of the GR:Hsp90:FKBP52 complex. Hsp90A, dark blue; Hsp90B, light blue; GR, yellow; FKBP52, teal. Color scheme is maintained throughout. **b**, Atomic model in cartoon representation with boxes corresponding to the interfaces shown in detail in **c–g**. **c**, Interface 1 of the Hsp90:GR interaction, depicting the Hsp90A Src loop (Hsp90A[345–360]) interacting with the GR hydrophobic patch. GR is in surface representation. **d**, Interface 2 of the Hsp90:GR interaction, depicting GR Helix 1 (GR[532–539]) packing against the entrance to the Hsp90 lumen.

Hsp90A/Hsp90B are in surface representation. **e**, Interface 3 of the Hsp90:GR interaction, depicting GR pre-Helix 1 (GR[519–531]) threading through the Hsp90 lumen. Hsp90A/Hsp90B are in surface representation. **f**, Interface 1 of the Hsp90:FKBP52 interaction, depicting FKBP52 TPR H7e (FKBP52[387–424]) interacting with the Hsp90A/Hsp90B CTD dimer interface. Hsp90A/Hsp90B are in surface representation. **g**, Interface 2 of the Hsp90:FKBP52 interaction, depicting the Hsp90B MEEVD motif (Hsp90B[700–706]) binding in the helical bundle of the FKBP52 TPR domain. FKBP52 is in surface representation.

endocrine-related diseases, including cancers, infertility, anxiety disorders and immune-related diseases[21,33,34]. Despite their importance, the absence of structures of the FKBPs bound to Hsp90:client complexes precludes a mechanistic understanding of how these co-chaperones regulate client function, or how to design selective small molecules[34–39]. In this Article, we present a 3.01 Å cryo-EM structure of the human GR:Hsp90:FKBP52 complex, revealing that FKBP52 directly binds the folded, ligand-bound GR, which we demonstrate is critical for FKBP52-dependent potentiation of GR activity in vivo and in vitro. We also present a 3.23 Å cryo-EM structure of the human GR:Hsp90:FKBP51 complex, which, surprisingly, mimics the GR:Hsp90:FKBP52 structure. We demonstrate that FKBP51 and FKBP52 bind in a mutually exclusive manner, leading to functional antagonism, and both unexpectedly compete with the co-chaperone p23, revealing an additional regulatory step in the GR chaperone cycle.

## Results

### GR:Hsp90:FKBP52 structure determination

The human GR:Hsp90:FKBP52 complex was prepared by in vitro reconstitution of the complete GR chaperone cycle. GR DBD–LBD (residues 418–777 containing the F602S solubilizing mutation) (hereafter, GR) was incubated with Hsp70, Hsp40, Hop, Hsp90, p23 and FKBP52, allowing GR to progress through the chaperone cycle to reach the GR:Hsp90:FKBP52 complex (Extended Data Fig. 1a–c). The complex

was stabilized with sodium molybdate, further purified and lightly crosslinked (Extended Data Fig. 1d,e). A 3.01 Å cryo-EM reconstruction of the GR:Hsp90:FKBP52 complex was obtained (Fig. 1a,b, Table 1 and Extended Data Figs. 1f and 2a,b), revealing a fully closed Hsp90 dimer (Hsp90A and Hsp90B) with a single GR and a single FKBP52, both occupying the same side of Hsp90 (Fig. 1a,b; for a discussion of Hsp90 nucleotide state, see Extended Data Fig. 3a and Methods). Despite using a multi-domain GR construct, only GR_LBD was visible on the map.

### Hsp90 stabilizes GR_LBD in a folded, ligand-bound state

In the GR:Hsp90:FKBP52 complex, GR_LBD is in a fully folded, ligand-bound conformation consistent with the conformation of the LBD in the GR–maturation complex (Extended Data Fig. 3b), but adopting a rotated position (discussed below). The folded GR is stabilized by Hsp90 at three major interfaces (Fig. 1c–e and Extended Data Fig. 3c–f): (1) Hsp90_Src-loop:GR_hydrophobic-patch, (2) Hsp90_CTD:GR_Helix1 and (3) Hsp90_lumen: GR_pre-Helix1 (for labeled structural motifs, see Extended Data Fig. 1c). In the first interface, Hsp90A_Src-loop (Hsp90[345–360]) flips out from the Hsp90_lumen to interact with the previously described GR_hydrophobic-patch[13] (approximately 767 Å² of buried surface area (BSA)) (Fig. 1c and Extended Data Fig. 3c). Along Hsp90A_Src-loop, Hsp90A[F349,L351,F352,E353] contact GR_Helix9/10 and the conserved, solvent-exposed Hsp90A[W320] interacts with GR[F774]. Notably, Hsp90A[W320,F349] also make contact with GR in the GR–loading complex and GR–maturation complex, although at

## Table 1 | Cryo-EM data collection, refinement and validation statistics

| | GR:Hsp90:FKBP52 (EMDB-29068), (PDB 8FFV) | | GR:Hsp90:FKBP51 (EMDB-29069), (PDB 8FFW) | |
|---|---|---|---|---|
| **Data collection and processing** | Dataset I | Dataset II | Dataset III | Dataset IV |
| Magnification | 105,000 | 105,000 | 105,000 | 105,000 |
| Voltage (kV) | 300 | 300 | 300 | 300 |
| Electron exposure ($e^-$ Å$^{-2}$) | 69 | 69 | 69 | 45.8 |
| Defocus range (μm) | 0.8–2.0 | 0.8–2.0 | 0.8–2.0 | 0.8–2.0 |
| Pixel size (Å) | 0.835 | 0.835 | 0.835 | 0.835 |
| Symmetry imposed | C1 | C1 | C1 | C1 |
| Initial particle images (no.) | 4,736387 | 2,783,284 | 3,788,005 | 6,268,573 |
| Consensus map resolution (Å) | 3.01 | | 3.23 | |
| FSC threshold | 0.143 | | 0.143 | |
| Map resolution range (Å) | 2.44–7.73 | | 2.66–9.62 | |
| Final particle images (no.) | 307,109 | | 171,778 | |
| Hsp90 focused map resolution (Å) | 2.96 | | 3.19 | |
| FSC threshold | 0.143 | | 0.143 | |
| Final particle images (no.) | 307,109 | | 171,778 | |
| GR:FKBP focused map resolution (Å) | 3.76 | | 4.14 | |
| FSC threshold | 0.143 | | 0.143 | |
| Final particle images (no.) | 106,318 | | 109,900 | |
| **Refinement** | | | | |
| Initial model (PDB/ AlphaFold code) | 7KRJ, AF-Q02790, 5NJX | | 7KRJ, 7L7I, 5NJX | |
| Model resolution (Å) | 3.13 | | 3.61 | |
| FSC threshold | 0.5 | | 0.5 | |
| Map sharpening *B* factor (Å$^2$) | −30 | | −30 | |
| Model composition | | | | |
| Non-hydrogen atoms | 15,829 | | 15,756 | |
| Protein residue atoms | 15,737 | | 15,664 | |
| Ligand atoms | 92 | | 92 | |
| Mean *B* factors [min–max] (Å$^2$) | | | | |
| Protein | 209.15 [0.03–600.0] | | 244.41 [60.07–600.0] | |
| Ligand | 140.07 [15.18–409.30] | | 222.63 [75.96–528.72] | |
| Root mean square deviations | | | | |
| Bond lengths (Å) | 0.023 | | 0.024 | |
| Bond angles (°) | 1.837 | | 1.936 | |
| Validation | | | | |
| MolProbity score | 0.81 | | 0.84 | |
| Clashscore | 1.07 | | 1.24 | |
| Poor rotamers (%) | 0.00 | | 0.00 | |
| Ramachandran plot | | | | |
| Favored (%) | 98.70 | | 98.55 | |
| Allowed (%) | 1.19 | | 1.40 | |
| Disallowed (%) | 0.10 | | 0.05 | |

quite different locations[12,13]. Additionally, there are multiple hydrogen bonds formed between Hsp90 N-terminal domain/middle domain (Hsp90$_{NTD/MD}$) to GR$_{Helix10}$ and GR$^{K777}$.

Interface 2 is composed of Hsp90$^{Y604}$ packing against GR$_{Helix1}$ (GR$^{532–539}$) and Hsp90$^{Y627}$ sticking into a hydrophobic pocket formed by GR$_{Helix3,4,9}$ (approximately 345 Å$^2$ BSA) (Fig. 1d and Extended Data Fig. 3d,e), which was previously identified in the androgen receptor (AR) as a druggable hydrophobic pocket (BF3)[40]. In interface 3, the unstructured GR$_{pre-Helix1}$ (GR$^{519–531}$) is threaded through the Hsp90$_{lumen}$ (approximately 758 Å$^2$ BSA)(Fig. 1e and Extended Data Fig. 3f). Two hydrophobic residues on GR (GR$^{P522,P526}$) occupy two hydrophobic pockets within the Hsp90$_{lumen}$. The interaction is further stabilized by multiple polar and hydrophobic interactions between GR$_{pre-Helix1}$ and the Hsp90A/Hsp90B amphipathic helical hairpin (Hsp90$^{606–628}$, Hsp90$_{amphi-α}$) and Hsp90A$_{MD}$/Hsp90B$_{MD}$.

### FKBP52 interacts with the closed Hsp90

FKBP52 engages the closed Hsp90 at three major interfaces (Fig. 1f,g and Extended Data Fig. 4a–c): (1) FKBP52$_{TPR/H7e}$:Hsp90A$_{CTD}$/Hsp90B$_{CTD}$, (2) FKBP52$_{TPR}$:Hsp90B$_{MEEVD}$ and (3) FKBP52$_{TPR}$:Hsp90B$_{CTD}$. In interface 1, FKBP52$_{H7e}$ (FKBP52$^{387–424}$) binds in a hydrophobic cleft formed by Hsp90A$_{CTD}$/Hsp90B$_{CTD}$ at the closed dimer interface (approximately 1,109 Å$^2$ BSA) (Fig. 1f, Extended Data Fig. 4a). Compared to the crystal structure, H7e breaks at positions FKBP52$^{411–414}$ to allow hydrophobic residues (FKBP52$^{L410,Y411,M414,F415,L418}$) to flip into the hydrophobic cleft formed by Hsp90$_{CTD}$, consistent with the FKBP51$_{H7e}$:Hsp90 interaction observed by cryo-EM[19]. Mutating the corresponding conserved residues on FKBP51$_{H7e}$ (FKBP51$^{M412,F413}$ corresponding to FKBP52$^{M414,F415}$) abolishes FKBP51:Hsp90 binding, indicating the importance of this binding site[19]. The interface is further stabilized by multiple hydrogen bonds and salt bridges from Hsp90A$_{CTD}$/Hsp90B$_{CTD}$ to H7e flanking the helix break (Extended Data Fig. 4a). Furthermore, a portion of the Hsp90B$_{MEEVD}$ linker (Hsp90B$^{700–706}$) binds along FKBP52$_{H7e}$ (Extended Data Fig. 4a). Helix 7e is found in many TPR-containing co-chaperones[19]; however, our structures, along with others, reveal the Helix 7e can bind Hsp90 in distinct positions due to sequence divergence[41,42].

In interface 2, Hsp90B$_{MEEVD}$ binds in the FKBP52$_{TPR}$ helical bundle (approximately 779 Å$^2$ BSA) (Fig. 1g and Extended Data Fig. 4b), with multiple hydrogen bonds, salt bridges and hydrophobic interactions, analogous to FKBP51:Hsp90$_{MEEVD}$ structures[18,19]. However, the MEEVD peptide binds in an opposite orientation relative to the FKBP52:Hsp90$_{MEEVD}$ crystal structure[17], which may have been incorrectly modeled, as suggested[18,43]. Interface 3 is composed of FKBP52$_{Helix5/6}$ in the TPR domain binding to Hsp90B$_{CTD}$, stabilized by multiple hydrogen bonds (approximately 193 Å$^2$ BSA) (Extended Data Fig. 4c), also observed in the FKBP51:Hsp90 cryo-EM structure[19]. While the interactions between FKBP52$_{TPR/H7e}$:Hsp90 are conserved in the FKBP51:Hsp90 structure, the positions of the FKBP52 FK1 and FK2 domains are notably altered (Extended Data Fig. 4d), owing to the presence of GR, as discussed below.

### FKBP52 directly binds GR, which is critical for GR function

Unexpectedly, FKBP52 directly and extensively interacts with GR, with all three FKBP52 domains wrapping around GR, cradling the folded, ligand-bound receptor near the GR ligand-binding pocket (Fig. 2a). The tertiary structure within each FKBP52 domain closely matches isolated domains from FKBP52 crystal structures; however, the inter-domain angles are significantly different (Extended Data Fig. 4d), probably owing to the extensive interaction with GR. There are three major interfaces between FKBP52 and GR (Fig. 2b–d): (1) FKBP52$_{FK1}$:GR, (2) FKBP52$_{FK2}$:GR and (3) FKBP52$_{FK2/TPR-linker}$:GR$_{Helix12}$.

In interface 1, FKBP52$_{FK1}$ interacts with a large surface on GR, canonically used for GR dimer formation, consisting of the post-Helix 1 strand (Helix 1–3 loop), Helix 5 and β1,2 (approximately 280 Å$^2$ BSA) (Fig. 2b). Three-dimensional variability analysis in CryoSparc revealed that the

interaction between FKBP52$_{FK1}$ and GR is highly dynamic, even as the other FKBP52 domains (FK2 and TPR) remain stably associated with GR (Supplementary Movies 1 and 2). At the FK1:GR interface, GR$^{Y545}$ on the post-Helix 1 strand interacts with a hydrophobic surface formed by the FKBP52$^{81-88}$ loop and forms a hydrogen bond with FKBP52$^{Y113}$. Supporting this interaction, residues of GR$_{post\text{-}Helix1}$ (GR$^{544-546}$) have previously been implicated in FKBP51/52-dependent regulation of GR activity[44,45]. In addition, the FKBP52 proline-rich loop (β4–β5 loop or 80s loop) contacts GR$_{Helix5/β1,2}$. Three-dimensional variability analysis in CryoSparc revealed that the proline-rich loop positioning is flexible, deviating from the position in the crystal structure (Protein Data Bank (PDB) ID: 4LAV) (ref. 46) and adopting different interfaces with GR (Supplementary Movies 3 and 4). In the consensus 3D refinement map, the proline-rich loop adopts a position similar to the crystal structure, and FKBP52$^{A116,S118,P119}$ interact with GR$_{Helix5/β1,2}$. The FKBP52$^{P119L}$ mutation has been shown to reduce GR and AR activation in vivo, while FKBP52$^{A116V}$ has been shown to increase AR activation in vivo[29]. We also demonstrate that the FKBP52$^{S118A}$ mutation significantly reduces FKBP52-dependent GR potentiation in vivo (Fig. 2e), further demonstrating the functional significance of this interaction site. In addition, S118 has been identified as a phosphorylation site on FKBP52, but not FKBP51 (qPTM database[47]) (possibly due to the unique adjacent proline, FKBP52$^{P119}$, which could recruit proline-directed kinases). Phosphorylation at FKBP52$^{S118}$ may help promote the interaction between the proline-rich loop and GR, which could also explain the large effect of the FKBP52$^{S118A}$ mutation in vivo.

While FKBP52$_{FK1}$ is known to have PPIase enzymatic activity, GR is not bound in the PPIase active site and, accordingly, no GR prolines were found to have been isomerized compared to other GR structures (PDB IDs: 1M2Z (ref. 48) and 7KRJ (ref. 13)). Consistent with this, mutation of GR prolines does not disrupt FKBP52-dependent regulation of GR[45]. Additionally, mutations that disrupt PPIase activity do not affect FKBP52-dependent GR potentiation in vivo[29]. Conversely, PPIase inhibitors have been shown to block the FKBP52-dependent potentiation of GR in vivo[23]. This can now be understood, as docking of PPIase inhibitors (FK506 and rapamycin) into the PPIase active site demonstrates that the inhibitors would sterically block the FKBP52$_{FK1}$:GR interface (Extended Data Fig. 4e), as previously hypothesized[23,29].

Interface 2 is composed of the FKBP52 FK2$^{Y161}$ sticking into a shallow hydrophobic pocket formed by GR$_{Helix3}$, GR$_{Helix11-12loop}$ (GR$^{T561,M565,E748}$), and a hydrogen bond between the FKBP52 backbone and GR$^{E748}$ (approximately 125 Å$^2$ BSA) (Fig. 2c). Supporting this interaction, we show that the FKBP52$^{Y161D}$ mutation significantly reduces FKBP52-dependent GR potentiation in vivo, demonstrating the importance of this interaction (Fig. 2e). In interface 3, the solvent exposed, conserved FKBP52$^{W259}$ on the FK2–TPR linker makes electrostatic and hydrophobic interactions with GR$_{Helix12}$ (approximately 235 Å$^2$ BSA) (Fig. 2d), which adopts the canonical agonist-bound position even in the absence of a stabilizing co-activator peptide interaction[48] (Extended Data Fig. 3b). We show that the corresponding FKBP52$^{W259D}$ mutation significantly reduces FKBP52-dependent GR potentiation in vivo, demonstrating the functional importance of this single residue (Fig. 2e). Interestingly, FKBP52$^{W259}$ is also conserved in the FKBP-like co-chaperone XAP2 and a recent structure reveals XAP2 engages with an Hsp90–client using the analogous XAP2$^{W168}$, suggesting this residue is critical more broadly for FKBP co-chaperone:client engagement[42]. At interface 3, FKBP52$^{K254,E257,Y302,Y303}$ make further polar interactions between the FK2–TPR linker and GR$_{Helix12}$ (Fig. 2d). While a significant portion of the GR$_{Helix12}$ co-activator binding site is available in the FKBP52-bound GR, the N-terminus of a co-activator peptide would sterically clash with FKBP52$_{TPR}$ based on the GR:co-activator peptide structure[48] (Extended Data Fig. 5b). Thus, co-activator binding in the nucleus could help release GR from its complex with Hsp90:FKBP52. We also find that the residues at the FKBP52:GR interfaces are conserved across

metazoans (Fig. 2f,g) and have been identified as sites that crosslink to GR in vivo (FKBP52$^{Y159,W257}$) (ref. 49), in agreement with our results that single point mutations at each of the three FKBP52:GR interfaces has a significant effect on GR function in vivo. Based on the observation that FKBP52$_{TPR}$ is sufficient to bind Hsp90 and Hsp90:SHR complexes[19,20], the FKBP52 single point mutants probably do not disrupt FKBP52 binding to the GR:Hsp90 complex, but specifically disrupt the GR:FKBP52 interaction and prevent stabilization of GR$_{LBD}$ by FKBP52.

### FKBP52 advances GR to the next stage of maturation

We previously described another GR:chaperone complex, the GR–maturation complex (GR:Hsp90:p23) (ref. 13), which also contains a closed Hsp90 dimer and a folded, ligand-bound GR (Fig. 3a). However, in the GR:Hsp90:FKBP52 complex, GR is rotated by approximately 45° relative to the GR–maturation complex (Fig. 3a). Hsp90A$_{Src\text{-}loop}$ interacts with GR$_{pre\text{-}Helix1}$ in the maturation complex, but flips out to stabilize the rotated GR position in the GR:Hsp90:FKBP52 complex by interacting with the GR$_{hydrophobic\text{-}patch}$ (Fig. 3b,c). In both complexes, GR$_{pre\text{-}Helix1}$ is threaded through the Hsp90$_{lumen}$; however, in the GR:Hsp90:FKBP52 complex, GR has translocated through the Hsp90$_{lumen}$ by two residues, positioning prolines (GR$^{P522,P526}$) in the hydrophobic pockets of the lumen rather than leucines (GR$^{L525,L528}$) (Fig. 3d). This translocation positions GR$_{LBD}$ further from Hsp90, probably allowing enough space for the observed GR rotation. Despite the translocation and rotation of GR, Hsp90 uses the same surfaces to bind GR (Hsp90B$_{amphi\text{-}α}$, Hsp90A$_{Src\text{-}loop}$ and Hsp90A$^{W320}$); however, the contact surfaces on GR are different. The rotation of GR may facilitate LBD dimerization, which is on pathway to activation. In the GR–maturation complex, dimerization of the LBD clashes with Hsp90$_{CTD}$; however, due to the rotation of the LBD in the GR:Hsp90:FKBP52 complex, LBD dimerization would now be sterically permitted after FKBP52 release (Extended Data Fig. 5c).

### FKBP52 competes with p23 through allostery

Surprisingly, FKBP52 competes with p23 to bind the GR:Hsp90 complex, although there is no direct steric conflict between FKBP52 and p23 binding (Fig. 3a). During 3D classification on the cryo-EM dataset, GR:Hsp90:p23 complexes were observed at low abundance (~74,000 particles); however, the GR:Hsp90:FKBP52 complexes showed no apparent p23 density (Extended Data Fig. 2a,b), despite p23 being present at high concentration in the reconstitution. Furthermore, FKBP52 was only found associated with the rotated GR position, while the GR position in the p23-containing classes was only consistent with the GR–maturation complex. Thus, FKBP52 appears to specifically bind the rotated GR position, which is not compatible with p23 binding. This is consistent with mass spectrometry studies, demonstrating FKBP52 competes with p23 to form a stable GR:Hsp90:FKBP52 complex[50]. In the rotated GR position, Hsp90A$_{Src\text{-}loop}$ flips out of the Hsp90$_{lumen}$ to bind the GR$_{hydrophobic\text{-}patch}$, which was previously engaged by the p23$_{tail\text{-}helix}$ (Fig. 3a–c). Thus, rotation of GR dictates the accessibility of the GR$_{hydrophobic\text{-}patch}$ to either Hsp90 or p23. FKBP52 stabilizes the rotated position of GR and therefore favors GR binding to Hsp90A$_{Src\text{-}loop}$ over p23.

### FKBP52 potentiates GR ligand binding in vitro

To quantitatively assess the functional significance of FKBP52 on GR activation, we added FKBP52 to the in vitro reconstituted GR chaperone cycle, using the GR DBD–LBD construct (residues 418–777 containing the F602S solubilizing mutation) and monitored GR ligand binding, as previously described[11,13]. Addition of FKBP52 to the GR chaperone cycle resulted in the enhancement of GR ligand binding above the already enhanced GR + chaperones control reaction at equilibrium (Fig. 3e), strongly suggesting FKBP52 potentiates the GR ligand binding affinity beyond the minimal chaperone mixture, consistent with reports in vivo[23]. We hypothesized that FKBP52 functions in a similar manner to the p23$_{tail\text{-}helix}$ in stabilizing the ligand-bound GR. As previously

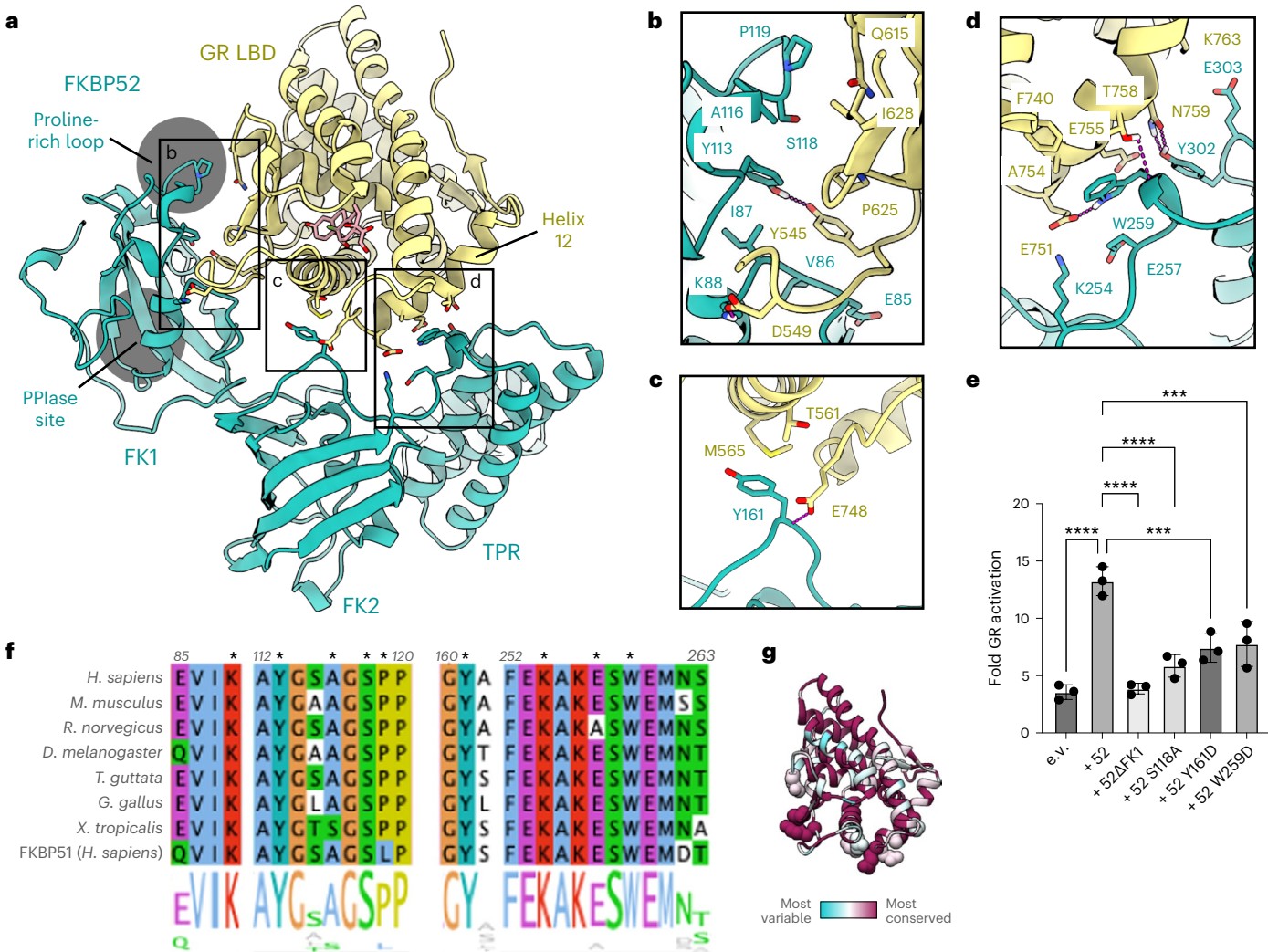

**Fig. 2 | The GR:FKBP52 interaction and functional significance. a**, Atomic model depicting the three interfaces between GR (yellow) and FKBP52 (teal) in the GR:Hsp90:FKBP52 complex. The FKBP52 proline-rich loop and PPIase catalytic site are highlighted in gray. Dexamethasone is colored in pink. **b**, Interface 1 between GR (yellow) and the FKBP52 FK1 domain (teal), showing interacting side chains and hydrogen bonds (dashed pink lines). **c**, Interface 2 between GR (yellow) and the FKBP52 FK2 domain (teal), showing interacting side chains and hydrogen bonds (dashed pink lines). **d**, Interface 3 between GR (yellow) and the FKBP52 FK2–TPR linker (teal), showing interacting side chains and hydrogen bonds (dashed pink lines). **e**, GR activation assay in wild-type yeast strain JJ762 expressing FKBP52 ('52') or FKBP52 mutants. The fold increase in GR activities compared to the empty vector (e.v.) control are shown (mean ± s.d.).

$n = 3$ biologically independent samples per condition. Significance was evaluated using a one-way analysis of variance ($F_{(5,12)} = 26.10$; $P < 0.0001$) with post-hoc Dunnett's multiple comparisons test (n.s. $P > 0.05$; *$P \le 0.05$; **$P \le 0.01$; ***$P \le 0.001$). $P$ values: $P$(e.v. versus 52) <0.0001, $P$(52 versus 52ΔFK1) <0.0001, $P$(52 versus 52 S118A) <0.0001, $P$(52 versus 52 Y161D) 0.0003, $P$(52 versus 52 W259D) 0.0005. **f**, Sequence alignment of eukaryotic FKBP52 showing conserved residues involved in the GR:FKBP52 interaction (denoted by a black asterisk). The bottom aligned sequence is human FKBP51. The alignment is colored according to the ClustalW convention. **g**, GR protein sequence conservation mapped onto the GR atomic model from the GR:Hsp90:FKBP52 complex. Residue conservation is depicted from most variable (cyan) to most conserved residues (maroon). GR residues that interact with FKBP52 are shown as spheres.

described, removal of the p23$_{tail-helix}$ (p23Δhelix) resulted in a decrease in GR ligand binding activity in the GR chaperone system[13]; however, addition of FKBP52 to the reaction fully rescued GR ligand binding in the p23Δhelix background (Fig. 3e). Thus, FKBP52 can functionally replace the p23$_{tail-helix}$, probably by also directly stabilizing the ligand-bound GR. Given that FKBP52 and the p23$_{tail-helix}$ bind different locations on GR, the mechanisms of stabilization may be unique. Additionally, in the p23Δhelix background, FKBP52 potentiated ligand binding to a greater extent than in the wild-type p23 background. We hypothesize that removing p23$_{tail-helix}$ alleviates the competition between p23 and FKBP52, allowing p23 to remain bound to the GR:Hsp90:FKBP52 complex. Given that p23 is known to stabilize the closed Hsp90 conformation[13,51], the enhanced ligand binding in the p23Δhelix background may be due to stabilization of closed

Hsp90 by p23Δhelix. Interestingly, FKBP52 also affected GR ligand binding independent of Hsp90, with addition of FKBP52 to GR resulting in enhanced ligand binding, probably due to an Hsp90-independent chaperoning effect[15,52] (Extended Data Fig. 5d).

## Upon Hsp90 closure, FKBP52 can functionally replace p23

Given that FKBP52 can functionally replace p23$_{tail-helix}$, we wondered whether FKBP52 could also functionally replace p23 altogether. p23 is known to stabilize Hsp90$_{NTD}$ closure through the globular p23 domain[13,51] in addition to stabilizing the ligand-bound GR through the p23$_{tail-helix}$[13]. Omitting p23 from the GR chaperone cycle drastically reduces GR ligand binding, as previously described[11,13]. The addition of FKBP52 in place of p23 results in a small increase in ligand binding but does not fully rescue ligand binding activity (Fig. 3f). We reasoned this

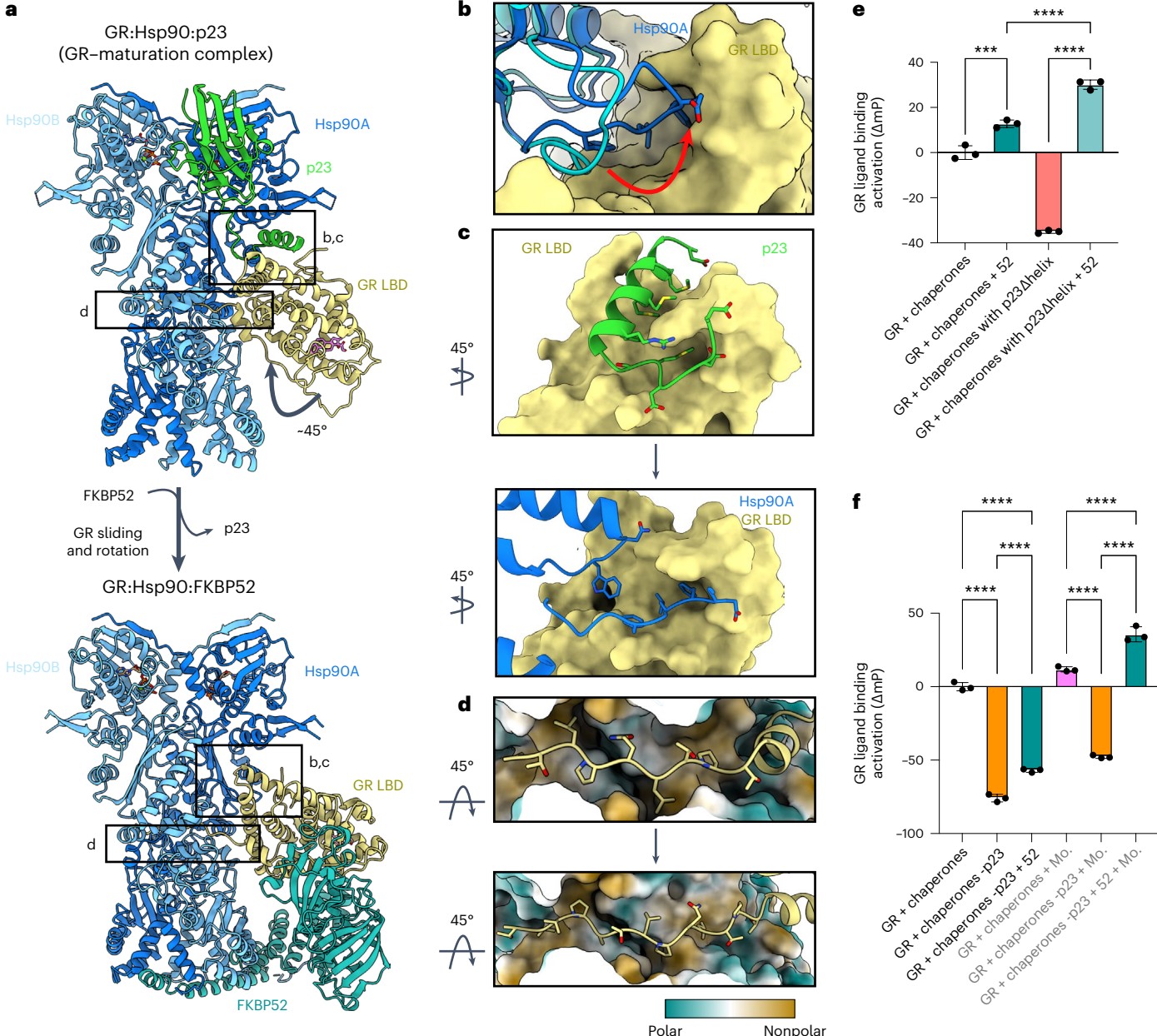

**Fig. 3 | FKBP52 competes with p23 to bind GR:Hsp90. a,** Atomic model of the GR–maturation complex (top) and the GR:Hsp90:FKBP52 complex (bottom) with boxes corresponding to the interfaces shown in detail in **b–d.** Hsp90A, dark blue; Hsp90B, light blue; GR, yellow; p23, green; FKBP52, teal. **b,** Position of the Hsp90A Src loop in the GR–maturation complex (Hsp90A, cyan) versus the GR:Hsp90:FKBP52 complex (Hsp90A, dark blue). GR (yellow, surface representation). Hsp90A Src loop residues interacting with the GR hydrophobic patch are shown. **c,** Interface between the p23 tail-helix (green) and the GR hydrophobic patch (yellow, surface representation) in the GR–maturation complex (top). The p23 tail-helix is replaced by the Hsp90A Src loop (dark blue) in the GR:Hsp90:FKBP52 complex (bottom), which flips up to interact with the GR hydrophobic patch (yellow, surface representation). Interacting side chains are shown. **d,** Interaction between GR pre-Helix 1 (GR[523–531]) in the Hsp90 lumen in the GR–maturation complex (top) versus the GR:Hsp90:FKBP52 complex (bottom). Hsp90A/Hsp90B are in surface representation colored by hydrophobicity. GR pre-Helix 1 translocates through the Hsp90 lumen by two residues in the transition from the GR–maturation complex to the GR:Hsp90:FKBP52 complex.

**e,** Equilibrium binding of 10 nM F-dex to 100 nM GR DBD–LBD with chaperones and 15 µM FKBP52 ('52') (mean ± s.d.). $n = 3$ biologically independent samples per condition. 'Chaperones': 15 µM Hsp70, Hsp90, Hop, and p23 or p23Δhelix; 2 µM Ydj1 and Bag-1. Significance was evaluated using a one-way analysis of variance (ANOVA) ($F_{(3,8)} = 541.2$; $P < 0.0001$) with post-hoc Šídák's test (n.s. $P > 0.05$; *$P ≤ 0.05$; **$P ≤ 0.01$; ***$P ≤ 0.001$; ****$P ≤ 0.0001$). $P$ values: $P$(chaperones versus chaperones + 52) 0.0002, $P$(chaperones + 52 versus chaperones with p23Δhelix + 52) <0.0001, $P$(chaperones with p23Δhelix versus chaperones with p23Δhelix + 52) <0.0001. **f,** Equilibrium binding of 10 nM F-dex to 100 nM GR DBD–LBD with chaperones, 15 µM FKBP52 ('52') and 20 mM sodium molybdate ('Mo.') (mean ± s.d.). $n = 3$ biologically independent samples per condition. 'Chaperones': 15 µM Hsp70, Hsp90, Hop and p23; 2 µM Ydj1 and Bag-1. '-p23' indicates p23 was left out of the chaperone mixture. Significance was evaluated using a one-way ANOVA ($F_{(5,12)} = 761.5$; $P < 0.0001$) with post-hoc Šídák's test (n.s. $P > 0.05$; *$P ≤ 0.05$; **$P ≤ 0.01$; ***$P ≤ 0.001$; ****$P ≤ 0.0001$). $P$ value <0.0001 for each comparison.

could be due to the inability of FKBP52 to sufficiently stabilize Hsp90 closure, as previously suggested[53]. Therefore, we added molybdate to these reactions, which stabilizes Hsp90$_{NTD}$ closure by acting as a γ-phosphate analog in the Hsp90$_{NTD}$ ATP-binding site[13,54]. Addition of molybdate to the reaction lacking p23 resulted in a small increase in GR ligand binding but did not fully rescue ligand binding activity. However, addition of molybdate to the reactions containing FKBP52 without p23 resulted in a full reactivation of ligand binding and even potentiated ligand binding over the control GR + chaperones reaction (Fig. 3f), much like with p23Δhelix. Thus, FKBP52 is able to functionally replace p23 if Hsp90$_{NTD}$ closure is stabilized. Taken together, these results suggest FKBP52 can stabilize the ligand-bound GR, like p23, but cannot stabilize the closed Hsp90$_{NTD}$ conformation, which requires p23.

### GR:Hsp90:FKBP51 structure determination

In vivo, the interplay between FKBP52 and the highly similar FKBP51 have profound implications for GR activity. FKBP51 is functionally antagonistic to FKBP52-dependent potentiation of GR in vivo; thus, the relative ratios of FKBP51 and FKBP52 dictate GR activity levels[23,28,55]. To understand how FKBP51 antagonizes FKBP52, we prepared the human GR:Hsp90:FKBP51 complex by in vitro reconstitution of the GR chaperone cycle with FKBP51 (Extended Data Fig. 1d–f). We obtained a 3.23 Å cryo-EM reconstruction of GR:Hsp90:FKBP51 (Fig. 4a,b, Table 1 and Extended Data Fig. 6a,b). Contrary to our expectations, the FKBP51-containing structure appears nearly identical to the FKBP52-containing structure. The GR:Hsp90:FKBP51 structure reveals a fully closed Hsp90 dimer complexed with a single GR and a single FKBP51, which occupy the same side of Hsp90 (Fig. 4a,b; for a discussion of Hsp90 nucleotide, see Extended Data Fig. 7a and Methods). As with the FKBP52 dataset, GR:Hsp90:p23 complexes were also observed during 3D classification and the GR:Hsp90:FKBP51 complexes showed no apparent p23 density (Extended Data Fig. 6a,b). The FKBP51:Hsp90 interactions are analogous to the FKBP52:Hsp90 interactions, including Hsp90B$_{MEEVD}$:FKBP$_{TPR}$ and Hsp90$_{CTD}$:FKBP$_{H7e}$, also seen in the Hsp90:FKBP51:p23 structure[19], but distinct from a previous nuclear magnetic resonance model[56] (Fig. 4b and Extended Data Fig. 7b–d). The GR:Hsp90 interfaces are nearly identical when comparing the FKBP51 and FKBP52-containing complexes, including Hsp90$_{Src-loop}$:GR$_{hydrophobic-patch}$ and Hsp90$_{lumen}$:GR$_{pre-Helix1}$ (Fig. 4b and Extended Data Fig. 7e–g).

FKBP51 also directly binds GR in an analogous manner to FKBP52 (Fig. 4c–e). FKBP51 binds the folded, ligand-bound, rotated GR using the same three major interfaces (1) FKBP51$_{FK1}$:GR, (2) FKBP51$_{FK2}$:GR$_{Helix3}$ and (3) FKBP51$_{FK2/TPR-linker}$:GR$_{Helix12}$. The GR:FKBP52 interaction residues are largely conserved for GR:FKBP51 (Fig. 2f). As with the FKBP52-containing structure, no GR prolines appear to be isomerized and the PPIase inhibitors rapamycin and FK506 sterically clash with the GR backbone. Interestingly, the small FKBP51-specific inhibitor, SAFit2 (PDB ID: 6TXX)[35,57], does not clash with the GR backbone and may be accommodated with only side chain rotations, consistent with in vivo data[49] (Extended Data Fig. 7h). Furthermore, the FKBP51 FK1 domain and FK1 proline-rich loop are highly dynamic, as revealed by CryoSparc 3D variability analysis, analogous to the FKBP52-containing structure (Supplementary Movies 5 and 6). However, in the GR:Hsp90:FKBP51 complex, the FK1 domain contacts GR at a different angle relative to the GR:Hsp90:FKBP52 complex. Thus, the FK1:GR interface is distinct between the two complexes, specifically at the functionally important, but divergent, residue 119 in the proline-rich loop (FKBP51$^{L119}$ and FKBP52$^{P119}$) (Figs. 2b and 4c)[29], which we investigated further below.

### FKBP51/52 functional difference is dependent on residue 119

To quantitatively assess the functional effect of FKBP51 on GR in vitro, we added FKBP51 to the GR chaperone cycle and measured ligand

binding activity. FKBP51 had no effect on the GR equilibrium value (Extended Data Fig. 8a), unlike FKBP52 which potentiated GR ligand binding. However, we found FKBP51 can functionally replace p23$_{tail-helix}$ (or p23, if molybdate is added), just as we observed with FKBP52 (Fig. 4f and Extended Data Fig. 8b). However, FKBP51 does not potentiate GR ligand binding in any of these conditions, unlike FKBP52, recapitulating in vivo findings[23,29].

The residues responsible for the functional difference between FKBP51 and FKBP52 in vivo are in the FK1 domain proline-rich loop, specifically the divergent residue 119 (FKBP51$^{L119}$ and FKBP52$^{P119}$)[29]. To assess whether this residue is responsible for the functional difference between FKBP51 and FKBP52 in vitro, we swapped residue 119 (FKBP51$^{L119P}$ and FKBP52$^{P119L}$) and added these mutants to the in vitro reconstituted GR chaperone cycle. We then measured ligand binding activity in the p23Δhelix background, where the largest potentiation due to FKBP52 was observed. Surprisingly, the residue 119 swapped mutants almost fully reversed the effects of FKBP51 and FKBP52 on GR: FKBP51$^{L119P}$ potentiated GR ligand binding over the GR + chaperones control reaction, while FKBP52$^{P119L}$ showed significantly less potentiation of ligand binding compared to wild-type FKBP52 (Fig. 4f). These results are consistent with the effects of the FKBP51/52 residue 119 swapped mutants in vivo[29]. Thus, residue 119 on the proline-rich loop provides a critical functional difference between FKBP51 and FKBP52 on GR activity in vitro and in vivo, probably driven via the differential positioning of the loop seen in our structures.

## Discussion

We present two cryo-EM structures, demonstrating how the co-chaperones FKBP51 and FKBP52 bind to an Hsp90–client complex. The 3.01 Å human GR:Hsp90:FKBP52 structure reveals that FKBP52 directly and extensively binds the client using three distinct interfaces that stabilize the folded, ligand-bound conformation of GR. We show that FKBP52 enhances GR ligand binding in vitro and that each of the three observed GR:FKBP52 interfaces is critical for FKBP52-dependent potentiation in vivo. We also provide a 3.23 Å human GR:Hsp90:FKBP51 structure, unexpectedly demonstrating FKBP51 binds to the GR:Hsp90 complex similarly to FKBP52, providing a molecular explanation for the functional antagonism between FKBP51 and FKBP52. Our structures contribute to an emerging theme in which Hsp90 co-chaperones bind to distinct Hsp90 conformations, while simultaneously stabilizing specific client conformations to regulate client activity[12,13,41,42,54].

A recent study[49] using in vivo chemical crosslinking validates our structures remarkably well, recapitulating all three major GR:FKBP51/52 contacts as well as the FKBP-mediated rotated GR position. Given that the in vivo crosslinking between GR and FKBP51/52 was performed in the absence of ligand, together our findings demonstrate that FKBP51 and FKBP52 bind apo GR$_{LBD}$ in a similar, if not identical manner to the ligand-bound GR$_{LBD}$ observed here in our structures. In vivo, ligand addition dissociates GR:Hsp90:FKBP51/52 complexes[49] probably due to the rapid ligand-dependent nuclear translocation of GR[22,58]. While our high-resolution reconstructions unambiguously contain ligand, apo GR:Hsp90:FKBP51/52 complexes probably also exist in our dataset, but are less well ordered (consistent with the GR–maturation complex[13]). In addition, that study provided further in vivo validation of our structural models by demonstrating FK506, but not SAFit2, inhibits FKBP51-dependent regulation of GR in vivo (Extended Data Figs. 4e and 7h) and that the FKBP51/52 FK1 domain is dynamically associated with GR (Supplementary Movies 1–6). Altogether, these studies complement each other extraordinarily well, demonstrating direct association of FKBP51 and FKBP52 with GR$_{LBD}$ in vivo and in vitro at single-residue resolution.

Unexpectedly, our structures also demonstrate that FKBP51 and FKBP52 compete with p23 to bind the GR:Hsp90 complex through an allosteric mechanism. Previous reports showed FKBP51

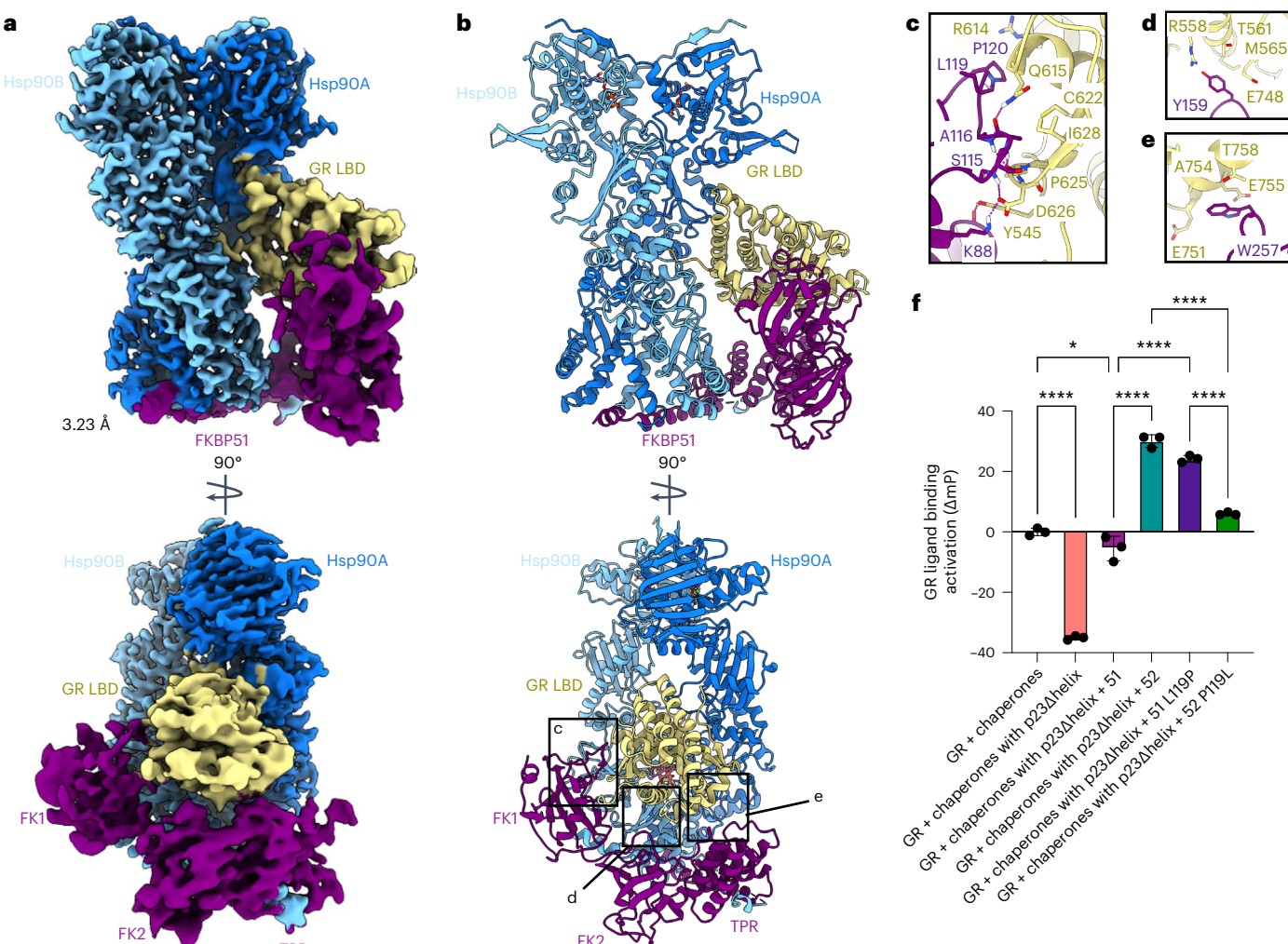

**Fig. 4 | Architecture of the GR:Hsp90:FKBP51 complex. a**, Composite cryo-EM map of the GR:Hsp90:FKBP51 complex. Hsp90A, dark blue; Hsp90B, light blue; GR, yellow; FKBP51, purple. Color scheme is maintained throughout. **b**, Atomic model in cartoon representation with boxes corresponding to the interfaces shown in detail in **c**–**e**. **c**, Interface 1 between GR (yellow) and the FKBP51 FK1 domain (purple), showing interacting side chains and hydrogen bonds (dashed pink lines). **d**, Interface 2 between GR (yellow) and the FKBP51 FK2 domain (purple), showing interacting side chains and hydrogen bonds (dashed pink lines). **e**, Interface 3 between GR (yellow) and the FKBP51 FK2–TPR linker (yellow), showing interacting side chains and hydrogen bonds (dashed pink lines). **f**, Equilibrium binding of 10 nM F-dex to 100 nM GR DBD–LBD with chaperones,

15 μM FKBP51 ('51'), 15 μM FKBP52 ('52') or mutants (mean ± s.d.). $n = 3$ biologically independent samples per condition. 'Chaperones': 15 μM Hsp70, Hsp90, Hop, and p23 or p23Δhelix; 2 μM Ydj1 and Bag-1. Significance was evaluated using a one-way analysis of variance ($F_{(5,12)} = 404.1$; $P < 0.0001$) with post-hoc Šídák's test (n.s. $P > 0.05$; *$P ≤ 0.05$; **$P ≤ 0.01$; ***$P ≤ 0.001$; ****$P ≤ 0.0001$). $P$ values: $P$(chaperones versus chaperones with p23Δhelix) <0.0001, $P$(chaperones versus chaperones with p23Δhelix + 51) 0.0343, $P$(chaperones with p23Δhelix + 51 versus chaperones with p23Δhelix + 51 L119P) <0.0001, $P$(chaperones with p23Δhelix + 51 versus chaperones with p23Δhelix + 52) <0.0001, $P$(chaperones with p23Δhelix + 52 versus chaperones with p23Δhelix + 52 P119L) <0.0001, $P$(chaperones with p23Δhelix + 51 P119L versus chaperones with p23Δhelix + 52 P119L) <0.0001.

and p23 could simultaneously bind the closed Hsp90 in the absence of client[19]. We demonstrate that the position of the client can dictate which co-chaperone is bound, with the FKBPs and p23 binding to distinct GR surfaces accessible in distinct GR orientations. FKBP51 and FKBP52 stabilize a rotated position of GR relative to the GR–maturation complex, which may facilitate post-translational modifications, interactor binding or GR dimerization, as suggested previously[59], raising the possibility that the FKBPs promote the next step in maturation. Although the FKBPs directly contact GR, they do not appear to isomerize GR prolines or engage GR_NLS1 (nuclear localization signal 1) (GR[467–505]) (ref. 60) to regulate GR activity, as previously hypothesized[8,61–63].

While FKBP51 binds GR:Hsp90 similarly to FKBP52, we find that, unlike FKBP52, FKBP51 does not enhance GR ligand binding in vitro, consistent with in vivo studies[23,27,29]. Importantly, we find proline 119 on FKBP52 is critical for enhancement of ligand binding in vitro, also

consistent with in vivo studies[29]. Proline 119 on FKBP52 was found to decrease dynamics of the proline-rich loop (80s loop, β4–β5 loop) by nuclear magnetic resonance, relative to leucine 119 on FKBP51 (ref. 64). Three-dimensional variability analysis of our structures demonstrates the FKBP51/52 proline-rich loop dynamically interacts with GR and differences in dynamics may dictate the specificity and/or stability of the interaction, leading to distinct regulation of GR activity by the FKBPs.

Based on our structures of the GR:Hsp90:FKBP51 and GR:Hsp90:FKBP52 complexes, we propose additional steps in the GR chaperone cycle accounting for FKBP51/52 incorporation and subsequent regulation of GR activity (Fig. 5). In the cytosol, GR cycles between Hsp70 and Hsp90, which locally unfold and refold GR to directly control ligand binding[11–13]. Once the folded GR reaches the GR–maturation complex (GR:Hsp90:p23), either FKBP51 or FKBP52 binds the complex and competes with p23 to advance GR to the next

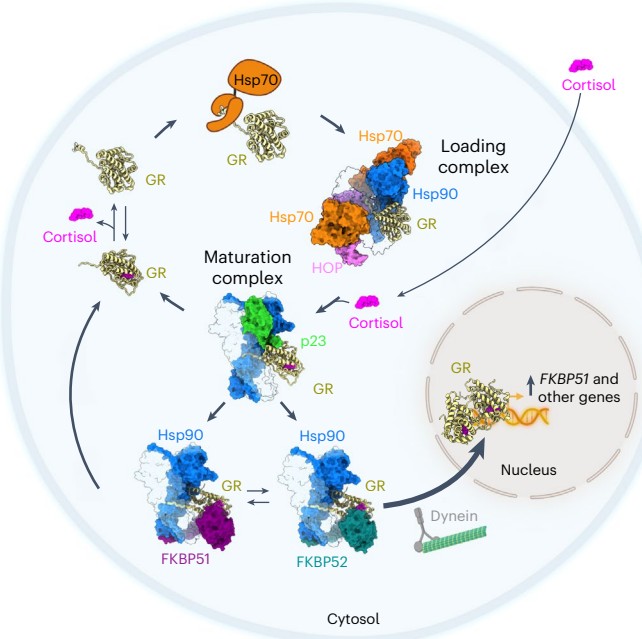

**Fig. 5 | Mechanism of GR regulation by FKBP51 and FKBP52 during the GR chaperone cycle.** Schematic depicting how the FKBP co-chaperones integrate with the GR chaperone cycle and how this cycle may take place within a cellular context. Starting on the top left, GR (yellow, cartoon representation) is in dynamic equilibrium between cortisol-bound and unbound (apo) states. Hsp70 (orange) binds GR and locally unfolds GR to inhibit cortisol binding, stabilizing GR in a partially unfolded, apo state. Hsp70 transfers the partially unfolded GR to Hsp90 (light and dark blue):Hop (pink) to form the GR–loading complex[12], in which GR is stabilized in a partially unfolded, apo state. Cortisol (pink), which enters the cell through diffusion, binds to GR during the transition from the GR–loading complex to the GR–maturation complex when Hsp90 refolds the GR to a native conformation[13]. In the GR–maturation complex, the cortisol-bound, folded GR is stabilized by Hsp90 and p23 (green) and is protected from Hsp70 re-binding. Depending on the relative concentrations of the FKBPs, either FKBP51 (purple) or FKBP52 (teal) can bind the GR:Hsp90:p23 complex, competing with p23, and stabilizing the rotated position of GR. FKBP51 sequesters GR:Hsp90 in the cytosol until ATP hydrolysis on Hsp90 allows release of GR back to the chaperone cycle. In contrast, FKBP52 promotes rapid nuclear translocation of GR:Hsp90 (refs. 22,24,25,65). Once in the nucleus, the cortisol-bound GR can dimerize, nucleate the assembly of transcriptional regulatory complexes, and regulate transcription, including activating expression of FKBP51, leading to a negative feedback loop that regulates GR activity in the cell[27,66–70].

stage of maturation. Given that the folded GR is strongly stabilized and tightly associated with Hsp90 and the FKBPs, we suggest that it is unlikely that ligand binding/unbinding happens in the context of FKBP-bound complexes. Instead, we propose that ligand binds before the formation of either the GR–maturation complex or the GR:Hsp90:FKBP complexes, and that unbinding mostly occurs by recycling GR back to Hsp70, as previously described[11–13].

The functional outcome for GR is dictated by which FKBP binds. FKBP52 stabilizes ligand-bound GR, resulting in enhanced ligand affinity, and facilitates rapid GR nuclear translocation on dynein[22,24,25,65], allowing GR to proceed with dimerization and transcription activation. In contrast, FKBP51 keeps GR sequestered in the cytosol, enabling GR to recycle back into the chaperone cycle, inhibiting GR translocation and transcription activation. Interestingly, the expression of FKBP51, but not FKBP52, is upregulated by GR (also PR and AR), leading to a short negative feedback loop, which may help dampen chronic GR activation and signaling[27,66–70]. Thus, the relative concentrations of FKBP51 and FKBP52 in the cell dictate the level of GR activity in vivo[23,28,55].

Beyond GR, FKBP51/52 are known to regulate the entire SHR class. Given the sequence and structural conservation of SHR$_{LBD}$ at the FKBP binding sites, we propose FKBP51/52 engage with all SHRs in a similar manner (Extended Data Fig. 9a,b). Thus, FKBP51/52 can fine-tune the activity of these critical and clinically important signaling molecules and allow for crosstalk between the hormone signaling pathways. Altogether, we demonstrate how Hsp90 provides a platform for the FKBP co-chaperones to engage Hsp90 clients after folding and promote the next step of client maturation, providing a critical layer of functional regulation.

## Online content

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

## Methods

### Data analysis and figure preparation

Figures were created using UCSF Chimera v.1.14 (ref. [71]), UCSF ChimeraX v1.0.0 (ref. [72]) and BioRender.com. Data from GR activation assays and GR ligand binding assays were analyzed using Prism v.9.4.0 for Mac (GraphPad Software, www.graphpad.com).

### Protein expression and purification

Human Hsp90α, Hsp70 (gene *Hsp70A1A*), Hop, p23, p23Δhelix (1–112), Bag-1 isoform 4 (116–345) and yeast Ydj1 (Hsp40) were expressed as previously described[13]. FKBP51 and FKBP52 were expressed in the pET151 bacterial expression plasmid with a cleavable N-terminal, 6x-His tag and purified in an analogous manner.

### GR DBD–LBD expression and purification

The human GR DBD–LBD construct contains the GR DNA binding domain (DBD), hinge, and ligand binding domain (LBD) (418–777) with solubilizing mutation F602S. The construct was codon optimized and expressed in the pMAL-c3X derivative with an N-terminal cleavable 6x-His–MBP tag. For datasets I, II and III, GR DBD–LBD was expressed and purified with ligand as follows. GR DBD–LBD were expressed in *Escherichia coli* BL21 star (DE3) strain. Cells were grown in either Luria broth or Terrific broth at 37 °C until $OD_{600}$ reached 0.6, and then 100 μM dexamethasone and 50 μM $ZnCl_2$ were added. Cells were induced with 1 mM isopropyl β-D-1-thiogalactopyranoside at $OD_{600}$ 0.8. Cells were grown overnight (~16–18 h) at 16 °C. Cells were collected and lysed in 50 mM Tris pH 8, 300 mM KCl, 50 μM dexamethasone, 5 mM imidazole pH 8, 10% glycerol, 2 mM dithiothreitol (DTT) and 0.2 mM phenylmethylsulfonyl fluoride. Lysate was centrifuged and the soluble fraction was affinity purified by gravity column with Ni-NTA affinity resin (QIAGEN). During Ni-NTA affinity purification, the resin was washed with a buffer containing 30 mM Tris pH 8, 500 mM KCl, 50 μM dexamethasone, 10% glycerol, 2 mM DTT, 2 mM ATP, 5 mM $MgCl_2$ and 0.1% Tween20. Then the resin was washed with a buffer containing 30 mM Tris pH 8, 500 mM KCl, 50 μM dexamethasone, 10% glycerol, 2 mM DTT and 5 mM ethylenediaminetetraacetic acid. The protein was eluted with 30 mM Tris pH 8, 150 mM KCl, 50 μM dexamethasone, 300 mM imidazole pH 8, 10% glycerol and 3 mM DTT. Protein was then purified by size exclusion in 30 mM HEPES pH 7.5, 150 mM KCl, 50 μM dexamethasone, 10% glycerol and 4 mM DTT using a HiLoad 16/60 Superdex 200 (GE Healthcare). Protein was purified a second time by size exclusion using a HiLoad 16/60 Superdex 200 (GE Healthcare) to further remove degradation products. Protein was concentrated, flash frozen and stored at −80 °C.

For dataset IV and GR ligand binding assays, apo GR DBD–LBD was expressed and purified as in a similar manner as described above; however, after Ni-NTA purification, the protein was dialyzed overnight in a buffer containing 30 mM Tris pH 7.5, 150 mM KCl, 10% glycerol and 2 mM DTT. Protein was then purified by hydrophobic interaction chromatography on a HiScreen Butyl-S FF (4.7 mL) column (Cytiva Life Sciences) to remove degradation products. First, solid KCl was slowly added to the protein solution at 4 °C to a final concentration of 2 M KCl. Then the protein was injected onto the hydrophobic interaction chromatography column and eluted over a gradient of 2 M–0 M KCl over 10 column volumes in a buffer containing 30 mM Tris pH 7.5, 10% glycerol and 2 mM DTT. Then the protein was further purified by size exclusion in 30 mM HEPES pH 8, 150 mM KCl, 10% glycerol and 0.5 mM tris(2-carboxyethyl) phosphine using a HiLoad 16/60 Superdex 200 (GE Healthcare). Protein was then dialyzed for 3 days, with fresh buffer each day, in a buffer containing 30 mM HEPES pH 8, 150 mM KCl, 10% glycerol and 2 mM DTT. Protein was concentrated, flash frozen and stored at −80 °C.

### GR:Hsp90:FKBP complex sample preparation

The GR chaperone cycle was reconstituted in vitro with purified components as previously described[11] with addition of 15 μM FKBP51 or FKBP52. After 60 min at room temperature, another 15 μM FKBP51 or FKBP52 was added, along with 15 μM Bag-1 and 20 mM sodium molybdate. The reaction was incubated at room temperature for another 30 min and then a pulldown on MBP–GR DBD–LBD was performed as previously described. The elution was analyzed by sodium dodecyl sulfate–polyacrylamide gel electrophoresis (SDS–PAGE) (4–12% acrylamide gel) (Extended Data Fig. 1d). The elution was concentrated and purified by size exclusion using a Superdex 200 Increase 3.2/300 (Cytiva Life Sciences), and fractions were analyzed by SDS–PAGE (4–12% acrylamide gel) (Extended Data Fig. 1e). Fractions containing the full complex were concentrated to ~2 μM and crosslinked with 0.02% glutaraldehyde for 20 min at room temperature (Extended Data Fig. 1d). Then 2.5 μl of sample was applied to glow-discharged QUANTIFOIL R1.2/1.3, 400-mesh, copper holey carbon grid (Quantifoil Micro Tools GmbH) and plunge-frozen in liquid ethane using a Vitrobot Mark IV (FEI) with a blotting time of 12–16 s, blotting force 3, at 10 °C, and with 100% humidity.

### Cryo-EM data acquisition

All data were acquired using SerialEM software v.4.0 (ref. [73]) and collected on an FEI Titan Krios electron microscope (Thermo Fisher Scientific) using a K3 direct electron camera (Gatan) and equipped with a Bioquantum energy filter (Gatan) set to a slit width of 20 eV (example micrographs in Extended Data Fig. 1f). For additional parameters, see Table 1. All datasets were acquired using fringe-free imaging (FFI) and multi-hole targeting using image shift in which three micrographs were collected per hole. Two small datasets on the GR:Hsp90:FKBP52 and GR:Hsp90:FKBP51 complex were collected before the larger datasets and used for initial model generation (described below).

### Cryo-EM data processing

The smaller GR:Hsp90:FKBP51 and GR:Hsp90:FKBP52 datasets consisted of 1,181 and 2,022 dose-fractionated image stacks, respectively, which were motion corrected using UCSF MotionCor2 (ref. [74]) and analyzed with RELION v.3.0.8 (ref. [75]). Motion-corrected images were used for contrast transfer function (CTF) estimation using CTFFIND v.4.1 (ref. [76]), and Laplacian-of-Gaussian particle picking was done in RELION. Multiple rounds of 3D classification with symmetry C1 were performed with the GR–maturation complex (PDB ID: 7KRJ) (ref. [13]) as a low-pass-filtered (20 Å) initial model until medium-resolution (~8 Å) GR:Hsp90:FKBP51 and GR:Hsp90:FKBP52 reconstructions were obtained. These reconstructions were used as an initial references for the larger GR:Hsp90:FKBP51 and GR:Hsp90:FKBP52 datasets, respectively.

Datasets I–IV were motion corrected using UCSF MotionCor2 and analyzed with RELION v.3.1.0. Motion-corrected images with dose weighting were used for CTF estimation using CTFFIND v.4.1 and reference-based picking was done in RELION using the corresponding references from the smaller datasets described above. The processing scheme for datasets I–IV are depicted in Extended Data Figs. 2a and 6a. After initial rounds of 3D classification with symmetry C1, the GR:Hsp90:FKBP52 datasets (I and II) were combined and GR:Hsp90:FKBP51 datasets (III and IV) were combined.

For the GR:Hsp90:FKBP52 combined dataset, a particle stack of ~496,000 particles was obtained representing a GR:Hsp90:FKBP52 reconstruction at nominal resolution 3.82 Å. This stack was then subjected to 3D classification without alignment and subsequent 3D refinement on the best classes (~307,000 particles), which yielded the best overall consensus reconstruction at a nominal resolution of 3.56 Å. Additionally, to improve the resolution of the GR:FKBP52 region, the ~496,000 particle stack was subjected to signal subtraction of the Hsp90 region. Focused refinement on the Hsp90-subtracted particle stack was then performed (initial angular sampling 1.8°, initial offset range 3 pixels, initial offset step 0.75 pixels, local searches from auto-sampling 1.8°) using a mask including GR and FKBP52 only.

Focused classification without alignment was then performed using a mask including GR and FKBP52 only. Focused refinement on the best class (~107,000 particles) was then performed (initial angular sampling 0.9°, initial offset range 3 pixels, initial offset step 0.75 pixels, local searches from auto-sampling 0.9°) using focused refinement using a mask including GR and FKBP52 only, which yielded a GR:FKBP52 reconstruction with a nominal resolution of 4.31 Å.

For the GR:Hsp90:FKBP51 combined dataset, a particle stack of ~500,000 particles was obtained, representing a GR:Hsp90:FKBP51 reconstruction at a nominal resolution of 4.05 Å. This stack was then subjected to 3D classification without alignment and subsequent 3D refinement on the best classes (~172,000 particles) to obtain the best overall consensus reconstruction at a nominal resolution of 4.18 Å. Additionally, to improve the resolution of the GR:FKBP51 region, the ~500,000 particle stack was subjected to signal subtraction of the Hsp90 region. Focused refinement on the Hsp90-subtracted particle stack was then performed (initial angular sampling 0.9°, initial offset range 3 pixels, initial offset step 1 pixels, local searches from auto-sampling 0.9°) using a mask including GR and FKBP51 only. Focused classification without alignment was then performed using a mask including GR and FKBP51 only. Focused refinement on the best class (~120,000 particles) was then performed (initial angular sampling 0.9°, initial offset range 3 pixels, initial offset step 1 pixels, local searches from auto-sampling 0.9°) using focused refinement using a mask including GR and FKBP51 only, which yielded a GR:FKBP51 reconstruction with a nominal resolution of 4.31 Å.

Per-particle CTF, beam-tilt refinement, trefoil and fourth order aberration refinement, and astigmatism were estimated for both the consensus reconstructions and GR:FKBP focused reconstructions in RELION. The corrected particle stacks were then imported to CryoSparc (v3.3.2), and 2D classification was performed to clean up the particle stacks. The consensus reconstructions were subjected to non-uniform refinement with an envelope mask and a mask including Hsp90 only. The GR:FKBP focused reconstructions were subjected to local refinement with a mask including GR and FKBP only.

All final reconstructions were post-processed in CryoSparc in which the nominal resolution was determined by the gold standard Fourier shell correlation (FSC) using the 0.143 criterion (Extended Data Figs. 2a and 6a). Maps were sharpened in CryoSparc and filtered to their estimated resolution. A composite map for both GR:Hsp90:FKBP51 and GR:Hsp90:FKBP52 was generated by combining the overall consensus refinement map with the GR:FKBP focused refinement map using vop maximum in Chimera. Note that the composite maps were only used for presentation in Figs. 1a and 4a, but not used in atomic model building or refinement.

CryoSparc 3D Variability Analysis was performed for the focused GR:FKBP51 and GR:FKBP52 reconstructions with the following parameters: number of modes to solve, 3; symmetry, C1; filter resolution, 6 Å; filter order, 1.5; high pass order, 8; per-particle scale, optimal; number of iterations, 20; lambda, 0.01.

For both the GR:Hsp90:FKBP51 and GR:Hsp90:FKBP52 complexes, no ligand-free GR complexes were identified during image analysis, despite many rounds of focused classification on GR at various stages of data processing. Only classes with clear ligand density in the GR ligand binding pocket were obtained, suggesting ligand-free GR is either too dynamic or quickly released from the complex, consistent with findings during processing of the GR–maturation complex[13].

## Model building and refinement

For the GR:Hsp90:FKBP52 atomic model, dexamethasone-bound GR LBD and the closed Hsp90 dimer from the GR–maturation complex (PDB ID: 7KRJ) (ref. 13) along with the AlphaFold[77,78] model of human FKBP52 (accession number AF-Q02790) were used as starting models. Additionally, the Hsp90 MEEVD peptide from the FKBP51:Hsp90 MEEVD crystal structure (PDB ID: 5NJX) (ref. 18) was

used. For the GR:Hsp90:FKBP51 atomic model, human FKBP51 from the Hsp90:FKBP51:p23 cryo-EM structure (PDB ID: 7L7I) (ref. 19) was used as a starting model.

Note that the nucleotide density in the Hsp90 NTD pockets were modeled as ATP in both the GR:Hsp90:FKBP51 and GR:Hsp90:FKBP52 atomic models; however, we cannot unambiguously determine whether the density corresponds to ATP or ADP:molybdate due to the difficulty in assigning the γ-phosphate density as molybdate. Our previous work demonstrated that Hsp90 needs to hydrolyze at least one ATP to reach the GR–maturation complex[11], strongly suggesting the nucleotide is in a hydrolyzed ADP:molybdate state. Furthermore, the γ-phosphate density is relatively strong compared to the rest of the map (Extended Data Figs. 3a and 7a), suggesting the presence of molybdate in at least some population of particles. However, Hsp90 may be in a hemi-hydrolyzed state with one protomer bound to ATP and one protomer bound to ADP:molybdate, which we cannot unambiguously determine from the reconstruction. Therefore, we have modeled ATP into the density, which is consistent with our treatment of the GR–maturation complex[13].

Models were refined using Rosetta v.3.11 throughout. Following the split map approach[79] to prevent and monitor overfitting, the Rosetta iterative backbone rebuilding procedure was used to refine models against one of the half maps obtained from RELION, with the other half map only used for validations. Structurally uncharacterized regions, including the FKBP52 TPR:Hsp90 CTD interaction, the FKBP51:HSP90 CTD interaction, and the Hsp90 lumen:GR pre-Helix 1 interaction, were built de novo into consensus maps or focused maps using RosettaCM[80]. The final refinement statistics are provided in Table 1.

## Fluorescence polarization assays

Fluorescence polarization of fluorescent dexamethasone (F-dex) (Thermo Fisher) was measured on a CLARIOstar Plus microplate reader (BMG LabTech) with excitation/emission wavelengths of 485/538 nm, and temperature control set at 25 °C. Buffer conditions were 50 mM HEPES pH 8, 100 mM KCl, 2 mM DTT. For equilibrium ligand binding in Figs. 3e,f and 4f, and Extended Data Fig. 8a,b, proteins were pre-equilibrated together at room temperature for 60 min before F-dex addition. Proteins and reagents were added at the following concentration: 10 nM F-dex, 100 nM GR DBD–LBD, 2 μM Hsp40, 2 μM Bag-1, 15 μM Hsp70, 15 μM Hsp90, 15 μM Hop, 15 μM p23 or p23Δhelix, 15 μM FKBP or FKBP mutants, 5 mM ATP/MgCl$_2$ and 20 mM sodium molybdate where indicated. Note that the dissociation constant ($K_D$) between GR and F-dex is ~150 nM (ref. 11). Ligand binding was initiated with 10 nM F-dex, and association was measured until reaching equilibrium. The plotted equilibrium values in Figs. 3e,f and 4f and Extended Data Fig. 8a,b represent the mean of three biologically independent samples, with error bars representing the standard deviation (s.d.). Polarization values are plotted as the change in polarization from the control sample (10 nM F-dex, 100 nM GR DBD–LBD, 2 μM Hsp40, 2 μM Bag-1, 15 μM Hsp70, 15 μM Hsp90, 15 μM Hop, 15 μM p23 and 5 mM ATP/MgCl$_2$). For equilibrium ligand binding in Extended Data Fig. 5d, proteins were pre-equilibrated together at room temperature for 30 min before F-dex addition. Proteins and reagents were added at the following concentration: 10 nM F-dex, 100 nM GR and 15 μM FKBP51 or FKBP52. Ligand binding was initiated with 10 nM F-dex, and association was measured until reaching equilibrium. The plotted data points for each reaction represent three biologically independent samples. GR ligand binding behavior was affected by buffer conditions; therefore, reactions were always normalized such that each reaction had equivalent amounts of buffer reagents.

## Sequence alignments

For FKBP52 (gene *FKBP4*) sequence alignments in Fig. 2f, sequences were obtained from Uniprot[81], aligned in Clustal Omega[82,83] and visualized in JalView 2.11.1.0 (ref. 84). Sequences in the alignment are:

*H. sapiens* FKBP52, *M. musculus* FKBP52, *R. norvegicus* FKBP52, *D. melanogaster* FKBP52, *T. guttata* FKBP52, *G. gallus* FKBP52, *X. tropicalis* FKBP52 and *H. sapiens* FKBP51 (Uniprot accession codes: Q02790, P30416, Q9QVC8, Q6IQ94, H0ZSE5, A0A3Q3B0L8, A0A310SUH5 and Q13451, respectively). For Fig. 2g, sequences were obtained from Uniprot[81], aligned in Clustal Omega[82,83], and conservation scores were calculated and mapped onto GR from the GR:Hsp90:FKBP52 atomic model using UCSF Chimera v.1.14 (ref. 71). Sequences in the alignment are: *H. sapiens* GR, *M. musculus* GR, *R. norvegicus* GR, *T. guttata* GR, *G. gallus* GR, *X. tropicalis* GR and *D. rerio* GR (Uniprot accession codes: P04150, P06537, P06536, A0A674H6U9, A0A1D5PRD7, Q28E31 and A0A2R8QN75, respectively).

For Extended Data Fig. 9b, the sequences were obtained from Uniprot[81], aligned in Clustal Omega[82,83], and mapped onto GR from the maturation complex using Chimera v.1.14 (ref. 71). Sequences in the alignment are the human SHRs: GR, mineralocorticoid receptor, AR, progesterone receptor, estrogen receptor α and β (Uniprot accession codes: P04150, P08235, P10275, P06401, E3WH19 and Q92731, respectively). Conservation was calculated using percent conservation in Chimera v.1.14 (ref. 71) (with AL2CO[85] parameters (unweighted frequency estimation and entropy-based conservation measurement)).

### Analysis of FKBP52 mutant expression by western blot
Wild-type (JJ762) cells expressing empty vector (e.v., pRS423GPD) or plasmid-borne wild-type or mutant FKBP52 (pRS423GPD-FKBP52) were lysed and subjected to SDS–PAGE (10% acrylamide gel) followed by immunoblot analysis with a monoclonal antibody (1:1,000 dilution) specific for FKBP52 (Hi52b, a gift from Dr. Marc Cox, The University of Texas at El Paso)[23] (Extended Data Fig. 5a). A monoclonal antibody against PGK1 (Invitrogen #459250) was used (1:10,000 dilution) as a loading control.

### In vivo GR activity assays
Relating to Fig. 2e, the effect of overexpression of wild-type FKBP52 on GR activity was determined as previously described[23]. GR activity was measured in the wild-type *Saccharomyces cerevisiae* strain (JJ762) expressing wild-type, human GR on a single-copy plasmid (p414GPD-GR) and the GRE-lacZ reporter plasmid pUCDSS-26X. Wild-type or mutant FKBP52 was expressed in the pRS423GPD plasmid. Cells were grown at 30 °C with shaking overnight in selective media, diluted tenfold and grown to $OD_{600}$ 0.4–0.5. Cultures were split in two, and one set was induced with ligand (50 nM deoxycorticosterone) (Sigma) for 1 h. The β-galactosidase (β-gal) activity of paired samples in the presence and absence of hormone was measured as described using the yeast β-gal assay kit from Thermo Fisher Scientific (catalog number #75768). Assays contained triplicate samples and were conducted at least twice with each mutant. A representative assay is shown.

Fold GR activity was determined by the increase in normalized β-gal activity in the hormone treated sample relative to the untreated paired sample. Relative GR activation was calculated by normalizing the fold GR activity of each sample to the average fold GR activity of strain JJ762 expressing p423GPD (e.v.). The fold increase in GR activities compared to the e.v. control is shown (mean ± s.d.).

### Statistics and reproducibility
All data were tested for statistical significance with Prism v.9.4.0 (Graph-Pad) (n.s. $P > 0.05$; *$P \leq 0.05$; **$P \leq 0.01$; ***$P \leq 0.001$; ****$P \leq 0.0001$). Statistical details (including sample sizes ($n$), $F$-statistics, $P$ values and degrees of freedom) are included in the figure legends for each experiment.

### Reporting summary
Further information on research design is available in the Nature Portfolio Reporting Summary linked to this article.

### Data availability
The cryo-EM maps generated in this study have been deposited in the Electron Microscopy Data Bank (EMDB) under the accession codes EMD-29068 (GR:Hsp90:FKBP52) and EMD-29069 (GR:Hsp90:FKBP51). The atomic coordinates have been deposited in the PDB under the accession code 8FFV (GR:Hsp90:FKBP52) and 8FFW (GR:Hsp90:FKBP51). Publicly available PDB entries used in this study are 7KRJ, 5NJX, 7L7I, 1M2Z, 4LAV, 6TXX, 1P5Q, 1Q1C, 4DRJ, 4DRI, 3O5R, 1A28, 2AA7, 1ERE, 1T7R and AlphaFold AF-Q02790. Protein sequence data for sequence alignments are available from Uniprot. Sequences used in the alignment for Fig. 2f are *H. sapiens* FKBP52, *M. musculus* FKBP52, *R. norvegicus* FKBP52, *D. melanogaster* FKBP52, *T. guttata* FKBP52, *G. gallus* FKBP52, *X. tropicalis* FKBP52 and *H. sapiens* FKBP51 (Uniprot accession codes: Q02790, P30416, Q9QVC8, Q6IQ94, H0ZSE5, A0A3Q3B0L8, A0A310SUH5 and Q13451 respectively). Sequences used in the alignment for Fig. 2g are *H. sapiens* GR, *M. musculus* GR, *R. norvegicus* GR, *T. guttata* GR, *G. gallus* GR, *X. tropicalis* GR and *D. rerio* GR (Uniprot accession codes: P04150, P06537, P06536, A0A674H6U9, A0A1D5PRD7, Q28E31 and A0A2R8QN75, respectively). Sequences used in the alignment for Extended Data Fig. 9b are the human SHRs: GR, mineralocorticoid receptor, AR, progesterone receptor and estrogen receptors α and β (Uniprot accession codes: P04150, P08235, P10275, P06401, E3WH19 and Q92731, respectively). Source data are provided with this paper.

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

## Acknowledgements

We thank members of the Agard Lab, past and present, including R. Wang for the suggestion of this project and E. Kirschke for helpful discussions. We thank members of the José-Maria Carazo lab for continuous flexibility analysis on the cryo-EM datasets. We thank J. Gestwicki for noting the potential importance of differential phosphorylation on the FKBPs. We thank D. Bulkley, G. Gilbert, Z. Yu and E. Tse from the W.M. Keck Foundation Advanced Microscopy Laboratory at the University of California, San Francisco (UCSF) for EM facility maintenance and help with data collection. We also thank M. Harrington and J. Baker-LePain for computational support with the UCSF Wynton cluster. Molecular graphics and analyses were performed with UCSF ChimeraX, developed by the Resource for Biocomputing, Visualization, and Informatics at the University of California, San Francisco, with support from National Institutes of Health R01-GM129325 and the Office of Cyber Infrastructure and Computational Biology, National Institute of Allergy and Infectious Diseases. C.M.N. is a National Cancer Institute Ruth L. Kirschstein Predoctoral Individual NRSA Fellow (F31CA265084-02). The work was supported by NIH grants R35GM118099 (D.A.A.), S10OD020054 (D.A.A.), S10OD021741 (D.A.A.), P20GM104420 (J.L.J.) and R01GM127675 (J.L.J.).

## Author contributions

C.M.N. designed and executed biochemical experiments, cryo-EM sample preparation, data collection, data processing and model building. J.L.J. executed yeast in vivo assays and interpreted the results. C.M.N. and D.A.A. conceived the project, interpreted the results and wrote the manuscript.

## Competing interests

The authors declare no competing interests.

## Additional information

**Extended data** is available for this paper at https://doi.org/10.1038/s41594-023-01128-y.

**Correspondence and requests for materials** should be addressed to David A. Agard.

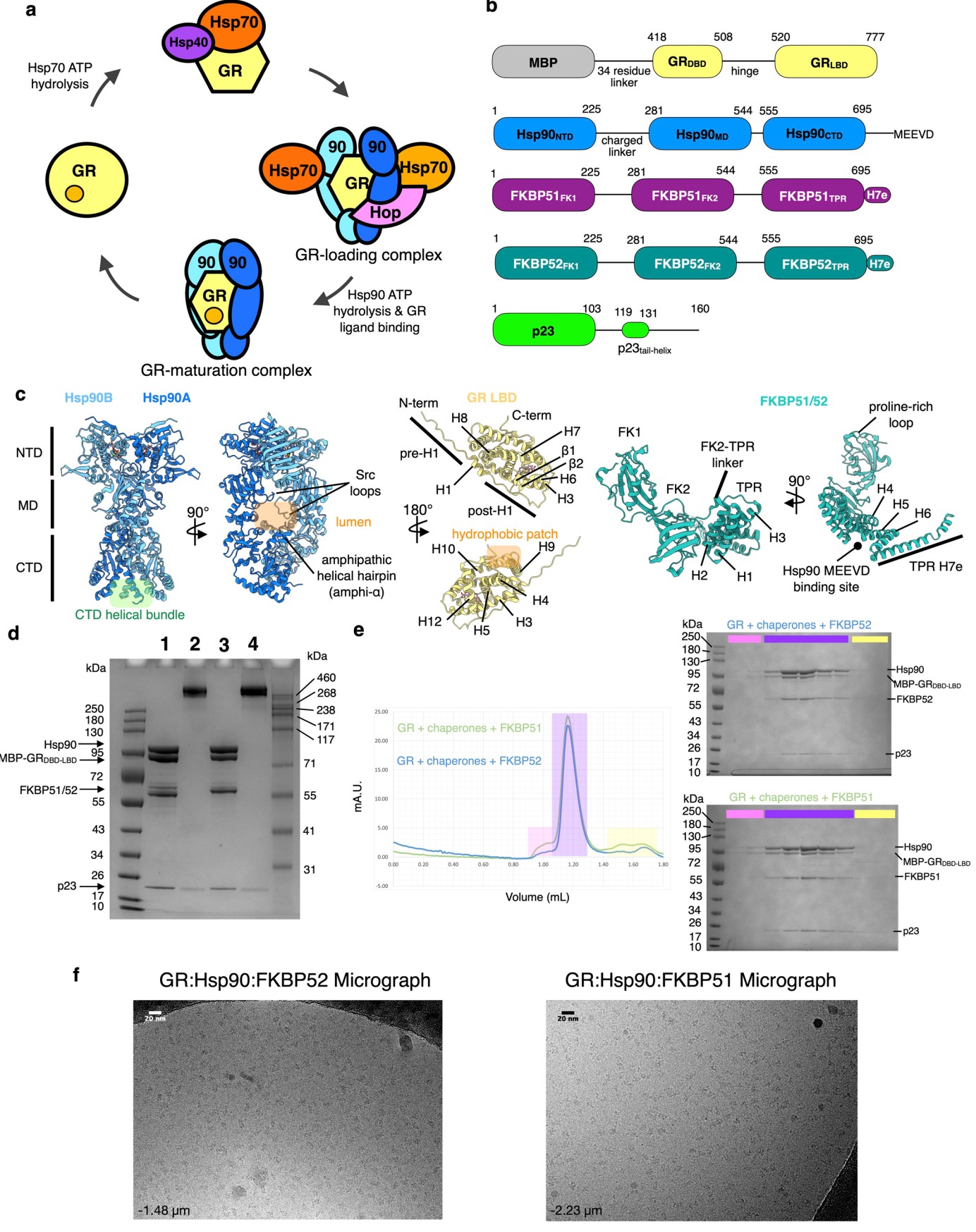

**Extended Data Fig. 1 | See next page for caption.**

**Extended Data Fig. 1 | Sample Preparation. a**, The *in vitro* reconstituted GR chaperone cycle. On the left, GR is active and able to bind ligand. Hsp70, aided by the co-chaperone Hsp40, engages GR and inhibits ligand binding. Hsp70 loads apo GR onto Hsp90 and Hop, which forms the 'GR-loading complex' (PDB ID 7KW7). Hsp70 and Hop are released, Hsp90 hydrolyzes ATP to fully close, and the co-chaperone p23 binds, forming the 'GR-maturation complex' (PDB ID 7KRJ). GR binds ligand in the transition from the GR-loading complex to the GR-maturation complex. In the maturation complex, GR is in a fully folded, native conformation and bound to ligand. Upon Hsp90 re-opening, GR is released from the complex to return to the cycle. **b**, Domain organization of the proteins in the GR:Hsp90:FKBP complexes and p23. **c**, Structural motifs of Hsp90, GR, and FKBP51/52. **d**, Coomassie-stained SDS-PAGE (4-12% acrylamide gel) with MBP-GR pulldown elutions from the *in vitro* reconstituted GR chaperone cycle. Lane 1- elution from MBP-GR pulldown for FKBP51-containing reaction; Lane 2- sample from Lane 1 after size exclusion chromatography (SEC) (**e**) and chemical crosslinking with 0.02% glutaraldehyde; Lane 3- elution from MBP-GR pulldown for FKBP52-containing reaction; Lane 4- sample from Lane 3 after SEC (**e**) and chemical crosslinking with 0.02% glutaraldehyde. This experiment was repeated 7 independent times with similar results. **e**, Size exclusion chromatography (SEC) profile of the elution from the MBP-GR pulldown. The green trace represents the SEC profile from the reconstituted GR chaperone cycle with FKBP51, while the blue trace represents the SEC profile from the reconstituted GR chaperone cycle with FKBP52. mAU = milli-absorbance units. Coomassie-stained SDS-PAGE (4-12% acrylamide gel) of the fractions from SEC corresponding to the GR:Hsp90:FKBP52 sample (top) or the GR:Hsp90:FKBP51 sample (bottom). Colors indicate which gel lanes correspond to specific regions of the SEC profile. Sample fractions from the region highlighted in purple were collected and used for cryo-EM data collection. This experiment was repeated 11 independent times with similar results. **f**, Representative electron micrograph for the cryo-EM dataset of the GR:Hsp90:FKBP52 complex (left) (−1.48 μm defocus) and GR:Hsp90:FKBP51 complex (right) (−2.23 μm defocus). A total of 11,162 and 26,413 micrographs were obtained, respectively.

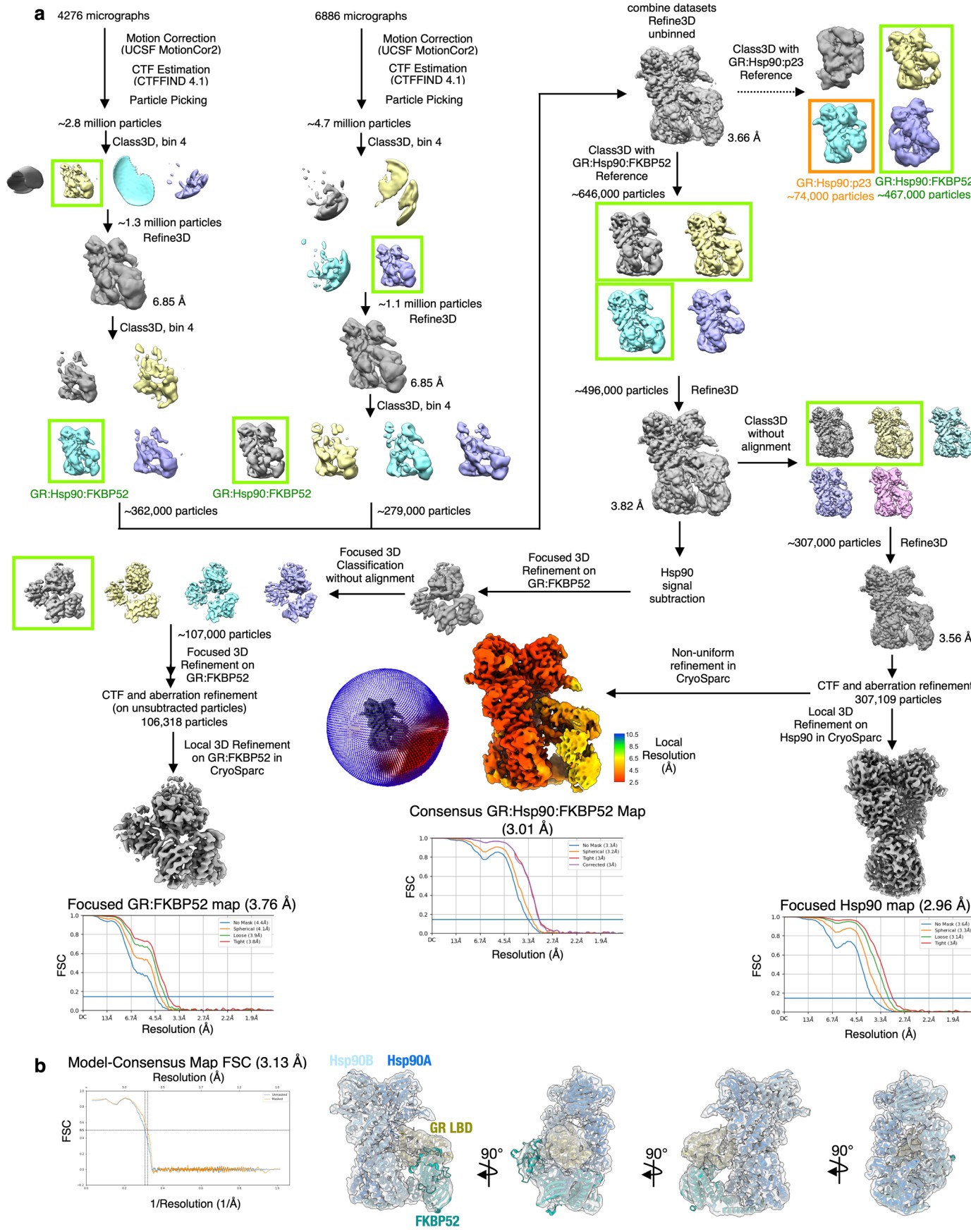

**Extended Data Fig. 2 | See next page for caption.**

**Extended Data Fig. 2 | Cryo-EM Data Analysis for the GR:Hsp90:FKBP52 Complex. a**, Cryo-EM data processing procedure for the GR:Hsp90:FKBP52 complex performed in RELION and CryoSparc. Gold-standard Fourier shell correlation (GSFSC) curves of the final 3D reconstructions, including the focused maps and the consensus map, are shown (bottom). The blue lines intercept the y-axis at an FSC value of 0.143. Angular distribution of particles and local resolution are shown for the consensus map (bottom, middle). **b**, Map-to-model FSC curves between the GR:Hsp90:FKBP52 atomic model and the consensus GR:Hsp90:FKBP52 map, along with different views of the model within the map. The black dotted line intercepts the y-axis at an FSC value of 0.5.

**a**

Hsp90B    Hsp90A

**b**

GR:Hsp90:FKBP52
GR:NCoA2 (1M2Z)
GR:Hsp90:p23 (7KRJ)

180°

NCoA2

Helix 12

**c**

**d**

**e**

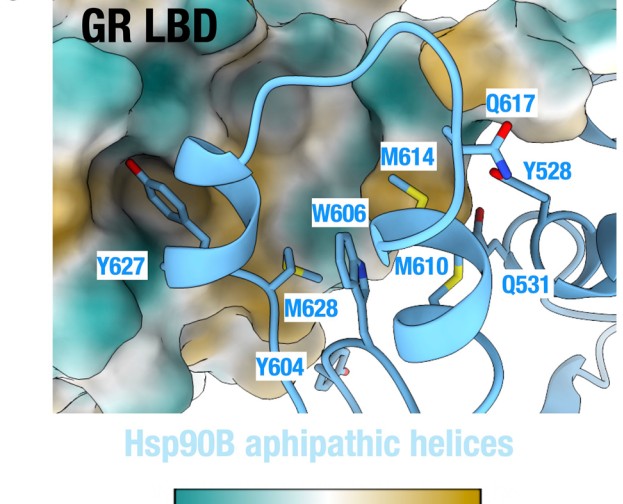

**f**

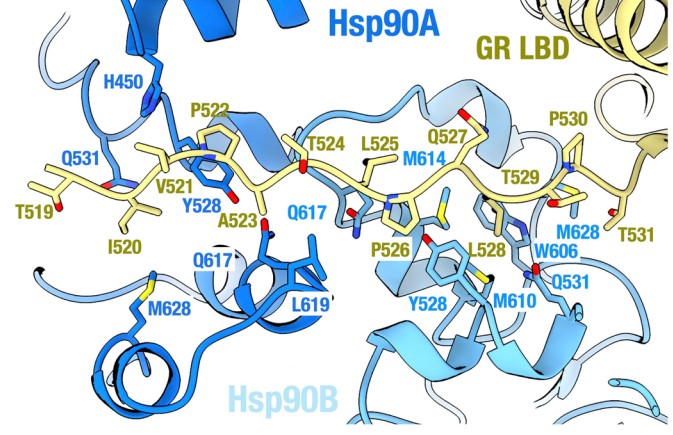

polar          nonpolar

**Extended Data Fig. 3 | See next page for caption.**

**Extended Data Fig. 3 | Hsp90:GR Interfaces in the GR:Hsp90:FKBP52 Complex. a**, Hsp90:GR:FKBP52 complex map density with atomic model showing ATP-magnesium density in both Hsp90 protomers (Hsp90A/B). Bottom images show increased contour level on the map density to indicate that the ATP γ-phosphate position has much stronger density relative to the α and β-phosphates, likely corresponding to molybdate, which may act as a γ-phosphate analog (see Methods). **b**, Atomic model of GR from the GR:Hsp90:FKBP52 complex (yellow) aligned with GR from the crystal structure (PDB ID 1M2Z) (light pink) with co-activator peptide NCoA2 (purple) and GR from the GR-maturation complex structure (PDB ID 7KRJ). GR Helix 12 is indicated. **c-f**, Atomic model of GR:Hsp90:FKBP52 complex with Hsp90A (dark blue), Hsp90B (light blue), GR (yellow). Side chains in contact between GR and Hsp90 are shown, along with hydrogen bonds (dashed pink lines). **c**, Interface 1 of the GR:Hsp90 interaction depicting the GR hydrophobic patch (GR Helices 9 and 10) interacting with the Hsp90A Src loop (Hsp90$^{345-360}$), Hsp90A$^{W320}$, and Hsp90A NTD/MD helices. **d**, Interface 2 of the GR:Hsp90 interaction depicting the GR pre-Helix 1 strand and Helix 1 packing up against the Hsp90B amphipathic α-helices. **e**, Interface 2 of the GR:Hsp90 interaction depicting GR in surface representation colored by hydrophobicity (green = polar, brown = nonpolar) with Hsp90B$^{Y627}$ sticking into the BF3 druggable hydrophobic pocket. **f**, Interface 3 of the GR:Hsp90 interaction depicting the GR pre-Helix 1 strand threading through the Hsp90 lumen between Hsp90A and Hsp90B.

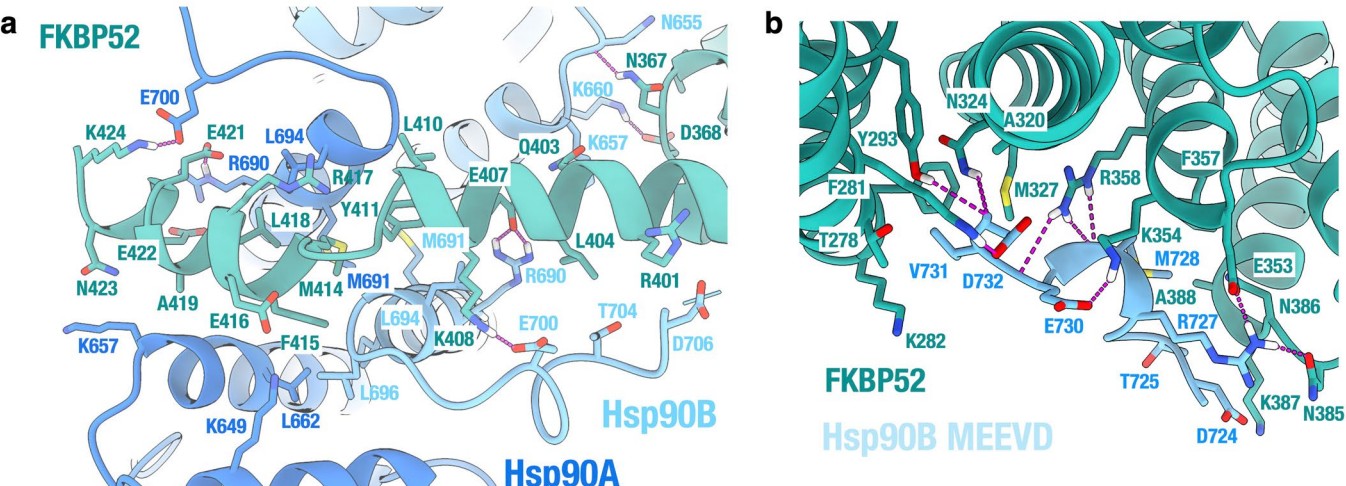

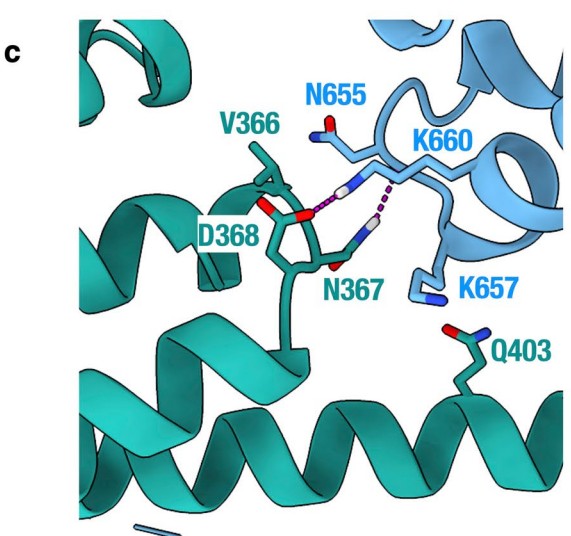

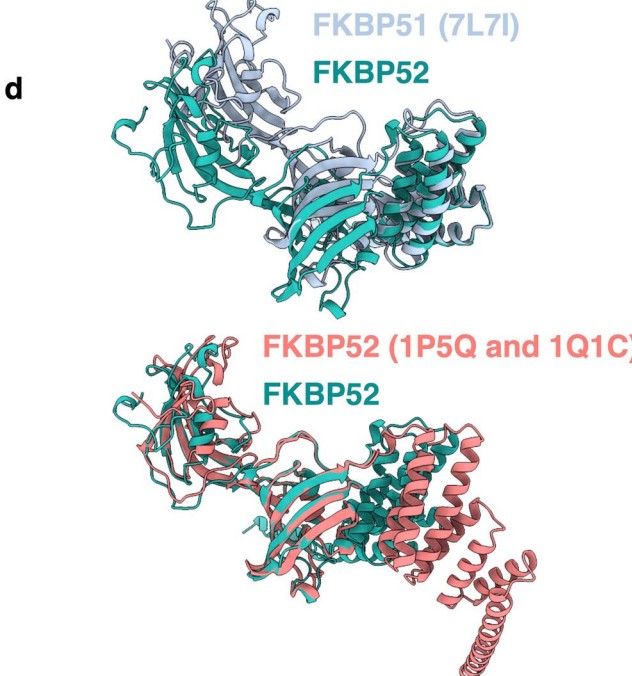

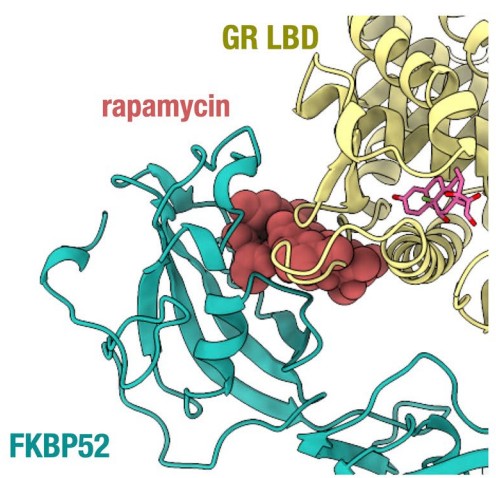

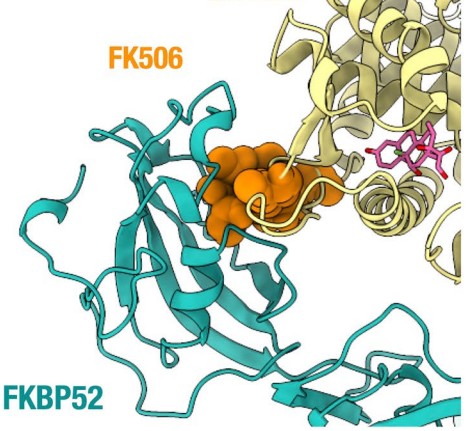

**Extended Data Fig. 4 | See next page for caption.**

**Extended Data Fig. 4 | Hsp90:FKBP52 Interfaces in the GR:Hsp90:FKBP52 Complex.** Atomic model of the GR:Hsp90:FKBP52 complex with Hsp90A (dark blue), Hsp90B (light blue), GR (yellow), and FKBP52 (teal). Side chains in contact between Hsp90 and FKBP52 are shown, along with hydrogen bonds (dashed pink lines). **a**, Interface 1 of the Hsp90:FKBP52 interaction depicting the FKBP52 TPR H7e binding to the Hsp90A/B CTD dimer interface. The helix of FKBP52 H7e breaks to fit into the cleft formed by the Hsp90 CTDs. **b**, Interface 2 of the Hsp90:FKBP52 interaction depicting the Hsp90B MEEVD motif binding the FKBP52 TPR helical bundle. **c**, Interface 3 of the Hsp90:FKBP52 interaction depicting the FKBP52 TPR Helices 5 and 6 binding to the Hsp90B CTD. **d**, FKBP52 (teal) from the GR:Hsp90:FKBP52 atomic model aligned with the cryo-EM structure of FKBP51 (light blue) (PDB ID 7L7I) (top) and crystal structures of FKBP52 (PDB ID 1P5Q, 1Q1C) (bottom) showing the difference in interdomain angles. 1P5Q contains the FKBP52 FK1 and FK2 domain, while 1Q1C contains the FKBP52 FK2 and TPR domains. **e**, The GR:Hsp90:FKBP52 atomic model with FKBP52 (teal), GR (yellow), and dexamethasone (pink) with proline-isomerase inhibitors, rapamycin (brown) or FK506 (orange), docked into the atomic model to indicate the steric clash with GR. Rapamycin was docked in based on the FKBP52:rapamycin crystal structure (PDB ID 4DRJ) and FK506 was docked in based on the FKBP52:FK506:FRB crystal structure (PDB ID 4LAX).

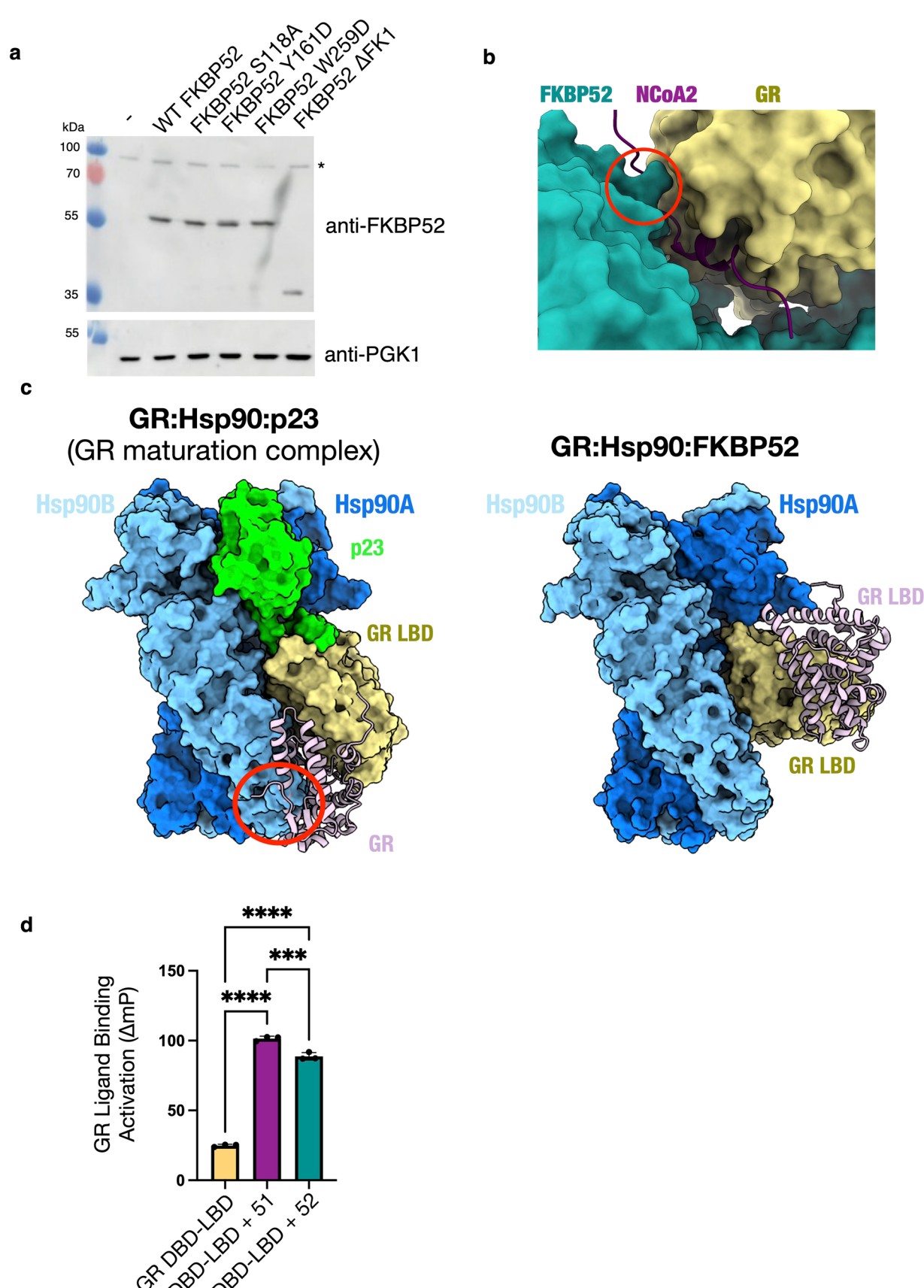

**Extended Data Fig. 5 | See next page for caption.**

**Extended Data Fig. 5 | Analysis of the GR:Hsp90:FKBP52 Structure.**
**a**, Expression of human FKBP52 or FKBP52 mutants in wild-type yeast strain JJ762 assayed by immunoblot with a monoclonal antibody specific for FKBP52. A monoclonal antibody against PGK1 was used as a loading control. The asterisk marks an unknown protein that cross reacts with the anti-FKBP52 antibody. This experiment was performed one time. **b**, Atomic model of the GR:Hsp90:FKBP52 complex shown in surface representation. FKBP52 (teal), GR (yellow). The NCoA2 (nuclear coactivator 2) co-activator peptide is docked in based on the GR:NCoA2 crystal structure (PDB ID 1M2Z). While most of the coactivator peptide binding is sterically permitted, the N-terminus of NCoA2 clashes with the FKBP52 TPR domain (red circle). **c**, Atomic models of the GR-maturation complex (GR:Hsp90:p23) (left) and the GR:Hsp90:FKBP52 complex without FKBP52 (right) depicting that GR LBD dimerization is permitted once FKBP52 is released. Hsp90A (dark blue, surface representation), Hsp90B (light blue, surface representation), GR (yellow, surface representation), p23

(green, surface representation). In both complexes, the GR LBD dimerization site is accessible, however; binding of the second GR LBD (light pink) to the GR-maturation complex clashes with the Hsp90B CTD, shown with a red circle (left). Binding of the second GR LBD (light pink) to the GR:Hsp90:FKBP52 complex (right). Docking of the dimerized GR LBD is based on the GR LBD dimer crystal structure (PDB ID 1M2Z). **d**, Equilibrium binding of 10 nM fluorescent dexamethasone to 100 nM GR DBD-LBD with addition of 15 µM FKBP51 ('51') or FKBP52 ('52') measured by fluorescence polarization (mean ± SD). n = 3 biologically independent samples per condition. Fluorescence polarization values are baseline subtracted in accordance with the measured fluorescent dexamethasone baseline polarization value. Statistical significance was evaluated by an ordinary one-way ANOVA ($F_{(2,6)}$ = 1414, p < 0.0001) with *post-hoc* Tukey's multiple comparisons test. P-values: p(GR vs. GR + 51) < 0.0001, p(GR vs. GR + 52) < 0.0001, p(GR + 51 vs. GR + 52) = 0.004. (n.s. P > 0.05; * P ≤ 0.05; ** P ≤ 0.01; *** P ≤ 0.001; **** P ≤ 0.0001). Source data

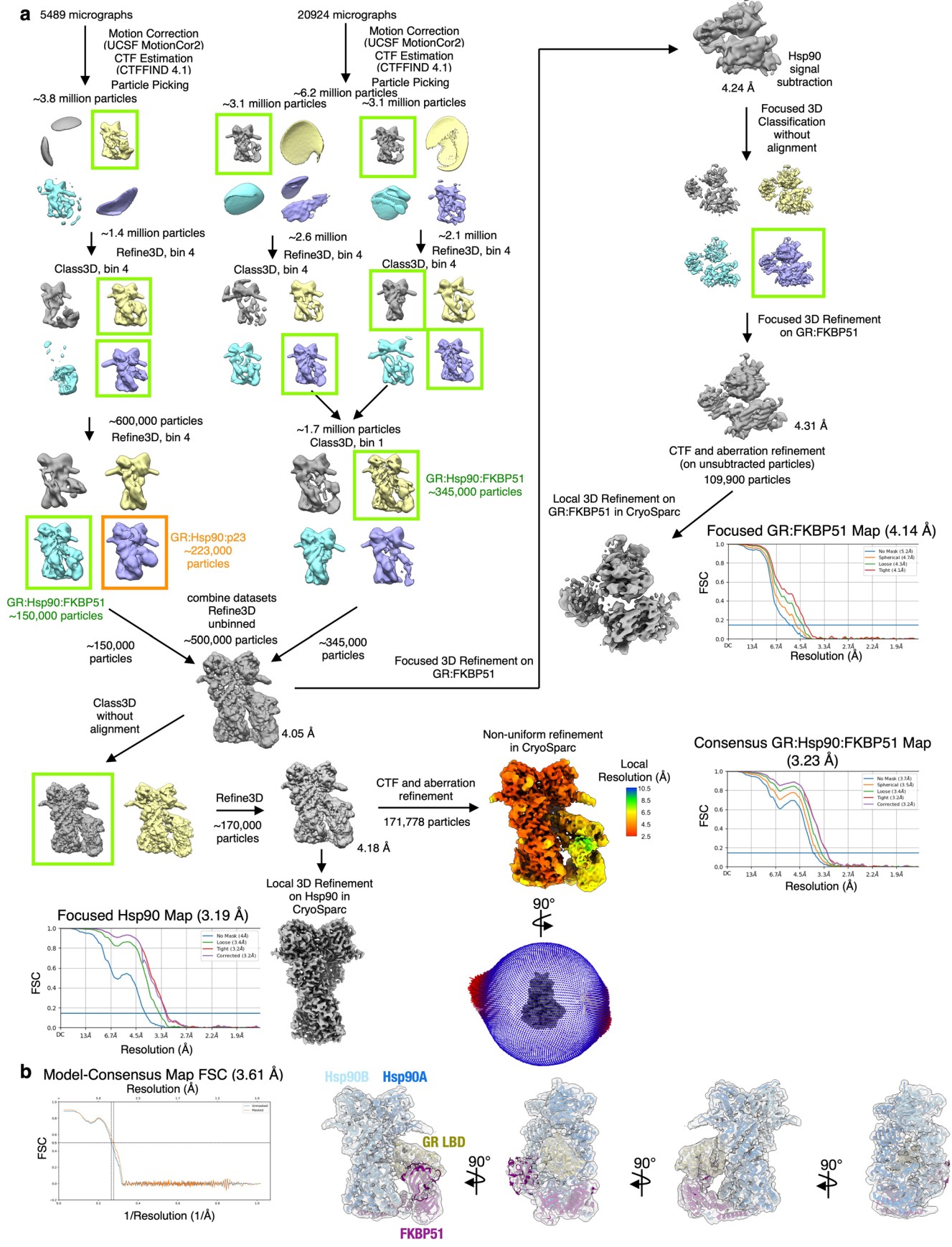

**Extended Data Fig. 6 | See next page for caption.**

**Extended Data Fig. 6 | Cryo-EM Data Analysis for the GR:Hsp90:FKBP51 Complex. a**, Cryo-EM data processing procedure for the GR:Hsp90:FKBP51 complex performed in RELION and CryoSparc. Gold-standard Fourier shell correlation (GSFSC) curves of the final 3D reconstructions, including the focused maps and the consensus map, are shown (bottom). The blue lines intercept the y-axis at an FSC value of 0.143. Angular distribution of particles and local resolution are shown for the consensus map (bottom, middle). **b**, Map-to-model FSC curves between the GR:Hsp90:FKBP51 atomic model and the consensus GR:Hsp90:FKBP51 map, along with different views of the model within the map. The black dotted line intercepts the y-axis at an FSC value of 0.5.

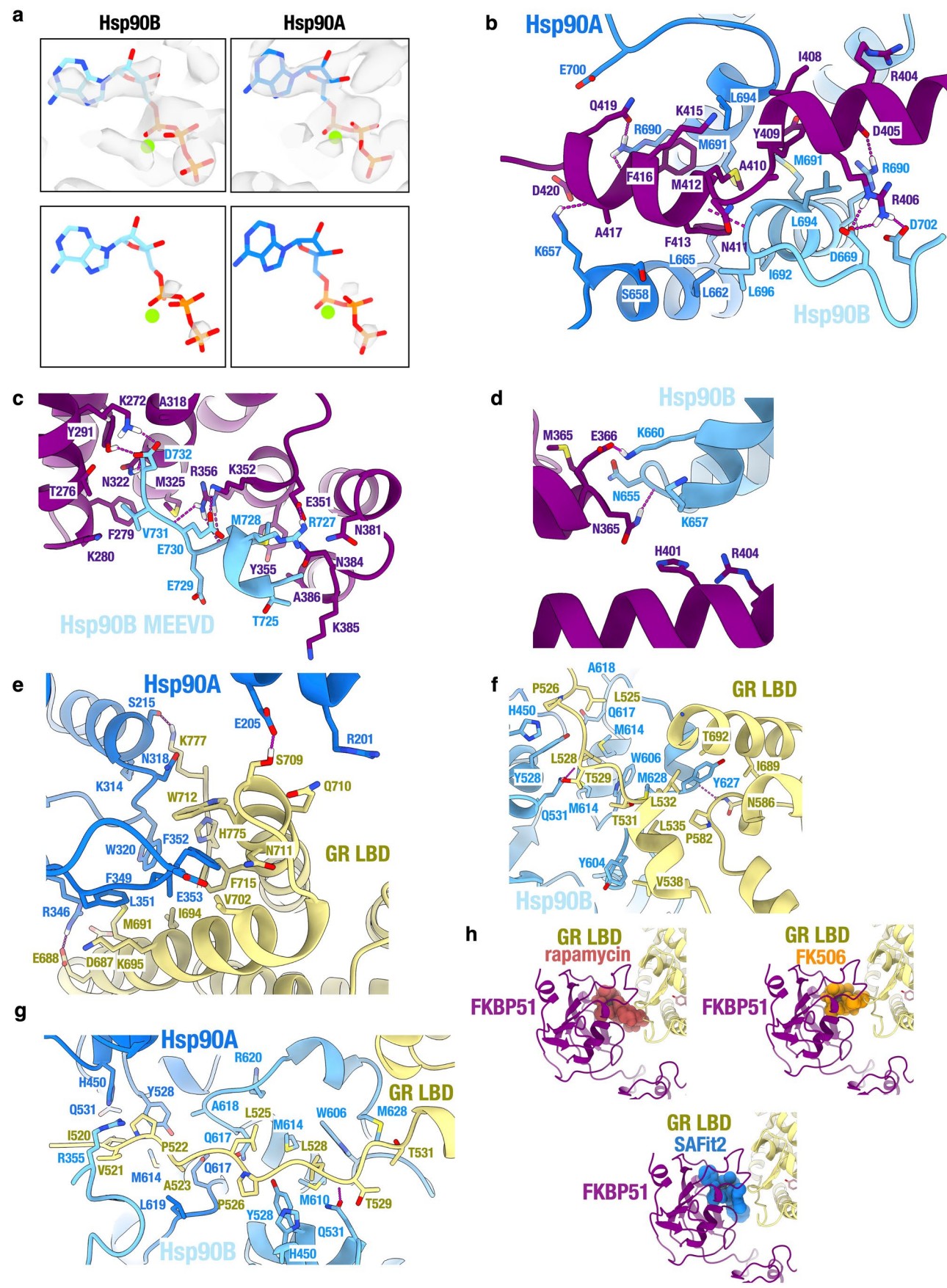

**Extended Data Fig. 7 | See next page for caption.**

**Extended Data Fig. 7 | Interfaces in the GR:Hsp90:FKBP51 Complex.** Atomic model of the GR:Hsp90:FKBP51 complex with Hsp90A (dark blue), Hsp90B (light blue), GR (yellow), and FKBP51 (purple). Side chains in contact between Hsp90 and FKBP51 or Hsp90 and GR are shown, along with hydrogen bonds (dashed pink lines). **a**, Hsp90:GR:FKBP51 complex map density with atomic model showing ATP-magnesium density in both Hsp90 protomers (Hsp90A/B). Bottom images show increased contour level on the map density to indicate that the ATP γ-phosphate position has much stronger density relative to the α and β-phosphates, likely corresponding to molybdate, which may act as a γ-phosphate analog (see Methods). **b**, Interface 1 of the Hsp90:FKBP51 interaction depicting the FKBP51 TPR H7e binding to the Hsp90A/B CTD dimer interface. The helix of FKBP51 H7e breaks to fit into the cleft formed by the Hsp90 CTDs. **c**, Interface 2 of the Hsp90:FKBP51 interaction depicting the Hsp90B MEEVD motif binding the FKBP51 TPR helical bundle. **d**, Interface 3 of the Hsp90:FKBP51 interaction depicting the FKBP51 TPR Helices 5 and 6 binding to the Hsp90B CTD. **e**, Interface 1 of the GR:Hsp90 interaction depicting the GR hydrophobic patch (GR Helices 9 and 10) interacting with the Hsp90A Src loop (Hsp90$^{345–360}$), Hsp90A$^{W320}$, and Hsp90A NTD/MD helices. **f**, Interface 2 of the GR:Hsp90 interaction depicting GR pre-Helix 1 strand and Helix 1 packing up against the Hsp90B amphipathic α-helices. **g**, Interface 3 of the GR:Hsp90 interaction depicting the GR pre-Helix 1 strand threading through the Hsp90 lumen between Hsp90A and Hsp90B. **h**, The GR:Hsp90:FKBP51 atomic model with FKBP51 (purple), GR (yellow), and dexamethasone (pink) with proline-isomerase inhibitors, rapamycin (brown) or FK506 (orange), docked into the atomic model to indicate the steric clash with GR. Rapamycin was docked in based on the FKBP51:rapamycin crystal structure (PDB ID 4DRI) and FK506 was docked in based on the FKBP51:FK506:FRB crystal structure (PDB ID 3O5R). The FKBP51-specific inhibitor SAFit2 was docked into the atomic model to indicate there is no steric clash with GR at the backbone level (although some side chains clash). SAFit2 was docked in based on the FKBP51:SAFit2 crystal structure (PDB ID 6TXX).

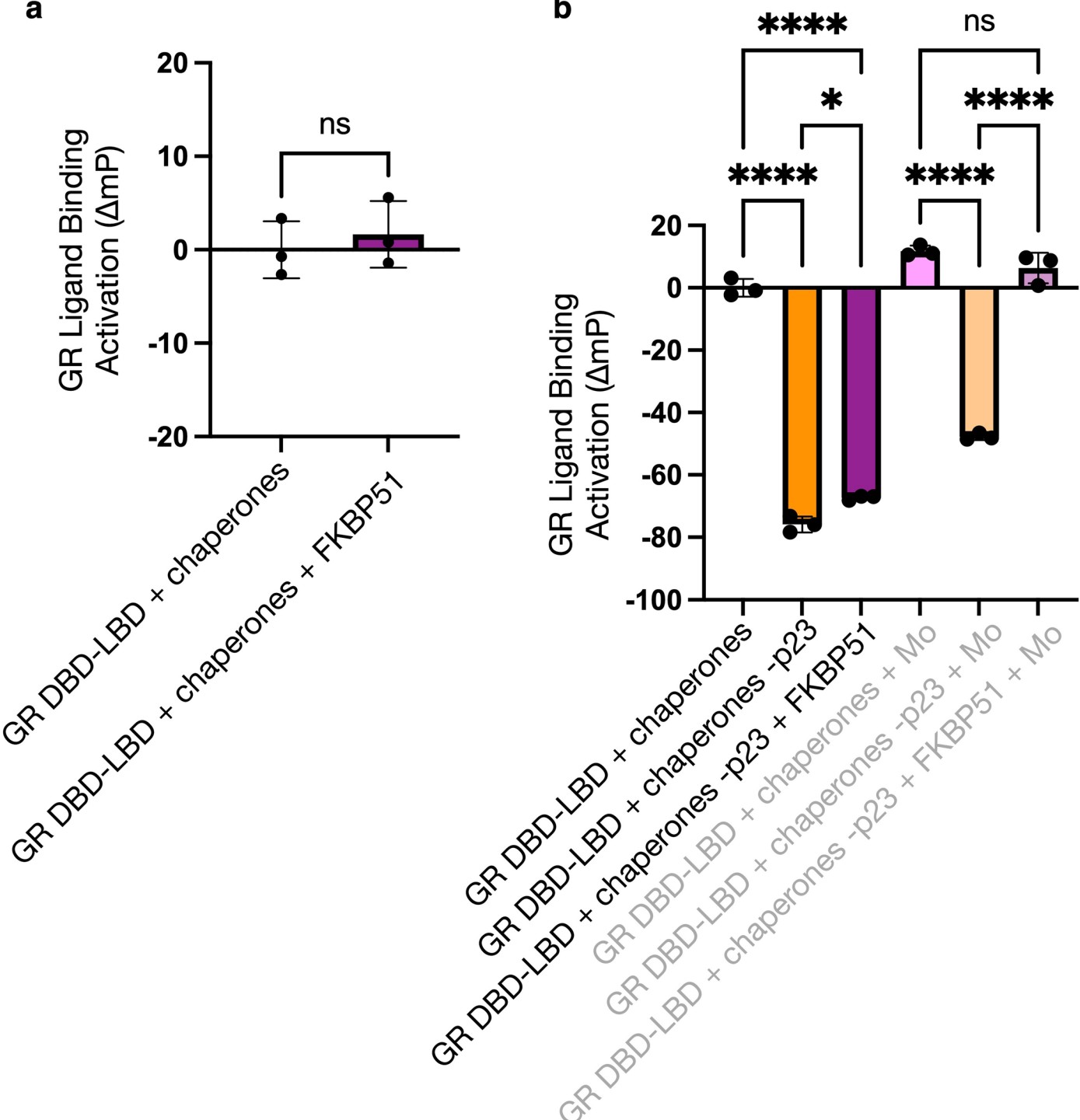

**Extended Data Fig. 8 | Effect of FKBP51 on GR Ligand Binding *in vitro*.**
**a**, Equilibrium binding of 10 nM fluorescent dexamethasone to 100 nM GR DBD-LBD with chaperones and 15 μM FKBP51 (mean ± SD). n = 3 biologically independent samples per condition. 'Chaperones' = 15 μM Hsp70, Hsp90, Hop, and p23; 2 μM Ydj1 and Bag-1. Statistical significance was evaluated by an unpaired two-tailed *t*-test, p-value = 0.5737. (n.s. P > 0.05; *P ≤ 0.05; **P ≤ 0.01; ***P ≤ 0.001; ****P ≤ 0.0001). **b**, Equilibrium binding of 10 nM fluorescent dexamethasone to 100 nM GR DBD-LBD with chaperones, 15 μM FKBP51, and 20 mM sodium molybdate ('Mo') (mean ± SD). n = 3 biologically independent

samples per condition. 'Chaperones' = 15 μM Hsp70, Hsp90, Hop, and p23; 2 μM Ydj1 and Bag-1. Statistical significance was evaluated by an ordinary one-way ANOVA (F$_{(5,12)}$ = 647.1, p < 0.0001) with *post-hoc* Šídák's multiple comparisons test. P-values: p(Chaperones vs. Chaperones -p23) < 0.0001, p(Chaperones vs. Chaperones -p23 + 51) < 0.0001, p(Chaperones -p23 vs. Chaperones -p23 + 51) = 0.0123, p(Chaperones + Mo. vs. Chaperones -p23 + Mo.) < 0.0001, p(Chaperones + Mo. vs. Chaperones -p23 + 51 + Mo.) = 0.1640, p(Chaperones -p23 + Mo. vs. Chaperones -p23 + 51 + Mo.) < 0.0001. (n.s. P > 0.05; *P ≤ 0.05; **P ≤ 0.01; ***P ≤ 0.001; ****P ≤ 0.0001).

**a**

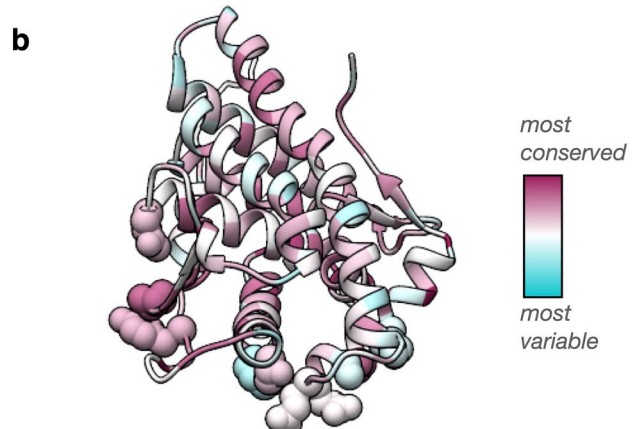

GR PR MR

ER AR

**b**

most
conserved

most
variable

**Extended Data Fig. 9 | See next page for caption.**

**Extended Data Fig. 9 | Modeling FKBP Binding to All Five Steroid Hormone Receptors (SHRs). a**, FKBP52 bound to the glucocorticoid receptor (GR, yellow), progesterone receptor (PR, tan), mineralocorticoid receptor (MR, orange), estrogen receptor (ERα, pink), or androgen receptor (AR, green) based on the structure of the GR:Hsp90:FKBP52 complex. Due to the structural conservation of the LBDs across the five SHRs, all SHRs fit well with FKBP52 using the GR:Hsp90:FKBP52 atomic model, with no backbone clashes between the SHRs and FKBP52. The PDB IDs used to dock in the SHRs are as follows: PR (1A28), MR (2AA7), ERα (1ERE), AR (1T7R). **b**, Sequence conservation across human steroid hormone receptors (GR, MR, ERα, ERβ, AR) plotted onto the GR structure from the GR:Hsp90:FKBP52 atomic model. Residues are colored from most variable (cyan) to most conserved (maroon). Residues that interact with FKBP51/52 are shown as spheres.

# Reporting Summary

## Statistics

For all statistical analyses, confirm that the following items are present in the figure legend, table legend, main text, or Methods section.

| n/a | Confirmed | |
|---|---|---|
| ☐ | ☒ | The exact sample size (*n*) for each experimental group/condition, given as a discrete number and unit of measurement |
| ☐ | ☒ | A statement on whether measurements were taken from distinct samples or whether the same sample was measured repeatedly |
| ☐ | ☒ | The statistical test(s) used AND whether they are one- or two-sided *Only common tests should be described solely by name; describe more complex techniques in the Methods section.* |
| ☒ | ☐ | A description of all covariates tested |
| ☐ | ☒ | A description of any assumptions or corrections, such as tests of normality and adjustment for multiple comparisons |
| ☐ | ☒ | A full description of the statistical parameters including central tendency (e.g. means) or other basic estimates (e.g. regression coefficient) AND variation (e.g. standard deviation) or associated estimates of uncertainty (e.g. confidence intervals) |
| ☐ | ☒ | For null hypothesis testing, the test statistic (e.g. *F*, *t*, *r*) with confidence intervals, effect sizes, degrees of freedom and *P* value noted *Give P values as exact values whenever suitable.* |
| ☒ | ☐ | For Bayesian analysis, information on the choice of priors and Markov chain Monte Carlo settings |
| ☒ | ☐ | For hierarchical and complex designs, identification of the appropriate level for tests and full reporting of outcomes |
| ☒ | ☐ | Estimates of effect sizes (e.g. Cohen's *d*, Pearson's *r*), indicating how they were calculated |

*Our web collection on statistics for biologists contains articles on many of the points above.*

## Software and code

Policy information about availability of computer code

| Data collection | Cryo-EM data collection was done on a Titan Krios (Thermo Fischer Scientific) electron microscope (300kV) with a Gatan K3 direct electron detector (Gatan) equipped with a Bioquantum energy filter (Gatan) set to a slit width of 20 eV. Automatic data collection was done with SerialEM v.4.0. |
|---|---|
| Data analysis | EM data processing was done with RELION v.3.0.8, CryoSparc v.3.3.2, UCSF MotionCor2, and CTFFIND v.4.1. Figures were created with UCSF Chimera v.1.14, UCSF ChimeraX v.1.0.0, and BioRender.com. Model building was done using Rosetta v.3.11 and AlphaFold (https://alphafold.ebi.ac.uk/). Graphical data was plotted and statistical analysis was performed using Prism (GraphPad) v.9.4.0. Sequence alignments were performed using Uniprot, Clutal Omega (https://www.ebi.ac.uk/Tools/msa/clustalo/), and JalView (v.2.11.1.0). |

For manuscripts utilizing custom algorithms or software that are central to the research but not yet described in published literature, software must be made available to editors and reviewers. We strongly encourage code deposition in a community repository (e.g. GitHub). See the Nature Portfolio guidelines for submitting code & software for further information.

## Data

Policy information about **availability of data**

All manuscripts must include a **data availability statement**. This statement should provide the following information, where applicable:
- Accession codes, unique identifiers, or web links for publicly available datasets
- A description of any restrictions on data availability
- For clinical datasets or third party data, please ensure that the statement adheres to our **policy**

The cryo-EM maps generated in this study have been deposited in the Electron Microscopy Data Bank (EMDB) under the accession codes EMD-29068 (GR:Hsp90:FBKP52) and EMD-29069 (GR:Hsp90:FKBP51). The atomic coordinates have been deposited in the PDB under the accession code 8FFV (GR:Hsp90:FKBP52) and 8FFW (GR:Hsp90:FKBP51). Publicly available PDB entries used in this study are: 7KRJ, 5NJX, 7L7I, 1M2Z, 4LAV, 6TXX and AlphaFold AF-Q02790. Protein sequence data for sequence alignments are available from Uniprot. Sequences in the FKBP51/52 alignment are: H. sapiens FKBP52, M. musculus FKBP52, R. norvegicus FKBP52, D. melanogaster FKBP52, T. guttata FKBP52, G. gallus FKBP52, X. tropicalis FKBP52, and H. sapiens FKBP51 (Uniprot accession codes: Q02790, P30416, Q9QVC8, Q6IQ94, H0ZSE5, A0A3Q3B0L8, A0A310SUH5, Q13451 respectively). Sequences in the GR alignment are: H. sapiens GR, M. musculus GR, R. norvegicus GR, T. guttata GR, G. gallus GR, X. tropicalis GR, D. rerio GR (Uniprot accession codes: P04150, P06537, P06536, A0A674H6U9, A0A1D5PRD7, Q28E31, A0A2R8QN75, respectively). Sequences in the human steroid hormone receptor alignment are: glucocorticoid receptor, mineralocorticoid receptor, androgen receptor, progesterone receptor, estrogen receptor α and β (Uniprot accession codes: P04150, P08235, P10275, P06401, E3WH19, Q92731, respectively).

## Human research participants

Policy information about **studies involving human research participants and Sex and Gender in Research.**

| | |
|---|---|
| Reporting on sex and gender | N/A |
| Population characteristics | N/A |
| Recruitment | N/A |
| Ethics oversight | N/A |

Note that full information on the approval of the study protocol must also be provided in the manuscript.

# Field-specific reporting

Please select the one below that is the best fit for your research. If you are not sure, read the appropriate sections before making your selection.

☒ Life sciences ☐ Behavioural & social sciences ☐ Ecological, evolutionary & environmental sciences

For a reference copy of the document with all sections, see **nature.com/documents/nr-reporting-summary-flat.pdf**

# Life sciences study design

All studies must disclose on these points even when the disclosure is negative.

| | |
|---|---|
| Sample size | No statistical method was used for sample size calculation. All in vitro biochemical experiments were performed using three biological replicates to account for pipetting variability. All in vivo yeast assays were performed using three biological replicates to account for pipetting variability and biological variability between yeast cultures. For cryoEM reconstructions, sample sizes were determined by available electron microscopy time and the number of particles on each micrograph obtained during the collection time. |
| Data exclusions | No data was excluded. |
| Replication | All experiments were confirmed with multiple biological replicates as detailed in the figure legends. |
| Randomization | No randomization was performed, since this study did not allocate experimental groups. |
| Blinding | No blinding was performed, since it is not relevant to this study. |

# Reporting for specific materials, systems and methods

We require information from authors about some types of materials, experimental systems and methods used in many studies. Here, indicate whether each material, system or method listed is relevant to your study. If you are not sure if a list item applies to your research, read the appropriate section before selecting a response.

## Materials & experimental systems

| n/a | Involved in the study |
|---|---|
| ☐ | ☒ Antibodies |
| ☒ | ☐ Eukaryotic cell lines |
| ☒ | ☐ Palaeontology and archaeology |
| ☒ | ☐ Animals and other organisms |
| ☒ | ☐ Clinical data |
| ☒ | ☐ Dual use research of concern |

## Methods

| n/a | Involved in the study |
|---|---|
| ☒ | ☐ ChIP-seq |
| ☒ | ☐ Flow cytometry |
| ☒ | ☐ MRI-based neuroimaging |

## Antibodies

| | |
|---|---|
| Antibodies used | Anti-FKBP52 (monoclonal), Hi52b, 1:1000 dilution, a gift from Dr. Marc Cox, The University of Texas at El Paso<br>Anti-PGK1 (monoclonal, Invitrogen #459250), Clone 22C5D8, RRID AB_2532235, 1:10000 dilution |
| Validation | For Anti-FKBP52, see: Riggs, D. L. et al. The Hsp90-binding peptidylprolyl isomerase FKBP52 potentiates glucocorticoid signaling in vivo. The EMBO Journal 22, 1158-1167, doi:10.1093/emboj/cdg108 (2003).<br>Anti-PGK1 has been validated by Western blot using a range of different dilutions in ~200 publications, see: https://www.thermofisher.com/antibody/product/PGK1-Antibody-clone-22C5D8-Monoclonal/459250 |

