## [Peer Review File · Nature Structural & Molecular Biology]

Peer Review Information

Manuscript Title: Cryo-EM reveals how Hsp90 and FKBP immunophilins co-regulate the Glucocorticoid Receptor

Corresponding author name(s): David Agard

Reviewer Comments & Decisions:

Decision Letter, initial version:

Message: 21st Mar 2023

Dear Dr. Agard,

Thank you again for submitting your manuscript "Cryo-EM reveals how Hsp90 and FKBP immunophilins co-regulate the Glucocorticoid Receptor". We now have comments (below) from the 4 reviewers who evaluated your paper. In light of those reports, we remain interested in your study and would like to see your response to the comments of the referees, in the form of a revised manuscript.

You will see that while all reviewers appreciate the results they raise several concerns which would need to be addressed. Specifically, competition between p23 and FKBP52 for GR binding is brought up by several reviewers, and agree that additional experimental support is needed to establish this, following guidance from reviewers. In line with reviewer's #2 comments, additional in vitro experiments to confirm disruption of interactions in interface mutants should be considered. Editorially, we agree that extending the discussion of the ligand binding state of GR would strengthen the manuscript.

Please be sure to address/respond to all concerns of the referees in full in a point-by-point response and highlight all changes in the revised manuscript text file. If you have comments that are intended for editors only, please include those in a separate cover letter.

We expect to see your revised manuscript within 6 weeks. If you cannot send it within this

time, please contact us to discuss an extension; we would still consider your revision, provided that no similar work has been accepted for publication at NSMB or published elsewhere.

Reporting Summary:

Please note that all key data shown in the main figures as cropped gels or blots should be presented in uncropped form, with molecular weight markers. These data can be aggregated into a single supplementary figure item. While these data can be displayed in a relatively informal style, they must refer back to the relevant figures. These data should be submitted with the final revision, as source data, prior to acceptance, but you may want to start putting it together at this point.

SOURCE DATA: we urge authors to provide, in tabular form, the data underlying the graphical representations used in figures. This is to further increase transparency in data reporting, as detailed in this editorial (<http://www.nature.com/nsmb/journal/v22/n10/full/nsmb.3110.html>). Spreadsheets can be submitted in excel format. Only one (1) file per figure is permitted; thus, for multi-paneled figures, the source data for each panel should be clearly labeled in the Excel file; alternately the data can be provided as multiple, clearly labeled sheets in an Excel file.

When submitting files, the title field should indicate which figure the source data pertains to. We encourage our authors to provide source data at the revision stage, so that they are part of the peer-review process.

Data availability: this journal strongly supports public availability of data. All data used in accepted papers should be available via a public data repository, or alternatively, as Supplementary Information. If data can only be shared on request, please explain why in your Data Availability Statement, and also in the correspondence with your editor. Please note that for some data types, deposition in a public repository is mandatory - more information on our data deposition policies and available repositories can be found below: <https://www.nature.com/nature-research/editorial-policies/reporting-standards#availability-of-data>

[redacted]

Sincerely,
Kat

Katarzyna Ciazynska
(she/her)
Associate Editor
Nature Structural & Molecular Biology
<https://orcid.org/0000-0002-9899-2428>

Referee expertise:

Referee #1: protein folding, structural biology

Referee #2: chaperones, structural biology

Referee #3: chaperones, structural biology

Referee #4: Hsp90 function

Reviewers' Comments:

Reviewer #1:

Remarks to the Author:

They authors present the cryo-EM structure of a complex consisting of the co-chaperones FKBP51 or FKBP52 bound to Hsp90 and the glucocorticoid receptor (GR). This complex is part of the maturation cycle of GR providing an additional regulation step in the signaling of this steroid hormone receptor. Despite high similarity in sequence and structure, FKBP51 and FKBP52 show antagonistic effects on the activity of GR in vivo. Interestingly, the two structures reveal that both FKBP51 or FKBP52 interact directly with the GR bound to Hsp90. Although this interaction is similar for FKBP51 and FKBP52 as deduced from the cryo-EM structure, the effects on ligand binding by GR are different which suggests that subtle differences in binding exist. This is an interesting, well performed study that provides new insight into the structure and function of the Hsp90 system and GR signaling.

Specific comments:

1. The conclusion that FKBP52 functionally replaces p23 tail-helix is plausible but not demonstrated directly in this study. Ligand binding was increased in both settings, but the underlying mechanisms could be different.
2. The extended figure 3a shows different orientations of the ATP for the two HSP90 protomers. Is that really the case? Is what the authors observe really ATP and not ADP + molybdate, as described by Gruszczuk et al. (2022 Nat Comm.) under similar conditions?
3. In extended figure 3b, it would be useful to also show the GR conformation in the current complex vs its conformation in the previously published HSP90:GR:p23 complex.
4. One of the differences in the conditions used for the current structure vs the HSP90:FKBP51:p23 complex presented by Lee et al. (2021 Mol. Cell) is the use of ATP+molybdate rather than AMP-PNP. The ADP+molybdate state is supposed to correspond to a post-hydrolysis state while the AMP-PNP could be equivalent to a pre-hydrolysis state. Would it be possible that the exclusion of p23 from the client complex is rather due to the change in nucleotide state rather than to the rotation of the client?

Would that fit to the results presented in figure 3f and extended data figure 8?

Reviewer #2:

Remarks to the Author:

The Glucocorticoid receptor (GR) is highly dependent on HSP70 and Hsp90 chaperone complexes for function. The authors of this manuscript have previously determined the cryoEM structures of a Hsp70-Hsp90-Hop-GR complex– referred to as the loading complex and the Hsp90-P23-GR maturation complex, which provide remarkable insight into how a partially unfolded GR is bound by Hsp70, which transfers it to a Hsp90-HOP complex in a state which ligand can bind to GR. The ligand bound GR is then shown to be in a folded state, stabilised by Hsp90 and P23 in the complex. The new findings presented in this manuscript are significant in further revealing how immunophilins FKBP51 and FKBP52 assemble into a Hsp90-GR complex, making direct interactions with both Hsp90 and GR in the regulation of the GR chaperone cycle.

Summary of key results:

1. Preparation of GR-Hsp90-FKBP51/FKBP52 complexes using an in-vitro reconstitution system developed in previous work by the authors, incorporating Hsp70, Hsp90, Hsp40, Hop, Hsp90, P23, FKBP51 or FKBP52 and MBP-tagged GR DBD-LBD. This complex was stabilised with sodium molybdate and cross-linked for Cryo-EM studies.
2. These first CryoEM structures of Hsp90-GR-FKBP52 (3.01 Å) and Hsp90-GR-FKBP51 (3.23 Å) complexes reveal a fully closed Hsp90 dimer in complex with a single GR and single FKBP51 or FKBP52 bound and the absence of P23.
3. In these GR-client bound complex structures, FKBP51 and FKBP52 interact with Hsp90 in a similar manner as observed in a Hsp90-P23-FKBP51 structure previously by Lee et al, Mol Cell, 2021.
4. The structures reveal that in these Hsp90-GR-FKBP51/FKBP52 complexes, the GR client interacts with HSP90 through three interfaces and HSP90 stabilises the GR in a folded – ligand bound state distinct, from that observed in a previous GR-Hsp90-P23 complex structure determined (Noddings et al, Nature, 2022)
5. Completing a three-way interaction, the structures reveal that the FBKP51 and FKBP52 make direct contacts with the GR-client, identifying three main interaction interfaces between the two. To test interface 1, the authors show that a single point mutation in the proline rich loop of FKBP52 (S118A) reduces FKBP52 dependant GR potentiation and suggest that phosphorylation at this site may help promote the interaction between the proline rich loop and GR in-vivo. They further show that mutation of Y161D within interface 2 and W259D within the FKBP52-FK2-TPR linker which forms interface 3 with GR, also reduces FKBP52-dependant GR potentiation.
6. Despite P23 being present in the reconstitution mix when preparing the complex, no density is visible for the P23 in either structures and the authors suggest this is due to FKBP52 competing with P23 for GR-Hsp90 binding through allostery. It is proposed that binding of FKBP52 to the rotated position of GR observed in the client-bound complexes is not compatible with P23 binding.
7. Consistent with in-vivo findings (Riggs et al, EMBO, 2003), the authors found that FKBP52 binding to the GR-Hsp90 complex potentiates ligand binding in-vitro. They show that FKBP52 rescued GR ligand binding in the context of a complex containing tail helix-deleted P23, which has been previously shown by the authors to result in a decrease in ligand binding. However, FKBP52 cannot fully restore ligand binding in the absence of P23

completely and the authors suggest this is due to the requirement of P23 for full Hsp90 closure. They tested this by allowing full closure in the presence of molybdate and this resulted in full reactivation of GR ligand binding.

8. The authors provide an explanation of the differences in GR function observed between FKBP51 and FKBP52, where the difference observed in the structures appears to be in the contact between FK1 domains of FKBP51 and FKBP52 and GR. The authors have specifically focused on residue 119 in the proline rich loop region (which is Leucine in FKBP51 and Proline in FKBP52) showing that swapping the leucine for proline in the FKBP51 protein and proline for leucine in the FKBP52 protein reversed the effects of these proteins on GR ligand binding in-vitro. This has previously been shown by Riggs et al, Mol Cell Biol, 2007.

The manuscript is well written such that it is straight-forward to follow the experiments and data. The cryo-EM experiments are well carried out. Extensive analysis of the structures of both complexes is performed and a detailed description of the conformations and interaction regions between the components of the three-way complex is provided in the text, supported by a sufficient number of figures and supplementary figures, adequately enabling the reader to see details of the overall structures and interaction interfaces. Overall, the structures and GR-ligand binding results presented in this manuscript, which are supported by findings from Baischew et al, 2023, in their bioRxiv preprint, would be of interest to readers in the Steroid Hormone Receptor and Hsp90 fields.

Some further experiments are needed to support and validate some of the mechanistic conclusions made based on the structural information.

1. Additional data is needed to support the statement 'FKBP52 advances GR to the next stage of maturation' (line 197) which is also stated in the abstract. Can the authors explain how they come to this conclusion based on the data described here? In this section (line 197) the structural interactions between HSP90-GR and FKBP52 are reiterated and described. Then it is suggested that rotation of the GR observed in this structure may facilitate dimerization but again no data to support this.

2. Mutations have been introduced in 3 residues at three different FKBP52-GR interaction interfaces identified in the three-way complex (FKBP52 S118A, Y161D and W259D), and the effect of these mutations on potentiation of GR has been tested in each case. However, it has not been shown whether these mutations disrupt the direct interaction between FKBP52 and GR in-vitro or in-vivo in this work. Are the authors aware of any such data supporting this in the literature?

If not, this can be tested using the very nice in-vitro reconstitution system the authors have developed. They can introduce these mutated versions of FKBP52 (or an FKBP52 with a combination of two or three of the sites mutated) into their system instead of wild-type FKBP52 and test whether they can still pull-down FKBP52 with the MBP-GR. Likewise, the equivalent mutations can be introduced into FKBP51 to see if they disrupt binding to GR in the same way.

3. In both the FKBP52-GR-Hsp90 and the FKBP51-GR-Hsp90 complex structures, GR binds in a conformation where it is rotated such that it is proposed to no longer be compatible with P23 binding and Hsp90 favours binding to GR over p23. From these observations the

authors conclude FKBP52 competes with P23 for GR:HSP90 binding through allostery and further show that adding FKBP52 in the context of a complex containing the tail helix-deleted P23 (which has been previously shown to result in a decrease in ligand binding), resulted in full reactivation of GR ligand binding in the presence of a molybdate induced closure of the Hsp90.

Previous work by Ebong et al, Cell Discov, 2016, has shown that when adding FKBP52 to the complex, the p23 indeed cannot bind but on the contrary a FKBP51-GR-Hsp90 complex can bind P23. The authors observe no binding of P23 in their FKBP51-Hsp90-GR complex model here. Can they give a possible explanation for the difference? Looking at SEC gels in Extended data figure 1d in fact, it looks like more p23 is binding in the FKBP51 complex – can authors comment on this? Also, can they provide more explanation as to why p23 is present in the complex on the gels (although not in stoichiometric amounts with the other components) but no density is visible for p23 in the models of either complexes obtained from these samples? Could they probe the crosslinked complex with a p23 antibody to confirm the presence of p23 in the crosslinked sample that was applied to grids? Is it possible that there is a mixture of P23-Hsp90-GR and FKBP51/52-Hsp90-GR complexes in that size exclusion peak that were not resolved (Extended data figure 1d)?

For assembly of complexes for cryoEM studies, the FKBP51 and FKBP52 were in this case, incubated together with the cochaperones, p23 and GR. To support their conclusions that FKBP52 competes with P23 for binding, the authors can again make use of the in-vitro reconstitution system they have developed. In this case they can pre-assemble the maturation complex (Hsp90-GR-P23) and add excess FKBP51 or FKBP52 and pull-down on the MBP-GR to observe whether either of these competes off the P23. This would provide more clarity on whether FKBP52 can dissociate P23 from the pre-assembled complex, providing more verification of the order of binding events proposed and to address some of the differences observed between binding of P23 to FKBP52/FKBP51-Hsp90-GR in the structures in this work, compared with the previous work mentioned above, which shows P23 is retained in a FKBP51-Hsp90-GR complex.

MINOR COMMENTS

1. Figure 2a - Could the authors add in the figure legend what the pink density corresponds to.
2. Figure 4 a and b- The FKBP51 is incorrectly labelled as FKBP52. Also, could the FKBP52 and FKBP51 domains be labelled in figures 1a and 4b respectively (if they cannot be colored differently), to help in better visualising where each domain is located in the two complexes. Also, could GR be re-labelled as GR LBD in the structure figures as this is the only part visible in the structures.
3. Extended data figure 1c – Could GR DBD-LBD be re-labelled MBP-GR DBD-LBD to show the MBP tag is on the GR and this is what the complex is being pulled-down on.
4. Extended data figure 1d- Also need to add MBP to the label pointing to the GR band (it is MBP-GR DBD LBD being shown)
5. Extended data figure 1c – could a scale bar be included in the images
6. Can a figure be included to show local resolution on the surface of both the FKBP51 and FKBP52 complexes as there are differences in the resolution for the HSP90, GR and FKBP51/52.
7. Extended data table 1 - AF-Q02790. Please check pdb code.
8. References need to be double-checked as journal names are missing in some references (e.g. as in references 47 and 52)

Reviewer #3:

Remarks to the Author:

Remarks to the Author:

In this manuscript the authors continue their long-term structural study of Hsp90 client maturation with yet another solid piece of research concerning the chaperone cycle of the glucocorticoid receptor (GR).

Hsp90 is a highly expressed, highly regulated molecular chaperone involved in the folding, activation and degradation of over two hundred clients. The glucocorticoid receptor has been the subject of many studies due to its implication in stress response, amongst others, and peptidyl prolyl isomerases FKBP51 and FKBP52 are Hsp90 co-chaperones involved in the maturation cycle of GR. While FKBP51 is a GR repressor, FKBP52 enhances GR transcriptional activity. Despite a number of structural and biochemical studies having been published in the recent years there still are many unanswered questions regarding steroid hormone receptor maturation, but also regarding Hsp90 involvement in the life cycle of many clients. Here Noddings et al satisfactorily answer some of these questions, mainly by presenting two structures comprising the complexes HSP90:GR:FKBP52 and HSP90:GR:FKBP51.

The HSP90:GR:FKBP52 structure shows 3 interaction interfaces between GR and FKBP52. Analysis of the effect on GR activation of a series of mutants engineered to disrupt the binding surfaces between FKBP52 and GR validates the novel structural findings. The structure also explains why certain PPIase inhibitors block FKBP52-mediated potentiation of GR but PPIase mutants don't.

The authors propose that phosphorylation of FKBP52S118 may help promote the interaction between the proline-rich loop of FKBP52 and the GR. This hypothesis is partially backed by the GR activation assays in which a marked reduction in GR activation can be observed in the FKBP52S118A mutant when compared to wild type (Fig. 2e). This is in agreement with the effect on ligand binding seen by swapping amino acids between 51 and 52 (FKBP51L119P and FKBP52P119L in Fig. 4f). As the authors propose, it is likely that phosphorylation at FKBP52S118 by a proline-directed kinase promotes the FKBP52:GR interaction, but it would be desirable to test as well the effect on GR activation of phosphomimetic mutants FKBP52S118E and FKBP51S118E.

The HSP90:GR:FKBP52 structure provides a remarkable insight into the chaperone cycle of GR by showing its rotation in comparison with the HSP90:GR:p23 maturation complex previously published. This is mainly due to the differential positioning of the Src loop of Hsp90 and the translocation of 2 GR residues through the lumen of Hsp90. The ~45° rotation explains the competition observed between FKBP52 and p23 to bind the GR:Hsp90 complex.

It is surprising that there is competition between p23 and FKBP52 to bind GR. This would disprove previous suggestions that p23 and FKBP52 might bind simultaneously at each side of the Hsp90 dimer in the presence of the client, as seen in its absence. In this respect, the authors cite a mass spectrometry study supporting that binding of FKBP52 occurs concomitantly with release of p23 from the Hsp90:GR:p23 maturation complex (Ebong et al 2016). The same study though also observed the complex FKBP51:GR1:Hsp90:p23 (ligand-free GR is used) when challenging FKBP51:GR1:Hsp90:Hsp702:Hop1 complexes with p23. The binding mode is almost identical in the two structures presented (FKBP51/52-induced GR rotation dictates the accessibility of the hydrophobic patch to Hsp90 instead of p23), but could there be a number of intermediate complexes containing ligand-free GR, Hsp90, FKBP51 and p23? To explain the greater potentiation on GR ligand binding by FKBP52 in the p23Δhelix the

authors propose alleviation of the competition between p23 and FKBP52 which would result in p23 being able to remain in the complex and stabilize it. This would explain why a single p23 molecule bound at the other side of the Hsp90 dimer wouldn't have the same effect. This is further supported by the effect of Molybdate stabilising Hsp90 closure. The authors suggest that FKBP51/52 would promote GR dimerization. This is supported by a previous report that suggests they both have this effect on AR dimerization (Maeda et al 2022). It is worth noting though that in this study the PPIase activity of FKBP51 was seen to be essential for AR dimerization. This suggests that the GR rotated positioning in the presence of FKBP51/52 is not the sole driver for dimerization, at least not in the case of AR and FKBP51, and new questions arise about the PPIase activity of the two immunophilins since, at least at the stage captured by the structures presented in this study, GR is not bound to the PPIase site and no GR prolines are found to be isomerized. It is worth noting that this study and the manuscript available at BioRxiv by Baischew et al (2023) in which the complexes Hsp90:FKBP51:GR and Hsp90:FKBP52:GR are studied by *in vivo* photocrosslinking coincide and complement each other regarding their major findings.

Interestingly though, Baischew et al. see that stimulation with Dexamethasone causes the dissociation of the FKBP51:GR and FKBP52:GR complexes *in vivo*, suggesting that FKBP51/52 bind GR in the apo form. The structures here presented have been obtained in the presence of Dexamethasone and there is density in the maps that supports unambiguously GR being ligand-bound. The authors concede that apo GR:Hsp90:FKBP51/52 complexes likely exist in their dataset but are less well ordered and hence couldn't be resolved. Based on the ligand-bound structures they suggest that when the ligand-bound GR reaches the maturation complex (GR:Hsp90:p23) FKBP51/52 are able to bind to GR and the closed Hsp90 dimer, competing with p23 in order to advance GR to the next stage of maturation. Could the life expectancy of the ligand-bound complexes be very short *in vivo*, not allowing for FKBP51/52 exchange? Could ligand binding or unbinding occur in the context of FKBP-bound complexes, despite the structures showing GR so strongly stabilized by FKBP51/52?

Minor points:

This reviewer thanks the authors for the detailed and thorough description of contacts between the subunits of the complexes. The quality of the figures is excellent with a minor criticism of the colour scheme, as it works well for structures and diagrams, but certain colour combinations make it difficult to discern, in a printed manuscript, to which chain some amino acids belong. For example between Hsp90A, Hsp90B and FKBP52 in ED Fig. 4a. Perhaps slightly adjusting the text colour or outline would be sufficient to improve the readability of that figure. Similarly, in Fig. 5 cortisol and FKBP51 are depicted in nearly the same colour.

Please include maps of the main complexes coloured by local resolution in Extended Data. Figure 4a and 4b: FKBP51 labelled FKBP52.

Figure 4f: missing significance bracket between bars 4 and 6.

Figure 5: "others genes"

Line 919 and figure legends 2, 3, 4, ED 5 and ED 8: correct "equal or bigger" sign for not significant probabilities, as $P=0.05$ would be simultaneously significant and not significant. Materials and Methods: please add spaces between numerical values and units throughout.

Line 660: C (Celsius) superscript.

Reviewer #4:

Remarks to the Author:

How does the Hsp90 chaperone machinery regulate the steroid hormone receptors? This paper by Noddings et al uses structural and functional methods to address this fundamental question. The authors present a 3.01 Å cryo-EM structure of the glucocorticoid receptor (GR):Hsp90:FKBP52 complex and also 3.23 Å cryo-EM structure of the GR:Hsp90:FKBP51 complex. In both structures, FKBP51 and FKBP52 directly binds to the folded ligand bound GR and facilitate release of p23 through an allosteric mechanism. They also reveal that FKBP52, but not FKBP51, potentiates GR ligand binding in vitro, in a manner dependent on FKBP52-specific interactions. The authors also show that FKBP51 and FKBP52 both stabilize a rotated position of GR relative to the GR-maturation complex. They speculate that functional consequence of this rotated position may be to promote GR dimerization, which is a required step in GR activation. The rotated GR position relieves this steric hindrance to dimerization in the GR-maturation complex and would allow the GR LBD to dimerize once FKBP51/52 release.

These findings are exciting and there are so many significant take home messages in this manuscript. The conclusion is supported by high quality data and appropriate statistical analysis. The chaperone and SHR fields both will benefit from these findings. Therefore this paper is worthy of publication in NSMB. However, there are some points that needs to be clarified.

Comments

- Throughout the manuscript the authors refer to the GR-F602S mutant as GR. I ask the authors to change this throughout the manuscript.

- Line 817- For both the GR:Hsp90:FKBP51 and GR:Hsp90:FKBP52 complexes, no ligand-free GR complexes were identified during image analysis, despite many rounds of focused classification on GR at various stages of data processing. Were the authors expecting to see ligand-free GR complex? The authors have used dexamethasone at multiple steps of GR expression and purification. Doesn't this suggest that addition of ligand helps with the stability of GR?

- Line 686- GR:Hsp90:FKBP complex sample preparation methods- At which stage of this complex formation ligand was added. Or the GR that was purified was in the presence of dexamethasone? My point is that in a cellular context, GR does not bind to the ligand first and then to the chaperone complex.

- Line 191- "Thus, coactivator binding in the nucleus could help release GR from its complex". I have debated about this point with David previously- I just cannot see how this is possible. In reality and based on any methodologies addition of any ligand to Hsp90-SHR lead to their disassociation. In fact, I am more than convinced that this is the the case after reading this elegant paper.

- Line 47- "In vivo, additional Hsp90 co-chaperones are found associated with the GR-chaperone cycle". Please cite Backe et al 2022 Cell Rep. This paper also attempts to address the similar issue with the co-chaperones FNIP1 and FNIP2 and their impact towards SHR activation, activity and ligand binding.

- Figure 1a- Labeling of Hsp90 as A and B may create a confusion. Please label as alpha 1 and alpha 2 or a1 and a2.

- Figure 2e- GR activation assay in wild-type yeast- Is this a human GR and have they or any other group compared its activity to GR-F602S mutant? (no data required)

Figure 4- Please provide a less elaborate schematic figure and certainly not in a cellular context. This may create a problem/confusion for the subsequent stories and other research groups that aim to decipher the role of Hsp90 in SHR activation and activity in mammalian cells.

Author Rebuttal to Initial comments

Below is a point-by-point response to the reviewers' comments.

Reviewer #1:

Remarks to the Author:

They authors present the cryo-EM structure of a complex consisting of the co-chaperones FKBP51 or FKBP52 bound to Hsp90 and the glucocorticoid receptor (GR). This complex is part of the maturation cycle of GR providing an additional regulation step in the signaling of this steroid hormone receptor. Despite high similarity in sequence and structure, FKBP51 and FKBP52 show antagonistic effects on the activity of GR in vivo. Interestingly, the two structures reveal that both FKBP51 or FKBP52 interact directly with the GR bound to Hsp90. Although this interaction is similar for FKBP51 and FKBP52 as deduced from the cryo-EM structure, the effects on ligand binding by GR are different which suggests that subtle differences in binding exist. This is an interesting, well performed study that provides new insight into the structure and function of the Hsp90 system and GR signaling.

We thank the reviewer for their comments.

Specific comments:

1. The conclusion that FKBP52 functionally replaces p23 tail-helix is plausible but not demonstrated directly in this study. Ligand binding was increased in both settings, but the underlying mechanisms could be different.

We agree with the reviewer that FKBP52 may increase GR ligand binding through a different mechanism than that of the p23 tail-helix. Given that the p23 tail-helix directly binds the ligand-bound, folded GR LBD, we hypothesized that the p23 tail-helix acts by stabilizing the ligand-bound, folded GR LBD, thus increasing GR ligand binding activity (Noddings et al. 2022). Our structure of GR:Hsp90:FKBP52 also shows that FKBP52 directly binds the ligand-bound, folded GR LBD, suggesting FKBP52 may also increase GR ligand binding by stabilizing this conformation of the GR LBD. However, the p23 tail-helix and FKBP52 bind distinct locations on the GR LBD and thus must stabilize the GR LBD through different means. In the FKBP52 complex, Hsp90 now makes interactions with the GR hydrophobic patch, which was previously used as a binding site by the p23 tail-helix. The rotation of GR and occlusion of the p23 tail-helix binding site by Hsp90 likely lead to the loss of p23 in the GR:Hsp90:FKBP52 complex.

We use the term “functional replacement”, not to indicate the mechanisms are necessarily the same, but to indicate that FKBP52 can functionally complement the p23 Δ helix mutant by rescuing GR ligand binding, which we demonstrate in Fig. 3e.

2. The extended figure 3a shows different orientations of the ATP for the two HSP90 protomers. Is that really the case? Is what the authors observe really ATP and not ADP + molybdate, as described by Gruszczuk et al. (2022 Nat Comm.) under similar conditions?

The two different orientations of the ATP in the two Hsp90 protomers represent two possible ATP orientations that were obtained from our modeling procedure using Rosetta and a half-map approach to prevent overfitting. Limitations in resolution of the nucleotide density make it difficult to unambiguously differentiate between these two orientations within the protomers. Furthermore, as discussed below, the density likely represents a mixture of ATP and ADP:molybdate states, making the specific nucleotide orientation difficult to assign.

The nucleotide bound to the Hsp90 protomers is likely ADP + molybdate; however, we cannot unambiguously assign the γ -phosphate density as molybdate. Our previous work demonstrated

that Hsp90 needs to hydrolyze at least one ATP to reach the GR-maturation complex (GR:Hsp90:p23) (Kirschke et al. 2014), strongly suggesting the nucleotide is in a hydrolyzed ADP:molybdate state. Furthermore, the γ -phosphate density is relatively strong compared to the rest of the map (Extended Data Fig. 3a), suggesting the presence of molybdate in at least some population of particles. However, Hsp90 may be in a hemi-hydrolyzed state with one protomer bound to ATP and one protomer bound to ADP:molybdate, which we cannot unambiguously determine from the reconstruction. Therefore, we have modeled ATP into the density, which is consistent with our treatment of the GR-maturation complex (GR:Hsp90:p23) (Noddings et al. 2022). Molybdate was also used to prepare the GR-maturation complex, but we modeled the nucleotide density as ATP, due to the ambiguity in accurately assigning this density.

3. In extended figure 3b, it would be useful to also show the GR conformation in the current complex vs its conformation in the previously published HSP90:GR:p23 complex.

We thank the reviewer for the suggestion and have now included this in Extended Data Fig. 3b.

4. One of the differences in the conditions used for the current structure vs the HSP90:FKBP51:p23 complex presented by Lee et al. (2021 Mol. Cell) is the use of ATP+molybdate rather than AMP-PNP. The ADP+molybdate state is supposed to correspond to a post-hydrolysis state while the AMP-PNP could be equivalent to a pre-hydrolysis state. Would it be possible that the exclusion of p23 from the client complex is rather due to the change in nucleotide state rather than to the rotation of the client? Would that fit to the results presented in figure 3f and extended data figure 8?

The exclusion of p23 from the client complex cannot be due to a change in nucleotide state because p23 can bind to Hsp90 in an ADP:molybdate state, as we previously showed for the GR-maturation complex (GR:Hsp90:p23) (Noddings et al. 2022). We acknowledge that p23 can also bind to Hsp90 in the AMP-PNP state; however, our previous work demonstrated that in the context of the GR maturation cycle, the p23-bound state can only be reached after ATP hydrolysis by Hsp90 (Kirschke et al. 2014).

We believe a more convincing explanation for the results in Fig. 3f and Extended Data Fig. 8 is that the tightly bound molybdate, like p23, stabilizes the closed state of Hsp90, which is a function the FKBP51s likely lack.

Reviewer #2:

Remarks to the Author:

The Glucocorticoid receptor (GR) is highly dependent on HSP70 and Hsp90 chaperone complexes for function. The authors of this manuscript have previously determined the cryoEM structures of a Hsp70-Hsp90-Hop-GR complex– referred to as the loading complex and the Hsp90-P23-GR maturation complex, which provide remarkable insight into how a partially unfolded GR is bound by Hsp70, which transfers it to a Hsp90-HOP complex in a state which ligand can bind to GR. The ligand bound GR is then shown to be in a folded state, stabilised by Hsp90 and P23 in the complex. The new findings presented in this manuscript are significant in further revealing how immunophilins FKBP51 and FKBP52 assemble into a Hsp90-GR complex, making direct interactions with both Hsp90 and GR in the regulation of the GR chaperone cycle.

Summary of key results:

1. Preparation of GR-Hsp90-FKBP51/FKBP52 complexes using an in-vitro reconstitution system developed in previous work by the authors, incorporating Hsp70, Hsp90, Hsp40, Hop, Hsp90,

P23, FKBP51 or FKBP52 and MBP-tagged GR DBD-LBD. This complex was stabilised with sodium molybdate and cross-linked for Cryo-EM studies.

2. These first CryoEM structures of Hsp90-GR-FKBP52 (3.01 Å) and Hsp90-GR-FKBP51 (3.23 Å) complexes reveal a fully closed Hsp90 dimer in complex with a single GR and single FKBP51 or FKBP52 bound and the absence of P23.

3. In these GR-client bound complex structures, FKBP51 and FKBP52 interact with Hsp90 in a similar manner as observed in a Hsp90-P23-FKBP51 structure previously by Lee et al, Mol Cell, 2021.

4. The structures reveal that in these Hsp90-GR-FKBP51/FKBP52 complexes, the GR client interacts with HSP90 through three interfaces and HSP90 stabilises the GR in a folded – ligand bound state distinct, from that observed in a previous GR-Hsp90-P23 complex structure determined (Noddings et al, Nature, 2022)

5. Completing a three-way interaction, the structures reveal that the FKBP51 and FKBP52 make direct contacts with the GR-client, identifying three main interaction interfaces between the two. To test interface 1, the authors show that a single point mutation in the proline rich loop of FKBP52 (S118A) reduces FKBP52 dependant GR potentiation and suggest that phosphorylation at this site may help promote the interaction between the proline rich loop and GR in-vivo. They further show that mutation of Y161D within interface 2 and W259D within the FKBP52-FK2-TPR linker which forms interface 3 with GR, also reduces FKBP52-dependant GR potentiation.

6. Despite P23 being present in the reconstitution mix when preparing the complex, no density is visible for the P23 in either structures and the authors suggest this is due to FKBP52 competing with P23 for GR-Hsp90 binding through allostery. It is proposed that binding of FKBP52 to the rotated position of GR observed in the client-bound complexes is not compatible with P23 binding.

7. Consistent with in-vivo findings (Riggs et al, EMBO, 2003), the authors found that FKBP52 binding to the GR-Hsp90 complex potentiates ligand binding in-vitro. They show that FKBP52 rescued GR ligand binding in the context of a complex containing tail helix-deleted P23, which has been previously shown by the authors to result in a decrease in ligand binding. However, FKBP52 cannot fully restore ligand binding in the absence of P23 completely and the authors suggest this is due to the requirement of P23 for full Hsp90 closure. They tested this by allowing full closure in the presence of molybdate and this resulted in full reactivation of GR ligand binding.

8. The authors provide an explanation of the differences in GR function observed between FKBP51 and FKBP52, where the difference observed in the structures appears to be in the contact between FK1 domains of FKBP51 and FKBP52 and GR. The authors have specifically focused on residue 119 in the proline rich loop region (which is Leucine in FKBP51 and Proline in FKBP52) showing that swapping the leucine for proline in the FKBP51 protein and proline for leucine in the FKBP52 protein reversed the effects of these proteins on GR ligand binding in-vitro. This has previously been shown by Riggs et al, Mol Cell Biol, 2007.

The manuscript is well written such that it is straight-forward to follow the experiments and data. The cryo-EM experiments are well carried out. Extensive analysis of the structures of both complexes is performed and a detailed description of the conformations and interaction regions between the components of the three-way complex is provided in the text, supported by a sufficient number of figures and supplementary figures, adequately enabling the reader to see details of the overall structures and interaction interfaces. Overall, the structures and GR-ligand binding results presented in this manuscript, which are supported by findings from Baischew et al, 2023, in their bioRxiv preprint, would be of interest to readers in the Steroid Hormone Receptor and Hsp90 fields.

We thank the reviewer for their comments.

Some further experiments are needed to support and validate some of the mechanistic conclusions made based on the structural information.

1. Additional data is needed to support the statement 'FKBP52 advances GR to the next stage of maturation' (line 197) which is also stated in the abstract. Can the authors explain how they come to this conclusion based on the data described here? In this section (line 197) the structural interactions between HSP90-GR and FKBP52 are reiterated and described. Then it is suggested that rotation of the GR observed in this structure may facilitate dimerization but again no data to support this.

We propose that FKBP52 “advances GR to the next stage of maturation” given that binding of FKBP52 represents a new step in the GR chaperone cycle that is on pathway to GR activation. Our biochemical data, combined with previous literature, demonstrates that FKBP52 potentiates GR ligand binding and transcription activation. Our structure reveals that FKBP52 likely potentiates ligand binding by stabilizing the ligand-bound, folded conformation of GR in a complex distinct from that of the GR-maturation complex (GR:Hsp90:p23).

In the GR:Hsp90:FKBP52 complex, GR is further translocated out of the Hsp90 lumen, allowing GR to rotate and adopt a new binding position on Hsp90, which we suggest may facilitate GR dimerization on pathway to GR activation. We have added a sentence at line 219 to clarify that the GR LBD dimer interface was previously occluded in the GR-maturation complex (GR:Hsp90:p23) and only becomes accessible upon GR rotation after FKBP52 binding. We have also added that this rotation of the GR LBD may play other functional roles, such as facilitating association with other binding partners or placement of PTMs, and we have added this to the text at line 222. We acknowledge that we do not know the functional consequence of the new GR LBD binding position on Hsp90 and this aspect will be interesting to explore in future studies.

2. Mutations have been introduced in 3 residues at three different FKBP52-GR interaction interfaces identified in the three-way complex (FKBP52 S118A, Y161D and W259D), and the effect of these mutations on potentiation of GR has been tested in each case. However, it has not been shown whether these mutations disrupt the direct interaction between FKBP52 and GR in-vitro or in-vivo in this work. Are the authors aware of any such data supporting this in the literature?

If not, this can be tested using the very nice in-vitro reconstitution system the authors have developed. They can introduce these mutated versions of FKBP52 (or an FKBP52 with a combination of two or three of the sites mutated) into their system instead of wild-type FKBP52 and test whether they can still pull-down FKBP52 with the MBP-GR. Likewise, the equivalent mutations can be introduced into FKBP51 to see if they disrupt binding to GR in the same way.

Previous data demonstrates that the FKBP51/52 TPR domain alone binds Hsp90:SHR complexes with similar apparent affinity to full-length FKBP (Cheung-Flynn et al. 2003). This is likely due to the extensive interface formed between the Hsp90 CTD and the FKBP TPR C-terminus/H7e revealed in our structures and Lee et al. 2021 (Hsp90:FKBP51:p23). In fact, even in the absence of interactions with a client, FKBP51 efficiently binds the Hsp90 dimer through this TPR/H7e domain interface (Lee et al. 2021). Thus, the other regions of FKBP52 (FK1, FK2, and the N-terminus of the TPR domain) appear dispensable for binding to Hsp90 and therefore, the FKBP52 single point mutants likely do not disrupt FKBP52 binding to Hsp90 in the

GR:Hsp90 complex. Instead, these point mutants are likely unable to productively bind the GR LBD, thus preventing stabilization of the ligand-bound, folded GR and resulting in the lower GR transcriptional activation *in vivo* (Fig. 2e). Furthermore, we also note that FKBP52 Y161 and W259 were both identified as sites for *in vivo* crosslinking to the GR LBD in the accompanying manuscript (Baischew et al.) (note that FKBP52 S118 was not tested for crosslinking). The corresponding sites on FKBP51 (S118, Y159, W257) were also identified as sites that crosslink to the GR LBD *in vivo*. We have added a sentence at line 198 to clarify the proposed mechanism of the FKBP52 mutants on GR activity and added on line 195 that these residues were specifically identified as sites of crosslinking to the GR LBD in the accompanying manuscript (Baischew et al.).

3. In both the FKBP52-GR-Hsp90 and the FKBP51-GR-Hsp90 complex structures, GR binds in a conformation where is rotated such that it is proposed to no longer be compatible with P23 binding and Hsp90 favours binding to GR over p23. From these observations the authors conclude FKBP52 competes with P23 for GR:HSP90 binding through allostery and further show that adding FKBP52 in the context of a complex containing the tail helix-deleted P23 (which has been previously shown to result in a decrease in ligand binding), resulted in full reactivation of GR ligand binding in the presence of a molybdate induced closure of the Hsp90. Previous work by Ebong et al, Cell Discov, 2016, has shown that when adding FKBP52 to the complex, the p23 indeed cannot bind but on the contrary a FKBP51-GR-Hsp90 complex can bind P23. The authors observe no binding of P23 in their FKBP51-Hsp90-GR complex model here. Can they give a possible explanation for the difference? Looking at SEC gels in Extended data figure 1d in fact, it looks like more p23 is binding in the FKBP51 complex – can authors comment on this? Also, can they provide more explanation as to why p23 is present in the complex on the gels (although not in stoichiometric amounts with the other components) but no density is visible for p23 in the models of either complexes obtained from these samples? Could they probe the crosslinked complex with a p23 antibody to confirm the presence of p23 in the crosslinked sample that was applied to grids? Is it possible that there is a mixture of P23-Hsp90-GR and FKBP51/52-Hsp90-GR complexes in that size exclusion peak that were not resolved (Extended data figure 1d)?

We do not have a satisfactory explanation as to why the previous work by Ebong et al. 2016 shows that FKBP52 competes with p23, but FKBP51 does not. Given that these two proteins are extremely similar in sequence and secondary structure, and that our structures reveal these two proteins bind in a largely similar manner to the GR:Hsp90 complex, it is unclear how these two proteins would have differential effects on p23 binding. Furthermore, in the Ebong et al. 2016 study, the GR:Hsp90:p23:FKBP51 contains two copies of p23 bound, which is also puzzling given that our previous structure of the GR-maturation complex (GR:Hsp90:p23), as well as the Hsp90:p23:FKBP51 cryo-EM structure, both only contain one copy of p23 (Lee et al. 2021).

It is possible that FKBP51 is not as potent at competing off p23 as FKBP52. Indeed, as the reviewer points out, the SEC gels in Extended Data Fig. 1d suggest more p23 co-elutes with the FKBP51 complex compared to FKBP52; however, without more quantitative measurements, we hesitate to make this claim.

As the reviewer also suggests, the GR:Hsp90:p23 and GR:Hsp90:FKBP complexes are not resolved on the SEC due to their similar molecular weight and thus co-elute. In fact, during processing of the GR:Hsp90:FKBP51 dataset, 3D classification yielded both GR:Hsp90:p23 and GR:Hsp90:FKBP51 complexes, which we now discuss at line 294 and highlight in Extended Data Fig. 6. GR:Hsp90:p23 complexes also exist in the FKBP52 dataset at low abundance

(~74,000 particles), which we discuss at line 228 and have now highlighted in Extended Data Fig. 2. The differences in the amount of GR:Hsp90:p23 complex remaining in the FKBP51 versus FKBP52 sample may suggest FKBP52 competes off p23 more readily than FKBP51, but we hesitate to make this suggestion without further evidence.

For assembly of complexes for cryoEM studies, the FKBP51 and FKBP52 were in this case, incubated together with the cochaperones, p23 and GR. To support their conclusions that FKBP52 competes with P23 for binding, the authors can again make use of the in-vitro reconstitution system they have developed. In this case they can pre-assemble the maturation complex (Hsp90-GR-P23) and add excess FKBP51 or FKBP52 and pull-down on the MBP-GR to observe whether either of these competes off the P23. This would provide more clarity on whether FKBP52 can dissociate P23 from the pre-assembled complex, providing more verification of the order of binding events proposed and to address some of the differences observed between binding of P23 to FKBP52/FKBP51-Hsp90-GR in the structures in this work, compared with the previous work mentioned above, which shows P23 is retained in a FKBP51-Hsp90-GR complex.

We thank the reviewer for this suggestion and agree that directly showing competition between p23 and the FKBP5s would be informative. We attempted the experiment suggested by the reviewer (Supplementary Fig. 1), where we first pre-formed the GR-maturation complex (GR:Hsp90:p23) with 5 μ M GR, 2 μ M Hsp40, 5 μ M Hsp70, 5 μ M Hop, 15 μ M Hsp90, and 5 μ M p23. We then added 40 μ M FKBP51 or FKBP52, incubated the mixture for 30 minutes at room temperature, and pulled down on MBP-GR. The elution revealed that fairly equal amounts of p23 were present in all reactions (Reactions 1-3) and this was not due to non-specific binding of p23 to the resin (Reaction 4).

Unfortunately, due to the complexity of the GR chaperone cycle, a competition binding assay is difficult to perform because high concentrations of p23 (at least 5 μ M) are required to first reach the GR-maturation complex (Kirschke et al. 2014). Thus, there is a relatively large excess of free p23 in the reaction.

To overcome this problem, we also tried pre-forming the GR-maturation complex, then incubating this complex with amylose resin to bind the MBP-GR maturation complex to the resin (Supplementary Fig. 2). We then added 40 μ M FKBP51 or FKBP52 into the wash buffer to compete off the bound p23 and then eluted the remaining complex. We used SYPRO Ruby to stain the gel for increased sensitivity. Unfortunately, once again, the gel revealed that fairly equal amounts of p23 were present in all reactions (Reactions 1-3) and this was not due to non-specific binding of p23 to the resin (Reaction 4).

Likely, a more sensitive competition binding assay is needed, such as a FRET-based assay or fluorescence anisotropy-based assay. Our cryo-EM structures clearly show that the dominant complexes in our solution contain either p23 or an FKBP, thus highly suggesting these co-chaperones compete for binding. Our structures also provide an explanation as to why p23 and FKBP are mutually exclusive, due to the different position of GR in these two complexes, which can either be bound by the p23 tail-helix or FKBP. This model also fits well with our biochemical data showing that potentiation of GR ligand binding by FKBP52 is further increased in the p23 Δ helix background. In our model, the p23 Δ helix mutant and FKBP52 should be able to bind simultaneously to the GR:Hsp90 complex, resulting in enhanced ligand binding activity.

MINOR COMMENTS

1. *Figure 2a - Could the authors add in the figure legend what the pink density corresponds to.*

This has now been added to the figure legend.

2. *Figure 4 a and b– The FKBP51 is incorrectly labelled as FKBP52. Also, could the FKBP52 and FKBP51 domains be labelled in figures 1a and 4b respectively (if they cannot be colored differently), to help in better visualising where each domain is located in the two complexes. Also, could GR be re-labelled as GR LBD in the structure figures as this is the only part visible in the structures.*

We thank the reviewer for catching this error. We have made these corrections.

3. *Extended data figure 1c – Could GR DBD-LBD be re-labelled MBP-GR DBD-LBD to show the MBP tag is on the GR and this is what the complex is being pulled-down on.*

We have added this label.

4. *Extended data figure 1d- Also need to add MBP to the label pointing to the GR band (it is MBP-GR DBD LBD being shown)*

We have added this label.

5. *Extended data figure 1c – could a scale bar be included in the images*

We believe the reviewer was referring to the micrographs in Fig. 1e. We have now added scale bars.

6. *Can a figure be included to show local resolution on the surface of both the FKBP51 and FKBP52 complexes as there are differences in the resolution for the HSP90, GR and FKBP51/52.*

We have added local resolution maps for both complexes.

7. *Extended data table 1 - AF-Q02790. Please check pdb code.*

AF-Q02790 is not a PDB code, but rather an AlphaFold model identifier. We have indicated this on Extended Data Table 1.

8. *References need to be double-checked as journal names are missing in some references (e.g. as in references 47 and 52)*

We have corrected these references.

Reviewer #3:

Remarks to the Author:

Remarks to the Author:

In this manuscript the authors continue their long-term structural study of Hsp90 client

maturation with yet another solid piece of research concerning the chaperone cycle of the glucocorticoid receptor (GR).

Hsp90 is a highly expressed, highly regulated molecular chaperone involved in the folding, activation and degradation of over two hundred clients. The glucocorticoid receptor has been the subject of many studies due to its implication in stress response, amongst others, and peptidyl prolyl isomerases FKBP51 and FKBP52 are Hsp90 co-chaperones involved in the maturation cycle of GR. While FKBP51 is a GR repressor, FKBP52 enhances GR transcriptional activity. Despite a number of structural and biochemical studies having been published in the recent years there still are many unanswered questions regarding steroid hormone receptor maturation, but also regarding Hsp90 involvement in the life cycle of many clients. Here Noddings et al satisfactorily answer some of these questions, mainly by presenting two structures comprising the complexes HSP90:GR:FKBP52 and HSP90:GR:FKBP51. The HSP90:GR:FKBP52 structure shows 3 interaction interfaces between GR and FKBP52. Analysis of the effect on GR activation of a series of mutants engineered to disrupt the binding surfaces between FKBP52 and GR validates the novel structural findings. The structure also explains why certain PPIase inhibitors block FKBP52-mediated potentiation of GR but PPIase mutants don't.

The authors propose that phosphorylation of FKBP52S118 may help promote the interaction between the proline-rich loop of FKBP52 and the GR. This hypothesis is partially backed by the GR activation assays in which a marked reduction in GR activation can be observed in the FKBP52S118A mutant when compared to wild type (Fig. 2e). This is in agreement with the effect on ligand binding seen by swapping amino acids between 51 and 52 (FKBP51L119P and FKBP52P119L in Fig. 4f). As the authors propose, it is likely that phosphorylation at FKBP52S118 by a proline-directed kinase promotes the FKBP52:GR interaction, but it would be desirable to test as well the effect on GR activation of phosphomimetic mutants FKBP52S118E and FKBP51S118E.

We thank the reviewer for this suggestion and believe understanding how differential phosphorylation of FKBP51 versus FKBP52 may affect GR activity is an important next step. However, we point out that our results in Fig. 4f show that FKBP residue 119 has an effect on GR ligand binding independent of the phosphorylation state of S118, as we purified the FKBP mutants from *E. coli* for our *in vitro* GR ligand binding assay. We propose that a leucine versus a proline at residue 119 alters the dynamics of the proline-rich loop, as previously shown through NMR studies (Mustafi et al. 2014), and this determines, in part, how the FKBP5s affect GR activation.

We suggested that phosphorylation at S118 may play an additional role in FKBP52-dependent potentiation of GR due to our surprising result showing the FKBP52 S118A mutant is unable to effectively potentiate GR *in vivo* despite this rather benign point mutation. We were further encouraged to see that the qPTM database indicates FKBP52 is phosphorylated at S118 (while FKBP51 is not) and this might help explain the large effect of this seemingly benign mutation.

We think testing FKBP51/52 phosphomimetics would be quite interesting, but we are concerned that testing these mutants in our GR activation assay in yeast may not yield interpretable results. For example, if FKBP52 S118E has the same effect on GR activation as FKBP52 wild-type, this could be because FKBP52 S118 is already highly phosphorylated in the yeast assay, or alternatively this result could indicate that phosphorylation at S118 has no effect. If FKBP52 S118E has a large effect on GR activation compared to FKBP52 wild-type, this would suggest FKBP52 S118 is minimally phosphorylated in the yeast assay, but then this would contradict the large effect of the S118A mutation, warranting further experiments.

In the case of FKBP51 phosphomimetics, our GR activation assay in yeast is not ideal. In yeast, FKBP51 has no effect on GR activity, while in mammalian cells FKBP51 is a negative regulator of GR activity (Denny et al. 2000, Wochnik et al. 2005). Due to this discrepancy, the effect of an FKBP51 phosphomimetic in yeast on GR activity may not be relevant in mammalian cells. As such it would be quite useful to study the effect of FKBP51/52 phosphomimetics in a mammalian cell-based GR activation assay, but we believe this out of the scope of our study.

The HSP90:GR:FKBP52 structure provides a remarkable insight into the chaperone cycle of GR by showing its rotation in comparison with the HSP90:GR:p23 maturation complex previously published. This is mainly due to the differential positioning of the Src loop of Hsp90 and the translocation of 2 GR residues through the lumen of Hsp90. The ~45° rotation explains the competition observed between FKBP52 and p23 to bind the GR:Hsp90 complex. It is surprising that there is competition between p23 and FKBP52 to bind GR. This would disprove previous suggestions that p23 and FKBP52 might bind simultaneously at each side of the Hsp90 dimer in the presence of the client, as seen in its absence. In this respect, the authors cite a mass spectrometry study supporting that binding of FKBP52 occurs concomitantly with release of p23 from the Hsp90:GR:p23 maturation complex (Ebong et al 2016). The same study though also observed the complex FKBP511:GR1:Hsp902:p232 (ligand-free GR is used) when challenging FKBP511:GR1:Hsp902:Hsp702:Hop1 complexes with p23. The binding mode is almost identical in the two structures presented (FKBP51/52-induced GR rotation dictates the accessibility of the hydrophobic patch to Hsp90 instead of p23), but could there be a number of intermediate complexes containing ligand-free GR, Hsp90, FKBP51 and p23?

We cannot exclude the possibility that there are transient intermediate complexes that contain ligand-free GR, Hsp90, FKBP51, and p23. These intermediate complexes may not be apparent in our cryo-EM data processing if these complexes are highly dynamic, heterogenous, and/or low abundance. As we discuss in the text, we cannot obtain high-resolution reconstructions of ligand-free GR in our cryo-EM dataset, likely due to the heterogenous and dynamic nature of the ligand-free GR. However, we point out that the accompanying manuscript (Baischew et al.) performed *in vivo* crosslinking between the FKBP51 and GR in the absence of ligand. The crosslinks match our ligand-bound GR:FKBP51 complex reconstructions remarkably well, indicating that the FKBP51s interact with ligand-bound and ligand-free GR with largely similar binding interfaces, consistent with a rotated position of GR in both states. Based on this data, we do not believe the ligand-free GR:Hsp90:FKBP51 complexes would be compatible with p23 binding. However, once again, we cannot exclude the possibility of transient intermediate complexes that may contain FKBP51 and p23 simultaneously bound.

To explain the greater potentiation on GR ligand binding by FKBP52 in the p23Δhelix the authors propose alleviation of the competition between p23 and FKBP52 which would result in p23 being able to remain in the complex and stabilize it. This would explain why a single p23 molecule bound at the other side of the Hsp90 dimer wouldn't have the same effect. This is further supported by the effect of Molybdate stabilising Hsp90 closure.

The authors suggest that FKBP51/52 would promote GR dimerization. This is supported by a previous report that suggests they both have this effect on AR dimerization (Maeda et al 2022). It is worth noting though that in this study the PPIase activity of FKBP51 was seen to be essential for AR dimerization. This suggests that the GR rotated positioning in the presence of FKBP51/52 is not the sole driver for dimerization, at least not in the case of AR and FKBP51, and new questions arise about the PPIase activity of the two immunophilins since, at least at the stage captured by the structures presented in this study, GR is not bound to the PPIase site and no GR prolines are found to be isomerized.

It is worth noting that this study and the manuscript available at BioRxiv by Baischew et al (2023) in which the complexes Hsp90:FKBP51:GR and Hsp90:FKBP52:GR are studied by in vivo photocrosslinking coincide and complement each other regarding their major findings. Interestingly though, Baischew et al. see that stimulation with Dexamethasone causes the dissociation of the FKBP51:GR and FKBP52:GR complexes in vivo, suggesting that FKBP51/52 bind GR in the apo form. The structures here presented have been obtained in the presence of Dexamethasone and there is density in the maps that supports unambiguously GR being ligand-bound. The authors concede that apo GR:Hsp90:FKBP51/52 complexes likely exist in their dataset but are less well ordered and hence couldn't be resolved. Based on the ligand-bound structures they suggest that when the ligand-bound GR reaches the maturation complex (GR:Hsp90:p23) FKBP51/52 are able to bind to GR and the closed Hsp90 dimer, competing with p23 in order to advance GR to the next stage of maturation. Could the life expectancy of the ligand-bound complexes be very short in vivo, not allowing for FKBP51/52 exchange?

We believe the ligand-bound GR:Hsp90 complexes are short-lived *in vivo*, as previous studies have shown GR is rapidly translocated to the nucleus upon ligand addition ($t_{1/2} \approx 5$ minutes) (Galigniana et al. 1998; Galigniana et al. 2001). Once in the nucleus, GR presumably disassembles from the Hsp90 complex and binds DNA to regulate transcription.

We believe the ligand-dependent dissociation of GR:FKBP complexes *in vivo* is likely due to the short-lived nature of these complexes, which cannot be captured quickly enough to detect. We point out that Baischew et al. also acknowledge this result was surprising given the previous data that FKBP52 facilitates translocation of the ligand-bound GR to the nucleus (Galigniana et al. 2001, Wochnik et al. 2005, Tatro et al. 2009, Galigniana et al. 2010). In addition, a previous study (Davies et al. 2002) reports a ligand-induced switching from FKBP51 to FKBP52 in GR:Hsp90 complexes. In this study, ligand was added to intact cells at 4°C for 3 hours to capture ligand-bound GR:Hsp90 complexes before GR dissociation. Using this procedure, GR:Hsp90:FKBP complexes were efficiently co-immunoprecipitated and this study does not report ligand-induced dissociation of GR:Hsp90:FKBP complexes when cells are incubated at 4°C.

Furthermore, Baischew et al. also reveal that the GR:p23 interaction dissociates upon ligand addition, which contradicts our previous structure of ligand-bound GR:Hsp90:p23 (GR-maturation complex) (Noddings et al. 2022). From our current studies, we would expect p23 to dissociate in favor of FKBP52 binding. Once again, we believe the inability to capture GR:Hsp90:p23 or GR:Hsp90:FKBP complexes *in vivo* is due to the short lifetime of the ligand-bound GR complexes upon rapid translocation to the nucleus and release of GR. We have now clarified this point at line 359 in the discussion.

Could ligand binding or unbinding occur in the context of FKBP-bound complexes, despite the structures showing GR so strongly stabilized by FKBP51/52?

We do not believe ligand binding or unbinding can occur in the context of the FKBP-bound complexes, as we mention at line 408 in the discussion. Both FKBP51 and FKBP52 extensively wrap around the GR LBD, making it difficult to imagine how the ligand-binding pocket would open to allow ligand binding/unbinding. Furthermore, as discussed in our previous manuscripts (Wang et al. 2022, Noddings et al. 2022), our previous structures reveal that the Helix 1 region likely needs to unfold from the body of the GR LBD to allow for ligand binding/unbinding from the buried, hydrophobic ligand-binding pocket. This result is strongly supported by single-molecule force spectroscopy studies (Suren et al. 2018), indicating that the Helix 1 region on the GR LBD functions as a lid over ligand-binding pocket, which must undock to allow ligand

release. From our FKBP complex structures, it is difficult to imagine how Helix 1 would undock from the GR LBD while the FKBP5s remain bound and thus we hypothesize that ligand binding/unbinding does not occur within these complexes.

Minor points:

This reviewer thanks the authors for the detailed and thorough description of contacts between the subunits of the complexes. The quality of the figures is excellent with a minor criticism of the colour scheme, as it works well for structures and diagrams, but certain colour combinations make it difficult to discern, in a printed manuscript, to which chain some amino acids belong. For example between Hsp90A, Hsp90B and FKBP52 in ED Fig. 4a. Perhaps slightly adjusting the text colour or outline would be sufficient to improve the readability of that figure. Similarly, in Fig. 5 cortisol and FKBP51 are depicted in nearly the same colour.

We have adjusted the text color and outline to improve the readability of the residue labels.

Please include maps of the main complexes coloured by local resolution in Extended Data.

We have now included local resolution maps in the Extended Data.

Figure 4a and 4b: FKBP51 labelled FKBP52.

We thank the reviewer for catching this error and have made this correction.

Figure 4f: missing significance bracket between bars 4 and 6.

We have added significance brackets to bars 4 and 6.

Figure 5: "others genes"

We have corrected this error.

Line 919 and figure legends 2, 3, 4, ED 5 and ED 8: correct "equal or bigger" sign for not significant probabilities, as $P=0.05$ would be simultaneously significant and not significant.

We have made this correction.

Materials and Methods: please add spaces between numerical values and units throughout.

We have made this correction.

Line 660: C (Celsius) superscript.

We have made this correction.

Reviewer #4:

Remarks to the Author:

How does the Hsp90 chaperone machinery regulate the steroid hormone receptors? This paper by Noddings et al uses structural and functional methods to address this fundamental question. The authors present a 3.01 Å cryo-EM structure of the glucocorticoid receptor (GR):Hsp90:FKBP52 complex and also 3.23 Å cryo-EM structure of the GR:Hsp90:FKBP51

complex. In both structures, FKBP51 and FKBP52 directly binds to the folded ligand bound GR and facilitate release of p23 through an allosteric mechanism. They also reveal that FKBP52, but not FKBP51, potentiates GR ligand binding in vitro, in a manner dependent on FKBP52-specific interactions. The authors also show that FKBP51 and FKBP52 both stabilize a rotated position of GR relative to the GR-maturation complex. They speculate that functional consequence of this rotated position may be to promote GR dimerization, which is a required step in GR activation. The rotated GR position relieves this steric hindrance to dimerization in the GR-maturation complex and would allow the GR LBD to dimerize once FKBP51/52 release.

These findings are exciting and there are so many significant take home messages in this manuscript. The conclusion is supported by high quality data and appropriate statistical analysis. The chaperone and SHR fields both will benefit from these findings. Therefore this paper is worthy of publication in NSMB. However, there are some points that needs to be clarified.

Comments

- Throughout the manuscript the authors refer to the GR-F602S mutant as GR. I ask the authors to change this throughout the manuscript.

We have clarified on line 72 that the GR DBD-LBD construct contains the F602S mutation.

- Line 817- For both the GR:Hsp90:FKBP51 and GR:Hsp90:FKBP52 complexes, no ligand-free GR complexes were identified during image analysis, despite many rounds of focused classification on GR at various stages of data processing. Were the authors expecting to see ligand-free GR complex? The authors have used dexamethasone at multiple steps of GR expression and purification. Doesn't this suggest that addition of ligand helps with the stability of GR?

Yes, the GR DBD-LBD sample was expressed and purified with ligand present at multiple steps. Based on our previous structure of the GR-maturation complex, we were not necessarily expecting to see ligand-free GR complexes (Noddings et al. 2022). We previously noted that although the GR-maturation complex sample was enriched for apo GR (through a multiday dialysis after purification of GR), coherent density for ligand-free GR complexes could not be obtained despite multiple rounds of focused classification on the GR density. We hypothesized this was due to the dynamic and/or heterogenous of ligand-free GR.

For the FKBP complexes, we hypothesized that ligand-free GR may be stabilized by FKBP51, based on previous literature (Davies et al. 2002). Once again, despite multiple rounds of focused classification on the GR density, we were not able to obtain coherent density for ligand-free GR in the FKBP51 or FKBP52 complexes. These results suggest the addition of ligand helps with GR stability, as the reviewer suggests, and this observation is likely also consistent with the difficulty in purifying ligand-free GR.

- Line 686- GR:Hsp90:FKBP complex sample preparation methods- At which stage of this complex formation ligand was added. Or the GR that was purified was in the presence of dexamethasone? My point is that in a cellular context, GR does not bind to the ligand first and then to the chaperone complex.

GR DBD-LBD was purified in the presence of dexamethasone; however, additional ligand was not added during the *in vitro* reconstitution of the GR chaperone cycle to prepare the GR:Hsp90:FKBP complexes.

The reaction begins with purified GR likely as a mixture of ligand-bound and ligand-free states. Once GR enters the reconstituted GR chaperone cycle with high concentrations of Hsp70 present, GR ligand binding is inhibited (Kirschke et al. 2014) and the ligand-free GR is loaded onto Hsp90:Hsp70 to form the GR-loading complex (Wang et al. 2022). Next, Hsp70 and Hsp90 leave, Hsp90 closes, and p23 binds to form the GR-maturation complex (Noddings et al. 2022), in which GR is fully folded and bound to ligand. To form the FKBP complexes, FKBP51 and FKBP52 were added to the *in vitro* GR chaperone cycle from the beginning with all other components (Hsp70, Hsp40, Hsp90, Hop, p23, and Bag-1). Therefore, GR is allowed to cycle through the chaperones, such that ligand-free GR is loaded onto Hsp90 before the GR-maturation and GR:FKBP complexes are formed.

- Line 191- *“Thus, coactivator binding in the nucleus could help release GR from its complex”. I have debated about this point with David previously- I just cannot see how this is possible. In reality and based on any methodologies addition of any ligand to Hsp90-SHR lead to their disassociation. In fact, I am more than convinced that this is the the case after reading this elegant paper.*

We disagree with the reviewer that all methodologies demonstrate addition of any ligand to Hsp90:SHR complex leads to disassociation. We refer to our previous study (Noddings et al. 2022) and this manuscript, clearly demonstrating ligand-bound GR is in complex with Hsp90 *in vitro*. We also refer to a previous study, Davies et al. 2002, which reports the ligand-induced switching of FKBP51 to FKBP52 in GR:Hsp90 complexes *in vivo*. In this study, ligand was added to intact cells at 4°C for 3 hours to capture ligand-bound GR:Hsp90 complexes before GR dissociation to bind GREs in the nucleus. Using this procedure, GR:Hsp90:FKBP complexes were efficiently co-immunoprecipitated and this study does not report ligand-induced dissociation of GR:Hsp90:FKBP complexes when cells are incubated at 4°C. In support of this, other work in our lab has shown that at 4°C, Hsp90:Cdc37:kinase complexes fail to re-open and release kinase clients despite the Hsp90 ATP being hydrolyzed to ADP. However, warming to 30°C or above leads to complex dissociation.

We acknowledge that the co-submitted manuscript, Baischew et al., reports ligand-induced dissociation of GR:Hsp90:FKBP complexes. We believe the discrepancy is due to the ligand-dependent rapid translocation of GR to the nucleus and the comparatively long timeframe of their experiments. We have now clarified this point at line 359 in the discussion. Once in the nucleus and upon complete ATP hydrolysis on Hsp90, GR dissociates from Hsp90 complexes to bind GREs and regulate transcription. We believe GR dissociation from Hsp90 complexes may be based on a nucleus-specific signal, such as a nuclear-localized GR binding partner. We hypothesize that binding of coactivators to GR may facilitate GR dissociation from Hsp90:FKBP52 because GR:coactivator binding is known to take place in the nucleus at GREs. Furthermore, coactivator binding would sterically clash with FKBP52, based on our structure, possibly facilitating dissociation of Hsp90:FKBP52 from GR. Understanding exactly how and when GR is released from Hsp90 complexes is an important future direction for this research and will provide clarity as to how GR ultimately exits the chaperone cycle to regulate transcription.

- Line 47- *“In vivo, additional Hsp90 co-chaperones are found associated with the GR-chaperone cycle”. Please cite Backe et al 2022 Cell Rep. This paper also attempts to address*

the similar issue with the co-chaperones FNIP1 and FNIP2 and their impact towards SHR activation, activity and ligand binding.

We have added this citation.

- Figure 1a- Labeling of Hsp90 as A and B may create a confusion. Please label as alpha 1 and alpha 2 or a1 and a2.

To keep the labeling consistent with our previous publications (Wang et al. 2022, Noddings et al. 2022, Jaime-Garza et al. 2023) we prefer to label the protomers as Hsp90A and Hsp90B.

- Figure 2e- GR activation assay in wild-type yeast- Is this a human GR and have they or any other group compared its activity to GR-F602S mutant? (no data required)

For the GR activation assay in yeast, the full-length wild-type human GR was used and we have now indicated this in the Methods. Previously, the Buchner group determined that human GR F602S has ~4-fold higher transactivation activity in yeast compared to wild-type human GR (Lorenz et al. 2014).

Figure 4- Please provide a less elaborate schematic figure and certainly not in a cellular context. This may create a problem/confusion for the subsequent stories and other research groups that aim to decipher the role of Hsp90 in SHR activation and activity in mammalian cells.

We believe the reviewer is referring to the final figure, Fig. 5. This figure is elaborate due to the need to illustrate how the GR:Hsp90:FKBP complexes integrate within the larger GR chaperone cycle. This manuscript directly follows from our previous work (Kirschke et al. 2014, Wang et al. 2022, Noddings et al. 2022), which has provided biochemical and structural insight into the complete GR chaperone cycle. We believe the strength of this manuscript is in characterizing additional regulatory steps within the GR chaperone cycle due to the incorporation of the FKBP co-chaperones. Thus, we believe it is appropriate to have a somewhat elaborate final figure to fully encapsulate the complexity of the GR chaperone cycle.

Furthermore, we believe we are justified in showing the GR chaperone cycle within a cellular context to illustrate the spatial separation between the GR chaperone cycle and the activated GR within the nucleus. We want to highlight the full GR activation pathway and how Hsp90 and the FKBP co-chaperones fit within that pathway. Since GR nuclear translocation has been shown to be dependent on Hsp90 and FKBP52, we would like to show the events of nuclear translocation during the GR activation pathway.

To minimize confusion about the data from our study, we have made a few changes to the figure legend. We have changed the figure title to remove the words “*in vivo*”. Within the figure legend we have edited the first sentence to “Schematic depicting how the FKBP co-chaperones integrate with the GR chaperone cycle and how this cycle may take place within a cellular context”. We have also shortened the description and added additional citations within the figure legend. We believe the figure provides a model that contextualizes our new data with the many decades of research on the GR chaperone cycle and bridges *in vitro* and *in vivo* studies into one model figure.

Decision Letter, first revision:

Message: Our ref: NSMB-A47236A

29th Jun 2023

Dear Dr. Agard,

Thank you for submitting your revised manuscript "Cryo-EM reveals how Hsp90 and FKBP immunophilins co-regulate the Glucocorticoid Receptor" (NSMB-A47236A). It has now been seen by the original referees and their comments are below. The reviewers find that the paper has improved in revision, and therefore we'll be happy in principle to publish it in Nature Structural & Molecular Biology, pending minor revisions to satisfy the referees' final requests and to comply with our editorial and formatting guidelines.

To facilitate our work at this stage, it is important that we have a copy of the main text as a word file. If you could please send along a word version of this file as soon as possible, we would greatly appreciate it; please make sure to copy the NSMB account (cc'ed above).

Sincerely,

Katarzyna Ciazynska
(she/her)
Associate Editor
Nature Structural & Molecular Biology
<https://orcid.org/0000-0002-9899-2428>

Reviewer #1 (Remarks to the Author):

The replies to my queries are appreciated. They explain the ambiguities connected with these points in detail and also the solutions the authors came up with. So this is fine. However, it seems these explanations and considerations were not integrated in the revised version. I think a discussion of these points in the final version will be useful for the readers.

Reviewer #2 (Remarks to the Author):

I feel the authors have successfully addressed comments 1 and 2.

I appreciate the efforts the authors have made in carrying out the experiments to show if adding excess FKBP to the GR maturation complex can result in release of P23 from the complex. They show 40uM concentration of FKBP (in 8 X excess of P23) could not compete off any P23. I feel the data presented here cannot conclusively support the conclusion that 'FKBPs facilitate the release of P23' bound to the complex:

Lines 24 and 25 in abstract – 'FKBP51 and FKBP52 directly engage the folded GR and unexpectedly facilitate release of p23 through an allosteric mechanism'

The structures nicely show how the binding of P23 and FKBP would be mutually exclusive, with binding of one hindering the binding of the other, but this does not directly show that FKBP competes off P23. The complexes here were assembled in the presence of both FKBP52 and P23 together and two complexes subsequently formed: one containing GR-HSP90-P23 and the predominant GR-HSP90-FKBP complex. Further the authors show that FKBP can bind a Molybdate-closed HSP90 and the presence of P23 is not required to assemble this complex or for GR activation. Again, this supports mutually exclusive binding but not necessarily that FKBP 'competes off' or releases the P23 in vitro or in vivo.

I feel it is important to distinguish between mutually exclusive binding which the data support, and competition with P23 already bound, as this is important for understanding order of events in the HSP90 cycle and SHR regulation.

Reviewer #3 (Remarks to the Author):

The authors have made the corrections I demanded and have satisfactorily addressed all my concerns. This piece of research should be published as soon as possible for the scientific community to see.

Reviewer #4 (Remarks to the Author):

The authors have addressed almost all my comments.
I was and still am very supportive of publishing this paper.

Author Rebuttal, first revision:

Reviewer #1 (Remarks to the Author):

The replies to my queries are appreciated. They explain the ambiguities connected with these points in detail and also the solutions the authors came up with. So this is fine. However, it seems these explanations and considerations were not integrated in the revised version. I think a discussion of these points in the final version will be useful for the readers.

We have added a sentence at line 240 to indicate that FKBP52 may increase GR ligand binding through a different mechanism than that of the p23 tail-helix given that these proteins bind distinct locations on the GR LBD.

We have added a paragraph in the Methods section at line 890 to acknowledge the ambiguity of the bound nucleotide on Hsp90 (ATP versus ADP-molybdate) and the rationale for our modeling choice. We have included a reference to this discussion at lines 73 and 279 in the main manuscript when describing the GR:Hsp90:FKBP52 and GR:Hsp90:FKBP51 structures, respectively.

Reviewer #2 (Remarks to the Author):

I feel the authors have successfully addressed comments 1 and 2.

I appreciate the efforts the authors have made in carrying out the experiments to show if adding excess FKBP to the GR maturation complex can result in release of P23 from the complex. They show 40uM concentration of FKBP (in 8 X excess of P23) could not compete off any P23. I feel the data presented here cannot conclusively support the conclusion that 'FKBPs facilitate the release of P23' bound to the complex:

Lines 24 and 25 in abstract – 'FKBP51 and FKBP52 directly engage the folded GR and unexpectedly facilitate release of p23 through an allosteric mechanism'

The structures nicely show how the binding of P23 and FKBP would be mutually exclusive, with binding of one hindering the binding of the other, but this does not directly show that FKBP competes off P23. The complexes here were assembled in the presence of both FKBP52 and P23 together and two complexes subsequently formed: one containing GR-HSP90-P23 and the predominant GR-HSP90-FKBP complex. Further the authors show that FKBP can bind a Molybdate-closed HSP90 and the presence of P23 is not required to assemble this complex or for GR activation. Again, this supports mutually exclusive binding but not necessarily that FKBP 'competes off' or releases the P23 in vitro or in vivo.

I feel it is important to distinguish between mutually exclusive binding which the data support, and competition with P23 already bound, as this is important for understanding order of events in the HSP90 cycle and SHR regulation.

We agree with the reviewer that we were unable to demonstrate that the FKBP51/52 "compete off" or "facilitate release of" p23 and therefore we have removed these phrases from our manuscript. Instead, we characterize the interplay of these proteins as competing to bind to the GR:Hsp90 complex since we demonstrate binding of p23 or FKBP51/52 is mutually exclusive.

We also would like to clarify to the reviewer that although we show the FKBP can bind to a molybdate-closed Hsp90 and reactivate GR without p23 present, as stated by the reviewer, this is not a physiologically relevant system given that the concentration of molybdate in the assay is 20 mM. We used molybdate as a method to stabilize closed Hsp90 in the absence of p23 to demonstrate that the FKBP can reactivate GR *if* Hsp90 closure is stabilized through another method. In the absence of molybdate or p23, we show GR ligand binding activity is very low and the FKBP cannot reactivate GR (Fig. 3f, Extended Data Fig. 8b). We reason this is because Hsp90 closure is not stabilized and therefore, FKBP binding is not possible. In a physiological system, p23 would first be needed to stabilize Hsp90 NTD closure to transition from the GR-loading complex to the GR-maturation complex, then the FKBP could bind to the closed Hsp90 CTD interface. Thus, there is likely a particular order of binding events (p23, then FKBP), but once p23 stabilizes Hsp90 closure, these complexes (GR:Hsp90:p23 and GR:Hsp90:FKBP) could exist in an equilibrium.

Reviewer #3 (Remarks to the Author):

The authors have made the corrections I demanded and have satisfactorily addressed all my concerns. This piece of research should be published as soon as possible for the scientific community to see.

We thank the reviewer for their comment.

Reviewer #4 (Remarks to the Author):

*The authors have addressed almost all my comments.
I was and still am very supportive of publishing this paper.*

We thank the reviewer for their comment.

Final Decision Letter:**Message** 18th Sep 2023

:

Dear Dr. Agard,

We are now happy to accept your revised paper "Cryo-EM reveals how Hsp90 and FKBP immunophilins co-regulate the Glucocorticoid Receptor" for publication as an Article in Nature Structural & Molecular Biology.

Your paper will be published online soon after we receive proof corrections and will appear in print in the next available issue. You can find out your date of online publication by

contacting the production team shortly after sending your proof corrections. Content is published online weekly on Mondays and Thursdays, and the embargo is set at 16:00 London time (GMT)/11:00 am US Eastern time (EST) on the day of publication. Now is the time to inform your Public Relations or Press Office about your paper, as they might be interested in promoting its publication. This will allow them time to prepare an accurate and satisfactory press release. Include your manuscript tracking number (NSMB-A47236B) and our journal name, which they will need when they contact our press office.

About one week before your paper is published online, we shall be distributing a press release to news organizations worldwide, which may very well include details of your work. We are happy for your institution or funding agency to prepare its own press release, but it must mention the embargo date and Nature Structural & Molecular Biology. If you or your Press Office have any enquiries in the meantime, please contact press@nature.com.

Please note that *Nature Structural & Molecular Biology* is a Transformative Journal (TJ). Authors may publish their research with us through the traditional subscription access route or make their paper immediately open access through payment of an article-processing charge (APC). Authors will not be required to make a final decision about access to their article until it has been accepted. <https://www.springernature.com/gp/open-research/transformative-journals> Find out more about Transformative Journals

Authors may need to take specific actions to achieve [compliance](https://www.springernature.com/gp/open-research/funding/policy-compliance-faqs) with funder and institutional open access mandates. If your research is supported by a funder that requires immediate open access (e.g. according to [Plan S principles](https://www.springernature.com/gp/open-research/plan-s-compliance)) then you should select the gold OA route, and we will direct you to the compliant route where possible. For authors selecting the subscription

publication route, the journal's standard licensing terms will need to be accepted, including [self-archiving policies](https://www.springernature.com/gp/open-research/policies/journal-policies). Those licensing terms will supersede any other terms that the author or any third party may assert apply to any version of the manuscript.

Sincerely,

Katarzyna Ciazynska
(she/her)
Associate Editor
Nature Structural & Molecular Biology
<https://orcid.org/0000-0002-9899-2428>
